# Regulatory role of the N-terminal intrinsically disordered region of the DEAD-box RNA helicase DDX3X in selective RNA recognition

Yuki Toyama [1,2] ✉, Koh Takeuchi [2] ✉ & Ichio Shimada [1,3] ✉

DDX3X, a member of the DEAD-box RNA helicase family, plays a central role in the translational regulation of gene expression through its unwinding activity toward complex RNA structures in messenger RNAs (mRNAs). Although DDX3X is known to selectively stimulate the translation of a subset of genes, a specific sequence motif has not been identified; thus, the molecular mechanism underlying this selectivity remains elusive. Using solution nuclear magnetic resonance (NMR) spectroscopy, we demonstrate that the N-terminal intrinsically disordered region (IDR) of DDX3X plays a critical role in the binding and unwinding of structured RNAs. We propose that the selectivity toward target transcripts is mediated by its preferential binding to structured motifs, particularly the G-quadruplex structure, through arginine-rich segments within the N-terminal IDR. Our results provide a molecular basis for understanding translational regulation by DDX3X and highlight the remarkable role of the flexible IDR in controlling the cellular translational landscape.

Gene expression is regulated through coupling with various stages of RNA metabolism, which include transcriptional control, splicing of precursor messenger RNA (mRNA), export of processed mRNA transcripts, translational control of mRNA, and mRNA decay. Many of these processes are finely regulated by numerous types of RNA-binding proteins, and understanding how these proteins specifically recognize RNA substrates at the molecular level would provide mechanistic insights into the elaborate regulation of gene expression. One of the key regulators is the DEAD-box RNA helicase (DDX), an enzyme that unwinds structured RNA into single-stranded RNA (ssRNA) in an ATP-dependent manner[1–4]. DDX constitutes the largest family of RNA helicases in humans and is involved in almost every aspect of RNA metabolism. The translation of mRNA transcripts, for example, is controlled by DDX-dependent unwinding of the higher-order structure in the 5′-untranslated region (5′-UTR), which otherwise negatively regulates translation by sterically inhibiting ribosome function. DDX3X, a member of the DDX family encoded on the X chromosome, is one of the most well-characterized translational initiators that stimulates the translation of mRNAs with long and structured 5′-UTR elements[5–9]. As commonly observed in many other DDX helicases, a number of studies have shown that DDX3X specifically regulates a subset of genes, allowing it to play crucial roles in the regulation of various physiological processes[5,7–12]. One such example is the specific translational regulation of the *RAC1* gene, which is involved in neurological development and the regulation of cell migration[7,12–17]. Notably, this specificity is observed both in vitro and in cell-based translational reporter assays[7,12], suggesting that the specificity is encoded at the level of the molecular interactions involving DDX3X and these transcripts. The physiological importance of the translational regulation dictated by DDX3X is supported by findings that the mutations causing DDX3X dysfunction are frequently linked to pathological processes such as tumor progression, developmental disorders, and intellectual disability[8,11,18–24].

[1]RIKEN Center for Biosystems Dynamics Research (BDR), Yokohama, Kanagawa, Japan. [2]Graduate School of Pharmaceutical Sciences, The University of Tokyo, Tokyo, Japan. [3]Graduate School of Integrated Sciences for Life, Hiroshima University, Higashi-Hiroshima, Japan. ✉e-mail: yuki.toyama@mol.f.u-tokyo.ac.jp; koh-takeuchi@mol.f.u-tokyo.ac.jp; ichio.shimada@riken.jp

Understanding how DDX3X selectively recognizes a subset of transcripts at the molecular level would contribute to our understanding of the diverse functional roles of DDX3X and potentially provide opportunities for developing therapeutic strategies for diseases caused by DDX3X malfunction. The 662-residue DDX3X protein contains a conserved folded core (residues 132–607) flanked by N- and C-terminal regions (Fig. 1a, Supplementary Fig. 1). The folded core region consists of tandem recombinase A (RecA)-like domains, D1 and D2, and is responsible for the ATP binding and helicase activity[2,25–27]. Both the N- and C-terminal regions are positively charged at physiological pH, with theoretical pIs of 9.47 and 8.48, respectively, and are predicted to form intrinsically disordered regions (IDRs)[21,28]. These IDRs play important roles in binding to the translation machinery[12,29], and in the formation of and localization to membrane-less organelles such as stress granules[21,28–31]. Previous structural studies, including ours, have demonstrated that the functional core region preferentially binds to the backbone sugar moiety of ssRNA to form a well-defined domain-closed structure[26,32,33]. This preferential binding to ssRNA serves as the driving force for its unwinding activity toward structured RNA molecules[34]. As expected from this binding mode, it has been widely believed that the DDX3X helicase core shows little to no sequence dependence in ssRNA recognition[2,3,9,35]. Consistently, a specific sequence motif recognized by DDX3X has not been identified in comprehensive cross-linking immunoprecipitation high-throughput sequencing (CLIP-seq) analyses of DDX3X targets so far[7,9,10,21]. These observations raise an important question: how does DDX3X specifically recognize a subset of mRNA molecules to be involved in physiological translational regulation?

In the DDX family, accessory domains outside the conserved core region are typically responsible for RNA specificity[1], suggesting that the N- and C-terminal IDRs would determine the substrate specificity of DDX3X. Consistent with this notion, Herdy et al. demonstrated that the arginine-glycine (RG) motifs in the N-terminal IDR are involved in binding to G-quadruplex (GQ) RNA found in the 5′-UTR of the *NRAS* transcript[36]. Additionally, in vitro characterizations of DDX3X helicase activity have shown that the N- and C-terminal IDRs greatly stimulate the helicase activity of the folded core, with the N-terminal IDR providing a larger contribution[26,37]. The functional importance of the N-terminal IDR has also been demonstrated in the context of DDX3X-RNA complex coacervation, which is related to the formation of membrane-less organelles such as stress granules[28,30,31]. These observations strongly suggest that the N- and C-terminal IDRs outside the folded core play critical roles in RNA recognition by DDX3X. However, since these flanking regions are predicted to be intrinsically disordered and do not form well-defined structures, how these IDRs recognize specific RNA targets needs to be understood beyond the classical structure-function paradigm.

In this study, we dissected the role of the N-terminal IDR of DDX3X in specific RNA recognition using solution nuclear magnetic resonance (NMR) spectroscopy, which provides site-specific information about molecular interactions involving flexible intrinsically disordered proteins. We show that the N-terminal IDR recognizes higher-order structures in target RNAs, particularly the GQ structure, via arginine-rich segments. This preferential binding involves not only electrostatic interactions but also π-interactions between the arginine side chains and the GQ structure. Along with biochemical RNA unwinding assays and bioinformatic analyses of the GQ propensity in DDX3X target RNA sequences, we propose that the GQ structure in the 5′-UTR serves as a marker that determines the selectivity of DDX3X binding. Our results provide the structural basis for translational regulation by DDX3X and underscore the remarkable role of IDRs in specifically recognizing highly ordered RNA structural elements and in controlling the cellular translational landscape.

## Results

### The influence of N- and C-terminal IDRs on DDX3X helicase function

We first investigated the contribution of the N- and C-terminal IDRs (hereafter referred to as N-IDR and C-IDR, respectively) to the helicase activity of DDX3X toward a model 36mer/18mer double-stranded RNA (dsRNA) substrate. In this dsRNA substrate, the 18mer strand (5′-CCCAAGAACCCAAGGAAC-3′), labeled with a fluorescent probe at its 5′-terminus, is annealed to the 36mer strand (5′-ACCAGCUUU-GUUCCUUGGGUUCUUGGGAGCAGCAGG-3′, with the underlined region complementary to the labeled 18mer), enabling us to directly monitor the displacement of the labeled 18mer strand by native polyacrylamide gel electrophoresis (native PAGE) as a result of DDX3X's helicase activity[38]. Since DDX3X proteins containing the N- and/or C-IDRs tend to aggregate at high concentrations, we measured the helicase activity of DDX3X with an N-terminal maltose-binding protein (MBP) tag added to enhance solubility. We compared the helicase activity of constructs with or without the IDRs: MBP-full length (FL) (residues 1–662), MBP-N-Core (residues 1–607), MBP-Core-C (residues 132–662), and MBP-Core (residues 132–607) (Fig. 1b). MBP-Core consists of the minimum functional region of DDX3X[26]. The effect of the N-terminal MBP tag was tested by comparing the activity of Core (residues 132–607) without the N-terminal MBP tag as a control. Protein purity was assessed by sodium dodecyl sulfate-polyacrylamide gel electrophoresis (Supplementary Fig. 2a, b). We also performed analytical size exclusion chromatography and confirmed that the purified proteins were predominantly monomeric. This allows for a comparison of activity differences attributable to the presence or absence of the N- and C-IDRs in the monomeric state, without the need to account for potential effects on oligomerization (Supplementary Fig. 2c).

The 36mer/18mer dsRNA was incubated with DDX3X proteins for 30 min at 37 °C, and then the 36mer/18mer dsRNA and the 18mer ssRNA displaced from the dsRNA were separated by electrophoresis on a native acrylamide gel. The activity of each DDX3X construct was compared based on the protein concentration-dependent increase in the ssRNA fraction (Fig. 1b, c). The assays were conducted with 50 mM added salt according to the previous studies, where low salt conditions were preferred (no added salt to 50 mM) to highlight the helicase activity[27,37–39]. Relative activity was assessed by comparing the apparent half-maximal concentration, $K_{1/2}$, at which the ssRNA fraction reaches 50%, assuming a sigmoidal relationship. The $K_{1/2}$ values for each protein construct were 33 nM for MBP-FL, 47 nM for MBP-N-Core, 190 nM MBP-Core-C, 1700 nM for MBP-Core, and 970 nM for Core. Consistent with the previous studies[26,37], truncation of the N- and/or C-IDR attenuated the helicase activity toward the 36mer/18mer dsRNA, with a 1.4-fold reduction by truncating the C-IDR, a sixfold reduction by truncating the N-IDR, and a 53-fold reduction by truncating both the N- and C-IDRs, as determined by the ratio of $K_{1/2}$ values. The apparent $K_{1/2}$ value of ~30 nM for MBP-FL was consistent with the previous report[26]. We note that the addition of the N-terminal MBP tag led to a slight decrease in activity (~1.8-fold larger apparent $K_{1/2}$ in MBP-Core compared to Core), likely reflecting the tag's potential influence on ligand binding. While we acknowledge the effect of the MBP tag, the impact of the N- and C-IDRs can be reliably assessed by comparing among the MBP-tagged proteins. These results indicate that the N- and C-IDRs play regulatory roles in enhancing the helicase activity of DDX3X toward dsRNA, presumably by promoting the binding of dsRNA via these regions. Hereafter, we focus on the molecular interaction involving the N-IDR, as it had the greater effect on helicase activity.

### Characterization of the interaction between the N-IDR and structured RNAs

RNA sequencing analyses of DDX3X targets so far have not identified a clear consensus sequence for DDX3X binding[7,9,10,21], suggesting that

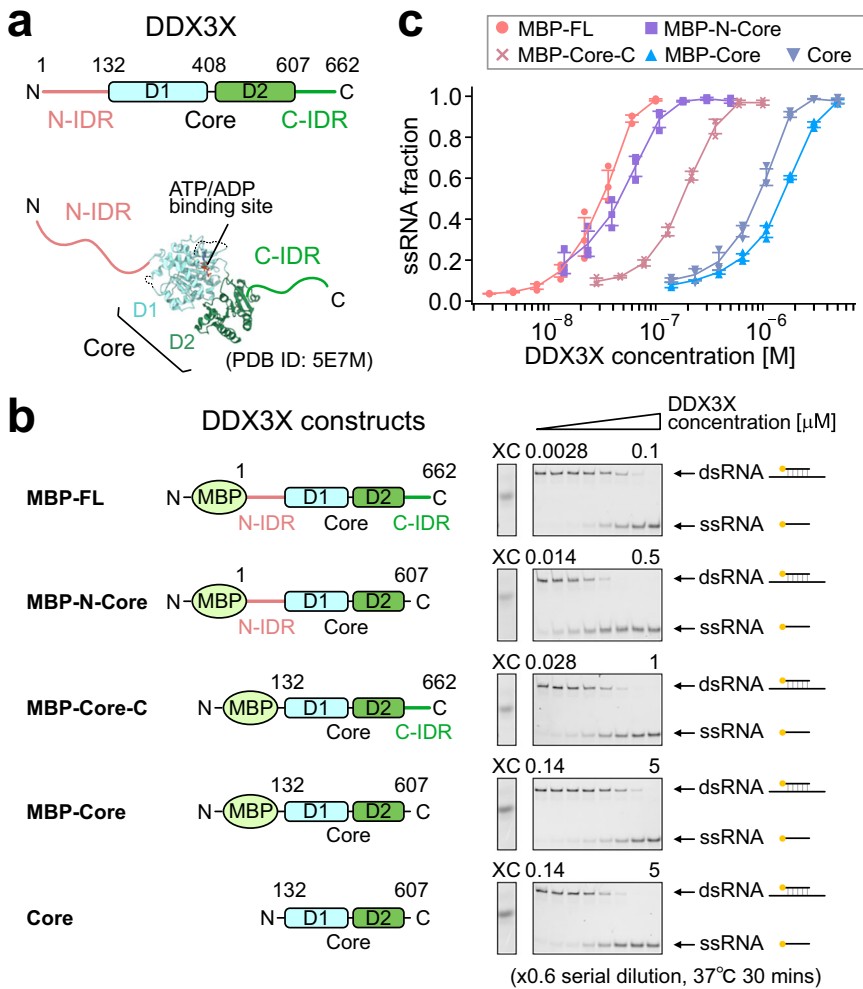

**Fig. 1 | Structure of DDX3X and its helicase activity toward dsRNA. a** The domain architecture of DDX3X. The model structure of full-length DDX3X is shown below (PDB 5E7M)[26]. **b** dsRNA unwinding assays for IDR-truncated DDX3X variants. The domain architecture of each construct is shown on the left, and representative results of the dsRNA unwinding assays are shown on the right. In the gels, the upper band corresponds to the 18mer/36mer dsRNA, while the lower band corresponds to the 18mer ssRNA displaced from dsRNA by the unwinding activity of DDX3X. Protein concentrations were varied with a 0.6-fold serial dilution, with the maximum and minimum concentrations indicated at the top of each gel. The band of the Xylene Cyanol FF (XC) was included on the left as a standard. The reaction mixture was incubated for 30 min at 37 °C. **c** Plots of ssRNA fractions as a function of DDX3X concentration [M] after 30 min of incubation, using MBP-FL (circle), MBP-N-Core (square), MBP-Core-C (cross), MBP-Core (triangle), and Core (inverted triangle) proteins. Error bars represent the standard deviation of three independent measurements, with center indicating mean values. Source data are provided as a Source Data file.

DDX3X recognizes higher-order structural elements rather than a specific sequence motif within the target RNAs. To test this hypothesis, we characterized the molecular interactions between the N-IDR and various structured RNAs using solution NMR, which provides atomic-level insights into interactions between the flexible IDR and RNA (Fig. 2a).

While the N-IDR was predicted to be intrinsically disordered by previous studies[21,28], we first experimentally confirmed this by NMR. The amide $^{15}$N-$^{1}$H heteronuclear single quantum coherence (HSQC) spectrum of the N-IDR (residues 1–132) showed limited dispersion in the $^{1}$H dimension, a hallmark of intrinsically disordered proteins (Fig. 2b, Supplementary Fig. 3). The secondary structure population was calculated for each residue based on the $^{1}$Hα, amide $^{1}$H$^{N}$, amide $^{15}$N, $^{13}$Cα, $^{13}$Cβ, and $^{13}$CO chemical shifts using the δ2D program[40]. The average populations of helix and β-sheet were 3.3% and 4.3%, respectively, indicating that the N-IDR does not form a well-defined secondary structure (Fig. 2c).

We then analyzed the interaction between the N-IDR and various structured RNA elements with distinct structural features. Typically, the higher-order structure of mRNA is described at the secondary structure level, where the energy of stability is assessed by the degree of base pairing and the nature of the loops between paired regions[41]. As representative examples of unpaired, base-paired, and loop regions, we used model substrates consisting of ssRNA, dsRNA, and hairpin RNA. For the loop structure, we used a tetraloop formed by antiparallel base-paired strands connected by a 4-nucleotides linker. In addition to these basic elements, we included GQ in our analyses, considering that GQ was previously identified in physiological substrates of DDX3X[36]. GQ is a characteristic structure typically formed by RNA containing repeats of two or three consecutive guanosines connected by linkers. The core of GQ consists of stacked guanine tetrads (G-tetrads), where each tetrad is formed by four guanines held together by Hoogsteen hydrogen bonds, and the structure is further stabilized by chelating monovalent metal ions such as potassium[42]. As RNA substrates, we used 10mer poly-uridine (poly-U$_{10}$) ssRNA (5'-UUUUUUUUUU-3'), GC-14mer self-complementary dsRNA (5'-GGGCGGGCCCGCCC-3'), 14mer UUCG tetraloop (5'-GGCACUUCGGUGCC-3'), which forms a stable hairpin structure[43,44], 12mer telomeric GQ RNA (5'-UAGGGUUAGGGU-3'), which forms a dimeric propeller-type parallel-stranded GQ structure[45,46], and 24mer NRAS GQ RNA (5'-UGUGGGAGGGGCGG

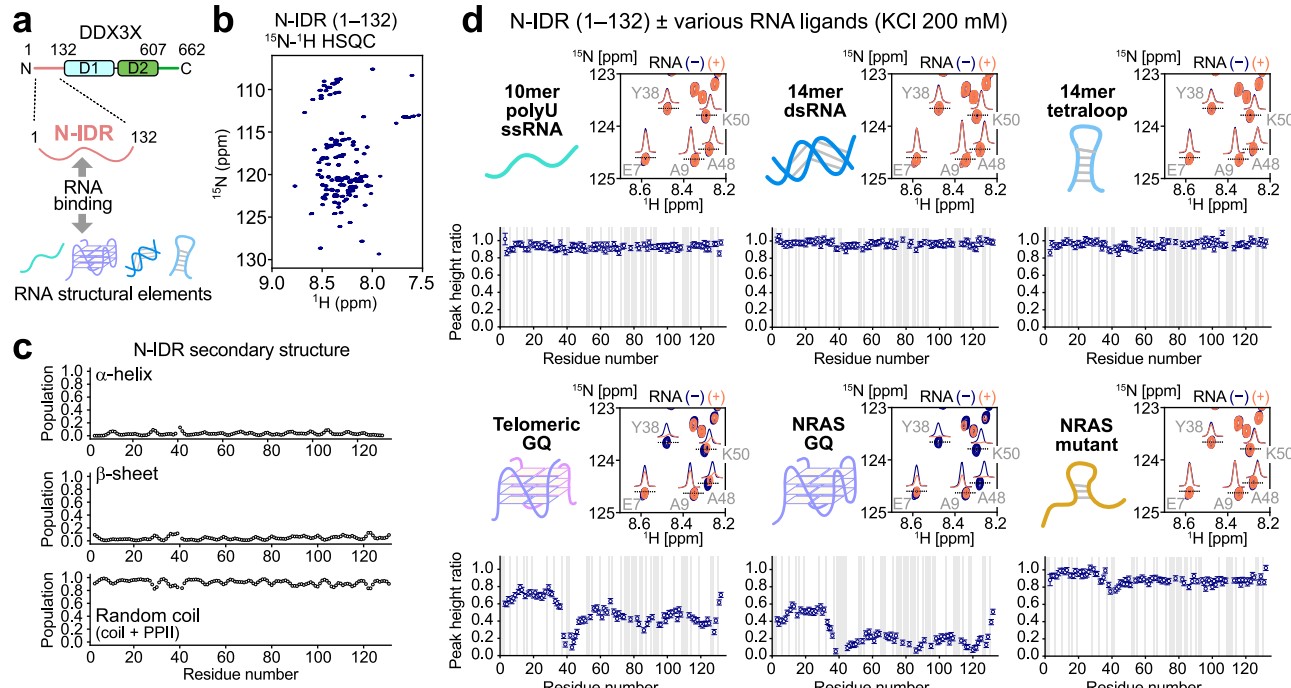

**Fig. 2 | NMR characterization of the interaction between the N-IDR and various RNA molecules. a** Schematic illustration of the NMR binding experiments between the N-IDR and various structured RNA molecules. **b** $^{15}$N-$^1$H HSQC spectrum of the [U-$^{15}$N]-labeled N-IDR in the absence of RNA. **c** Plots of the secondary structure populations of the N-IDR obtained using the δ2D method[40]. **d** NMR spectra and peak height ratios of the N-IDR signals obtained with and without each RNA molecule. For each sub-panel, the overlay of $^{15}$N-$^1$H HSQC spectra of the [U-$^{15}$N]-labeled N-IDR in the absence (navy) and presence (orange-red) of RNA, and the plot of peak height ratios are shown. The ratio was calculated by dividing the peak

height in the presence of RNA by that in the absence of RNA. Residues that were not analyzed are indicated by gray backgrounds. Error bars are calculated using the signal-to-noise ratios. All NMR measurements were performed at 10 °C and 1 GHz in a buffer containing 20 mM potassium phosphate (pH 7.0), 200 mM KCl, 5 mM DTT, 260 units/mL RNAsin® Plus RNAase inhibitor, and 5% D$_2$O. The protein concentration was 50 μM, and the RNA concentration was 50 μM (for poly-U10 ssRNA, GC-14mer dsRNA, 14mer tetraloop RNA, and telomeric GQ RNA) or 25 μM (for NRAS GQ RNA and NRAS mutant RNA). The 1D slices of the labeled signals are shown in each spectrum. Source data are provided as a Source Data file.

GUCUGGGUGC-3′) that was identified as a physiological binder of DDX3X[36,47,48] (Supplementary Fig. 4). Although physiological mRNA structures might be more complex due to the presence of other structural elements and tertiary contacts, comparing these RNA structural elements would provide insights into how the base pairing and chain flexibility influence the binding specificity of the N-IDR.

Under the low-salt conditions (50 mM KCl) used in the helicase assay in the previous section, a reduction in the intensity of the N-IDR NMR signals was observed upon the addition of 0.5 (for NRAS GQ) or 1 (for others) equimolar RNA (Supplementary Fig. 5). This shows that the N-IDR can bind to any RNA in a rather non-specific manner, presumably through electrostatic interactions between the positively charged N-IDR and negatively charged RNA molecules. However, the degree of intensity reduction varied significantly between RNA molecules, indicating a clear structural preference of the N-IDR for certain RNA secondary structures.

To emphasize this binding preference, we analyzed the interactions in the presence of 200 mM KCl to reduce the contributions of non-specific electrostatic interactions (Fig. 2d). Note that we used an ionic strength closer to, but slightly higher than, the physiological range (~110–130 mM)[49] to highlight the differences in binding. Notably, substantial intensity reductions of the N-IDR signals were observed in the presence of telomeric GQ and NRAS GQ RNAs, while such changes were not as pronounced with poly-U$_{10}$ ssRNA, GC-14mer dsRNA, and 14mer tetraloop RNA. As a control, we also analyzed the interaction with an NRAS mutant (5′-UGUA**G**AAA**G**AGC**G**GA**U**CUA**G**AUGC-3′, where base substitutions were introduced at the underlined G-tetrad positions). In the NRAS mutant, GQ formation is abolished by multiple

guanosine-to-adenosine substitutions to G-tetrads[36]. The intensity reduction was largely suppressed in the NRAS mutant, further validating that the N-IDR specifically recognizes the GQ structure. We note that very similar structural preferences were observed in a set of NMR experiments conducted under more physiologically relevant salt conditions (120 mM KCl) (Supplementary Fig. 6). These results demonstrate that the N-IDR preferentially recognizes and binds to the GQ structure of RNA, while it does not strongly interact with ssRNA, dsRNA, or hairpin structures. We note that, although the physiological substrate of the N-IDR is expected to be RNA, very similar binding was observed with the DNA counterpart, suggesting that the N-IDR does not specifically recognize RNA (*i.e.*, the ribose 2′-OH moiety) (Supplementary Fig. 7).

As an orthogonal test, we conducted electrophoretic mobility shift assays (EMSA)[50] using fluorescently labeled RNAs to verify the binding preference for GQ RNAs and to obtain rough estimates of the apparent dissociation constant ($K_d$) values (Supplementary Fig. 8). Consistent with the NMR results, little to no band-shift was observed for poly-U$_{12}$ ssRNA, GC-14mer dsRNA, and 14mer tetraloop RNA, whereas clear binding was observed for GQ RNAs. The apparent affinities for telomeric and NRAS GQ RNAs were estimated to be $1.0 \pm 0.2$ μM and $54 \pm 12$ nM, respectively, at 125 mM KCl. Although binding to the NRAS mutant RNA was also observed, its apparent $K_d$ ($6.0 \pm 0.8$ μM) was about two orders of magnitude weaker than that of the NRAS GQ RNA, further confirming the GQ-specific recognition by the N-IDR. Additionally, we performed isothermal titration calorimetry (ITC) to analyze the binding of the N-IDR to these GQ RNA molecules (Supplementary Fig. 9). In the presence of 50 mM KCl, the apparent $K_d$ values were calculated to be 3.4 μM for telomeric GQ and 1.4 μM for

NRAS GQ, based on fitting to the standard $n$-site binding model, confirming the direct binding of N-IDR to GQ RNAs. We note that, whereas the apparent $K_d$ values for telomeric GQ were comparable between EMSA and ITC, those for NRAS GQ differed significantly. We believe this discrepancy reflects the complex nature of the binding interaction, in which multiple N-IDR molecules may be involved in binding a single GQ RNA molecule, supported by Hill coefficients deviating from one in EMSA and the number of binding sites, $N$, being less than one in ITC analyses. In addition, ITC is sensitive to heat changes arising from potential self-association of either the N-IDR or GQ RNA, further complicating interpretation. Indeed, apparent discrepancies between ITC and other methods when assuming an oversimplified binding model have recently been reported in other systems involving IDRs[51,52]. Given the inherent complexity of the interaction between the N-IDR and GQ RNA, the apparent $K_d$ values are highly dependent on the experimental method used, as well as on whether the N-IDR or GQ RNA is used as the titrant. Therefore, we do not attempt to derive a unified binding model that can comprehensively explain the NMR, ITC, and EMSA results, as this would be beyond the scope of the present study.

### N-IDR-RNA interaction in the full-length context

Before investigating the molecular interaction between the N-IDR and GQ RNAs in detail, we confirmed that the interaction observed with the isolated N-IDR is retained in the full-length protein context. To assess the binding of the N-IDR to GQ RNA when tethered to the folded helicase core, we prepared [U-$^{15}$N] N-Core-MBP, in which an MBP tag was fused to the C-terminus of DDX3X residues 1–607 (N-IDR + Core) to maintain solubility while minimizing structural interference with N-IDR-RNA interactions. In the $^{15}$N-$^1$H HSQC spectrum of [U-$^{15}$N] N-Core-MBP, signals from the N-IDR region could be observed due to its inherent structural flexibility, while signals from the helicase core and the MBP-tag were broadened beyond detection (Supplementary Fig. 10a). Notably, the chemical shifts of the observed N-IDR signals were consistent with those in the isolated N-IDR, except for residues in the middle to C-terminal portion (after residue 34), whose signals were severely broadened, similar to those from the core and MBP regions. This suggests that the middle-to-C-terminal portion of the N-IDR forms some interactions with the helicase core, restricting its structural flexibility. Similar interactions have recently been proposed for the yeast counterpart Ded1p as well[53]. Although it remains challenging to characterize these potential N-IDR-core interactions, they are likely weak and transient in nature. This is supported by a clear difference in elution volume (~0.5 mL) in analytical size exclusion chromatography between DDX3X constructs with and without the N-IDR, indicating a substantial difference in hydrodynamic radius, which would be expected when the N-IDR retains a flexible conformation rather than being tightly associated with the core (Supplementary Fig. 2c).

Although only a limited number of NMR probes from the N-IDR were available due to line broadening from these potential N-IDR-core interactions, we were still able to monitor the N-IDR signals in [U-$^{15}$N] N-Core-MBP to evaluate binding to GQ RNA. As in the isolated N-IDR, signal intensity reductions were consistently observed upon the addition of NRAS GQ RNA, indicating that the interaction persists when the N-IDR is part of the larger protein construct, while retaining its disordered nature upon binding (Supplementary Fig. 10b). Notably, these reductions in intensity were less pronounced with the NRAS mutant RNA, suggesting that the structural selectivity is preserved even in the full-length background (Supplementary Fig. 10c).

To further validate the binding preference, we performed EMSA experiments using MBP-FL as an orthogonal test. Consistent with results from the isolated N-IDR, MBP-FL exhibited high-affinity binding to telomeric GQ RNA ($K_d = 54 \pm 14$ nM) and NRAS GQ RNA ($K_d = 43 \pm 4$ nM), while showing little to no band-shift for poly-U$_{12}$ ssRNA or 14mer tetraloop RNA, and significantly weaker binding to GC-14mer dsRNA ($K_d = 655 \pm 31$ nM) (Supplementary

Fig. 11a). The apparent affinity for NRAS mutant RNA ($K_d = 222 \pm 9$ nM) was about fivefold weaker than that for NRAS GQ RNA, supporting a structural preference for the GQ moiety in the full-length construct. Notably, the N-IDR-truncated construct, MBP-Core-C, exhibited significantly reduced affinity for both telomeric GQ RNA ($K_d = 579 \pm 69$ nM) and NRAS GQ RNA ($K_d = 201 \pm 5$ nM), strongly suggesting that the N-IDR plays a major role in GQ-specific binding (Supplementary Fig. 11b). Collectively, these NMR and EMSA results provide strong evidence that DDX3X preferentially recognizes GQ RNA, and that this selectivity is primarily mediated through N-IDR-RNA interactions.

### Structural basis for the interaction between the N-IDR and GQ RNA

We next sought to investigate the detailed molecular interaction between the isolated N-IDR and GQ RNA to gain structural insights into the observed specificity. The titration series of telomeric GQ and NRAS GQ RNA showed that the chemical shift changes upon binding to GQ RNA were small in both the $^1$H (<0.04 ppm) and $^{15}$N (<0.12 ppm) dimensions, and the effect of binding was mainly observed as an intensity reduction in a subset of residues (Fig. 3a, Supplementary Fig. 12a). Such small chemical shift changes have been reported in other IDR-nucleic acid interactions[54,55], likely reflecting the modest change in the local chemical environment due to the absence of drastic rearrangement in secondary or tertiary structures around the spin probe or hydrogen bond network. We note that, although the absolute chemical shift change values are small, they still provide site-specific information about the interaction consistent with that obtained from intensity changes (Supplementary Fig. 13).

To gain further insight into the interaction, we investigated the origin of the signal intensity reduction observed upon the addition of GQ RNA. The simplest explanation is that the free and bound states are exchanging on the microsecond-to-millisecond timescale, resulting in exchange broadening[56]. This is a common feature of relatively weak binding interactions with $K_d$ values in the micromolar range, as is the case here. To test this, we performed $^{15}$N and $^1$H Carr–Purcell–Meiboom–Gill (CPMG) relaxation dispersion experiments[57,58] to extract such potential exchange-induced broadening ($R_{ex}$) contributions in the presence of GQ RNA (Supplementary Fig. 14a, b). Intriguingly, the $R_{ex}$ values measured in the presence of 0.5 equimolar telomeric GQ RNA or 0.2 equimolar NRAS GQ RNA were generally smaller than 5 s$^{-1}$, suggesting that the exchange broadening between free and bound states is not the major cause of the observed intensity reductions. Instead, we observed an overall increase in the apparent $^{15}$N and $^1$H $R_2$ rates measured at a 1 kHz CPMG field (1.3-fold and 1.2-fold increases for $^{15}$N and $^1$H, respectively, in the presence of telomeric GQ RNA; and 1.5-fold and 1.3-fold increases, respectively, for NRAS GQ RNA, averaged over all residues). These increases in apparent $R_2$ rates are broadly consistent with the observed signal intensity reductions. For example, the average decrease in signal height in the presence of 0.5 equimolar telomeric GQ RNA is expected to be ~0.64 (=1/1.3 × 1/1.2), considering the broadening in both the $^{15}$N and $^1$H dimensions. This value closely matches the observed average decrease of ~0.67 (Fig. 3c), suggesting that the signal reduction is unlikely to result from irreversible aggregation or precipitation of the complex that escape detection in the NMR sample tube. Given that exchange between free and bound states occurs in the fast exchange regime, the increase in $R_2$ rates likely reflects elevated apparent $R_2$ in the GQ RNA-bound state, which may be attributed to heterogeneity within the bound ensemble, restricted picosecond-to-nanosecond motions of the amide $^1$H-$^{15}$N bond vector upon binding, and/or an increase in the apparent molecular weight due to complex formation involving multiple N-IDR and GQ RNA molecules. In what follows, we focus on signal intensity reduction as a hallmark of GQ binding owing to its high intrinsic sensitivity.

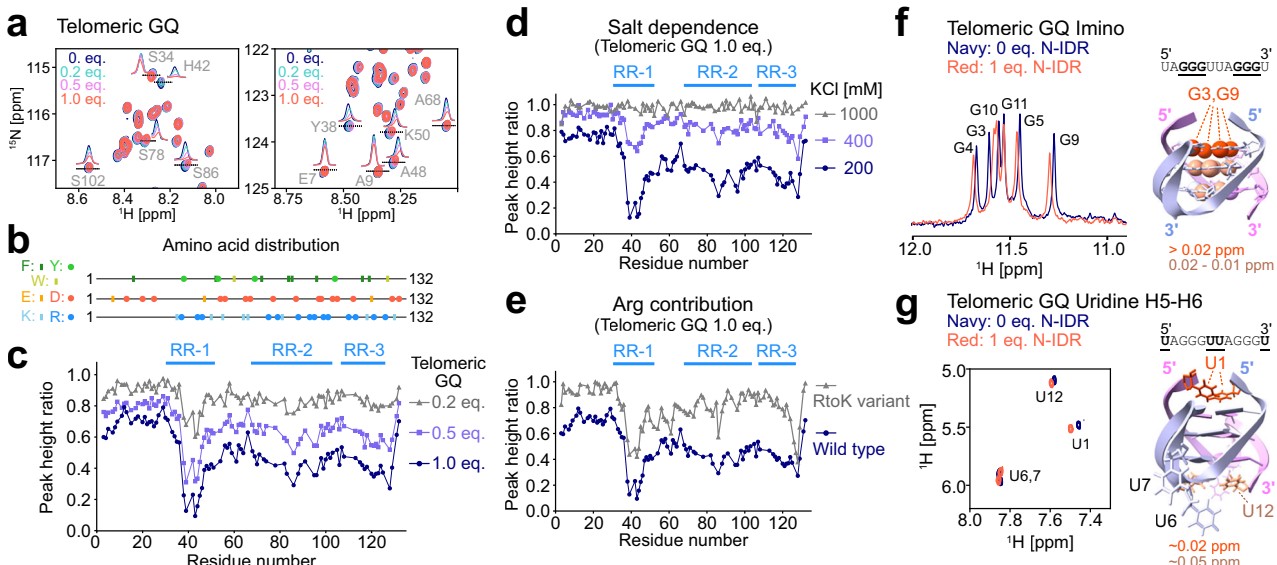

**Fig. 3 | Telomeric GQ RNA titration experiments. a** Close-up views of $^{15}$N-$^{1}$H HSQC spectra of the [U-$^{15}$N]-labeled N-IDR in the presence of varying concentrations of telomeric GQ RNA (navy: 0 equimolar, turquoise: 0.2 equimolar, pink: 0.5 equimolar, and red: 1 equimolar). The KCl concentration was 200 mM. The 1D slices of the labeled signals are shown in each spectrum. **b** Amino acid distributions of aromatic (top, green and khaki), negatively charged (middle, red), and positively charged (bottom, blue) residues. The x-axis scale for the residue number is consistent with panel (**c**) below. **c** Plots of the peak height ratios of the N-IDR signals obtained with 0.2 (gray), 0.5 (purple), or 1.0 (navy) equimolar telomeric GQ RNA. **d** Plots of the peak height ratios of the N-IDR signals obtained with 200 mM (navy), 400 mM (purple), or 1 M (gray) KCl. **e** Plots of the peak height ratios of the wild-type (navy) and RtoK variant (gray) N-IDR signals obtained with 200 mM KCl. In panels

**d**, and **e**, the ratio was calculated by dividing the peak height in the presence of 1 equimolar (50 µM) telomeric GQ RNA by that in the absence of RNA. Overlays of the imino $^{1}$H 1D (**f**) and uridine H5-H6 $^{1}$H-$^{1}$H TOCSY (**g**) NMR spectra of unlabeled telomeric GQ RNA recorded with and without the 1 equimolar unlabeled N-IDR. Residues with marked chemical shift changes are mapped onto the structure (PDB 2KBP)[45]. The sequence of the telomeric GQ RNA is shown above the structure. For panels (**a**), (**c**), (**d**), and (**e**), NMR measurements were performed at 10 °C and 1 GHz, with a protein concentration of 50 µM. Three arginine-rich (RR) regions are highlighted with blue lines in panels (**c**), (**d**), and (**e**). For panels (**f**) and (**g**), NMR measurements were performed at 25 °C and 1 GHz, with an RNA concentration of 50 µM. Source data are provided as a Source Data file.

The intensity profile at each GQ RNA concentration showed at least three binding sites in the N-IDR, residues around 40, 90, and 120 (Fig. 3c, Supplementary Fig. 12b). This multivalent binding mode likely contributes to the inhomogeneity of the bound state, as described above. To investigate the molecular interactions responsible for binding, we examined the amino acid composition of the binding sites, focusing on charged and aromatic residues (Fig. 3b). The binding sites for the GQ RNAs were highly enriched in positively charged arginine residues and aromatic residues, such as phenylalanine and tyrosine, but contained relatively few lysine residues. This suggests that binding is not solely mediated by electrostatic interactions; rather, π-interactions involving the side chains of arginine and aromatic residues also play significant roles (these arginine-rich, RR, regions are highlighted with blue lines in Fig. 3, Supplementary Figs. 12 and 13). To validate these contributions, we first conducted NMR binding experiments under varying KCl concentrations to probe the electrostatic contributions and then performed the binding experiments with several N-IDR variants to identify key interactions.

The salt concentration dependence of electrostatic interactions between negatively charged RNA and positively charged peptides has been extensively characterized. For example, the interaction between a 19-residue box B RNA hairpin and a 22-residue arginine-rich peptide from the N protein of phage λ (containing 5 arginine and 2 lysine residues, theoretical pI of 11.44) exhibited strong dependence on KCl concentration. This system, comparable in length and size to the interaction between the GQ RNA and the arginine-rich segments of the N-IDR studied here, showed that the electrostatic contribution to the binding free energy ($\Delta G_{dock}$) varied significantly with KCl concentration, with $\partial(\Delta G_{dock})/\partial\log[\text{KCl}]$ of ~6 kcal/mol[59]. This corresponds to a ~24-fold decrease in binding affinity when the KCl concentration is doubled at 10 °C. Assuming a $K_d$ range of 1–10 µM as estimated from

the EMSA and ITC experiments with telomeric GQ RNA, this ~24-fold increase in $K_d$ (i.e. a shifted $K_d$ of 24–240 µM) results in a change in bound fraction from ~64–87% to ~15–51%, assuming 50 µM concentrations for both protein and RNA and a simple one-site binding model. Based on this, we monitored GQ RNA binding at 400 mM KCl and compared the results with those at 200 mM KCl (Fig. 3d, Supplementary Fig. 12c). At 400 mM KCl, intensity reductions were ~40–60% less pronounced than those observed at 200 mM KCl. Although it is difficult to estimate the exact bound fraction from intensity reductions due to the heterogeneity of the bound state, this result aligns reasonably well with the expected decrease in binding as noted above. As expected from the sharp decrease in binding affinity with increasing salt concentration, the binding was nearly completely abolished by further increasing the KCl concentration to 1000 mM. Collectively, these results demonstrate that electrostatic interactions play a critical role in determining the binding free energy of the N-IDR-GQ RNA interaction.

Despite the dominant role of electrostatic interactions in the N-IDR-RNA interaction, the N-IDR exhibited a clear structural preference depending on the secondary structure of the ligand RNA, suggesting that non-electrostatic interactions, such as π-interactions involving arginine side chains[60], also play significant roles in binding and determining substrate specificity. To test this, we monitored the interactions with GQ RNA by using the RtoK variant, in which all arginine residues were replaced with lysine, for probing the contributions of π-interactions. Notably, binding was largely suppressed when using the RtoK variant (Fig. 3e, Supplementary Fig. 12d). This demonstrates that π-interactions involving arginine side chains, as well as electrostatic interactions, are critical factors that define the strength and specificity of binding to the GQ structure. The direct involvement of aromatic side chains was further tested through NMR binding

experiments using the FYtoA variant (Y38A, F96A, F116A, and F119A), in which tyrosine or phenylalanine residues in each RR segment were replaced with alanine. In this variant, the intensity reductions upon the addition of telomeric or NRAS GQ RNAs were less pronounced, particularly for residues close to the mutated sites, further supporting the importance of π-interactions involving aromatic side chains (Supplementary Fig. 15).

To gain structural insights into the binding site on GQ RNA, we then turned to NMR observation of GQ RNA with and without the N-IDR. For this purpose, we used telomeric 12mer GQ RNA, which was shown to form a homogeneous GQ structure by both NMR and X-ray crystallography, and for which NMR signal assignments are available[45,46]. We conducted binding experiments observing the imino $^1$H signals of the guanine bases and the H5-H6 correlations of the uridine bases. Upon the addition of an equimolar amount of the N-IDR, chemical shift changes were observed for all imino $^1$H signals of the guanine bases forming the G-tetrad (Fig. 3f). In the 2D total correlation spectroscopy (TOCSY) spectrum observing the uridine H5-H6 correlations, chemical shift changes were observed for the H5-H6 signals of U1 and U12, whose bases are stacked on the G-tetrad, but not for those of U6 and U7, which are located in the propeller loop connecting the G-tetrads (Fig. 3g). These results suggest that the N-IDR specifically recognizes the G-tetrad core structure, presumably forming π-interactions through arginine and aromatic side chains.

### Binding of individual Arg-rich clusters to GQ RNA

As mentioned above, the intensity reduction profiles suggest that GQ RNA binds mainly to three RR clusters in the sequence, defined as RR-1, RR-2, and RR-3. To determine whether these RR clusters cooperatively bind to GQ RNA, we performed NMR binding experiments using three variants: RKtoA-1 (K35A, R37A, R44A, R46A, K50A), RKtoA-2 (R75A, R79A, K81A, R88A, R93A, R95A, R99A, R101A), and RKtoA-3 (R110A, R113A, K118A, R121A, R126A, K130A), where the positively charged arginine and lysine residues in each RR cluster were substituted with alanine to abolish GQ RNA binding (Supplementary Fig. 16). In all three RKtoA variants, intensity reduction in the mutated region was not observed with 1 equimolar telomeric GQ RNA, consistent with the above notion that electrostatic interactions and π-interactions are both important for binding. Interestingly, residues in the other RR clusters outside the mutated site exhibited intensity reductions comparable to those observed in the wild-type proteins. These results suggest that the structural coupling between the three RR clusters is relatively weak and that each RR cluster interacts with GQ RNA rather independently.

To further investigate the independent binding activity and individual contributions of each RR cluster to GQ RNA recognition, we employed an alternative approach by monitoring the interaction using isolated RR fragments. To this end, we prepared three RR fragment proteins, RR-1 (residues 31–61), RR-2 (residues 73–102), and RR-3 (residues 106–127), and performed NMR binding experiments with various RNA ligands (Supplementary Figs. 17, 18, and 19). Most of the signal assignments for each RR region could be readily transferred from those of the full N-IDR, indicating that the disordered nature of the N-IDR is preserved in each isolated RR construct. Notably, for all RR fragment proteins, significant intensity reductions and chemical shift changes were observed only with telomeric and NRAS GQ RNAs, while little to no spectral changes were seen for poly-U$_{10}$ ssRNA, GC-14mer dsRNA, and NRAS mutant RNA, demonstrating that each RR region independently exhibits GQ-specific binding. The intensity reduction profiles were overall similar among three RR regions, although minor differences were observed in their preferences between telomeric and NRAS GQ RNAs. For example, RR-2 showed comparable intensity reductions upon the addition of 1 equimolar telomeric GQ and 0.5 equimolar NRAS GQ (Supplementary Fig. 18), whereas RR-3 exhibited larger intensity reductions for

telomeric GQ than for NRAS GQ (Supplementary Fig. 19). These subtle differences may reflect distinct structural preferences or binding modes toward GQ RNAs with different topologies[61]; however, we did not observe clear functional differences among these three RR regions, at least in the context of the N-IDR's GQ-specific recognition. Direct binding of RR-1 and RR-2 to GQ RNAs was further confirmed by ITC analyses, which yielded apparent $K_d$ values ranging from 2 to 11 μM, comparable to those of the full N-IDR ($K_d$ ~ 1–3 μM) (Supplementary Fig. 20). These results further support the notion that there is no strong structural coupling or binding cooperativity among the three RR clusters. In the NMR binding experiment with telomeric GQ RNA, we observed slightly larger intensity reductions around residue 40 of the N-IDR (Fig. 3c). However, we did not detect stronger affinity for the RR-1 fragment protein that contains this region. This suggests that the observed larger intensity reductions are more likely due to restricted motion involving the characteristic successive proline residues (Pro40-Pro41) in the bound state, rather than the preferential binding of GQ RNA to this region.

### Binding of the N-IDR to GQ segments in physiological mRNA substrates

While the NMR experiments so far demonstrated the preferential binding of the N-IDR to GQ RNA substrates, we were interested in whether this preference is relevant in the binding to physiological substrates. To address this, we characterized the interaction between the N-IDR and the physiological substrates of DDX3X. As physiological substrates, we chose the 5′-UTR of the small guanine-nucleotide binding protein *RAC1* gene[7,12–17] (197 nt) and the ornithine decarboxylase *ODC1* gene[5,7,12,62] (511 nt), both of which are translationally regulated by DDX3X in vitro and in cells[7,12].

First, we analyzed the 5′-UTR sequences of these transcripts to identify putative GQ-forming sequences. GQ propensity was predicted using the detection algorithm implemented in the rG4detector software[63], which evaluates sequence patterns of consecutive guanosines and assigns a prediction score, an indicator of GQ stability, to each nucleotide. In the prediction score plots along each transcript sequence, local maxima were found in nucleotides 53–77 of *RAC1* and in nucleotides 150–174 and 279–303 of *ODC1* (Fig. 4a). These regions were consistently identified as putative GQ-forming sequences using other predictors, G4Hunter[64] and pqsfinder[65], indicating a high likelihood that these segments form GQ structures (Supplementary Fig. 21). We prepared fragments of these putative GQ-forming sequences and negative control sequences: nucleotides 129–153 of *RAC1* and 449–473 of *ODC1*, which showed local minima in the rG4detector predictions (Fig. 4a). The formation and absence of GQ structures in these RNA fragments were validated by $^1$H NMR. The $^1$H NMR signals in the imino region are known to sensitively reflect base-paired structures, reliably distinguishing between canonical Watson–Crick (WC) base pairs and Hoogsteen base pairs, the hallmark of GQ structures[66]. As a reference, the $^1$H NMR spectra of telomeric GQ, NRAS GQ, and GC-14mer dsRNA are shown (Fig. 4b), where imino signals were observed at ~11 ppm for the Hoogsteen base-paired GQ RNA and at ~12–14 ppm for the WC base-paired GC-14mer dsRNA. Note that the imino fingerprints of GQ RNA are frequently broad and heterogeneous as seen in NRAS GQ, reflecting conformational polymorphism of GQ[64,66,67]. The imino NMR spectra of the *RAC1* and *ODC1* fragments are summarized in Fig. 4c. Consistent with the predictions, imino signals were observed at ~11 ppm for the putative GQ-forming fragments (nucleotides 53–77 of *RAC1*, 150–174 and 279–303 of *ODC1*), while imino signals were mainly observed at ~12–14 ppm for the negative control fragments (nucleotides 129–153 of *RAC1* and 449–473 of *ODC1*). In the GQ-forming *ODC1* fragments, weak imino signals were observed in the WC region as well, indicating that these fragments are rather heterogeneous and contain a minor population of WC base-paired structures.

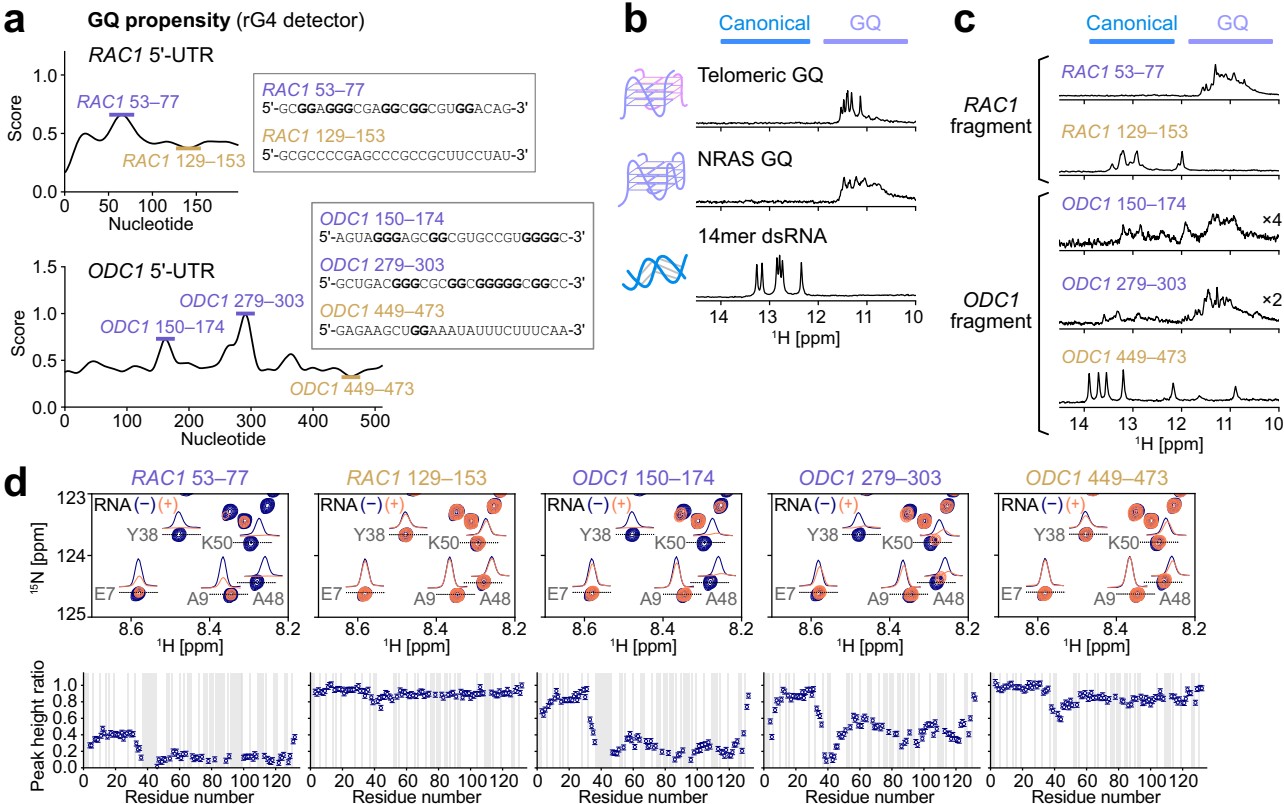

**Fig. 4 | GQ propensity of *RAC1* and *ODC1* 5'-UTR and its interaction with the N-IDR. a** Plots of the prediction scores for GQ propensity of *RAC1* (top) and *ODC1* (bottom) 5'-UTR sequences obtained using the rG4detector software[63]. In the insets, the sequences of the segments with high prediction scores (purple) and those of the negative controls (gold) are highlighted. **b** The imino ¹H 1D NMR spectra of telomeric GQ RNA (top), NRAS GQ RNA (middle), and GC-14mer dsRNA (bottom). Regions with signals from canonical base pairs (blue) and Hoogsteen base pairs in GQ (purple) are indicated above the spectra. **c** The imino ¹H 1D NMR spectra of *RAC1* and *ODC1* fragments are shown as in panel (**b**). For the *ODC1* 150–174 and *ODC1* 279–303 fragments, signal intensities were scaled by 4-fold and 2-fold, respectively, for visualization. **d** NMR spectra and peak height ratios of the N-IDR signals obtained with and without each RNA fragment in the presence of 200 mM KCl. In the top row, overlays of ¹⁵N-¹H HSQC spectra of the [U-¹⁵N]-labeled N-IDR in the absence (navy) and presence (orange-red) of RNA are shown. The 1D slices of the labeled signals are shown in each spectrum. In the bottom row, plots of peak height ratios are shown. The ratio was calculated by dividing the peak height in the presence of 1 equimolar RNA by that in the absence of RNA. Error bars were calculated using the signal-to-noise ratios. Residues that were not analyzed are indicated with gray backgrounds. All NMR measurements were performed at 10 °C and 1 GHz, with protein and RNA concentrations of 50 μM. Source data are provided as a Source Data file.

We then analyzed the interaction of the N-IDR with these 5'-UTR fragments of *RAC1* and *ODC1* by NMR. In the presence of equimolar amounts of the GQ-forming fragments (*RAC1* 53–77, *ODC1* 150–174, and *ODC1* 279–303), binding was observed as a marked intensity reduction of the N-IDR amide signals. In contrast, such intensity change was less prominent when the negative control fragment (*RAC1* 129–153 and *ODC1* 449–473) was added (Fig. 4d). Notably, the most significant intensity reduction was observed with nucleotides 53–77 of *RAC1*, where the most homogeneous GQ structure was formed, as indicated by the imino ¹H NMR spectrum (Fig. 4c). This suggests that the interaction strength qualitatively correlates with the stability and/ or abundance of the GQ structure. These results support the notion that the N-IDR selectively recognizes GQ segments in mRNA transcripts, which serves as an important factor defining DDX3X target specificity.

The binding of these mRNA fragments was also validated by EMSA (Supplementary Fig. 22), which showed stronger binding for the GQ-forming fragments, *RAC1* 53–77 ($K_d = 0.51 \pm 0.08\,\mu M$) and *ODC1* 150–174 ($K_d = 1.97 \pm 0.23\,\mu M$), compared to the non-GQ fragments ($K_d > 10\,\mu M$). One exception was *ODC1* 279–303, which exhibited binding in the NMR experiments but behaved similarly to the non-GQ fragments in EMSA. We speculate that this discrepancy most likely arises from the higher sensitivity of NMR for detecting relatively weak interactions with $K_d$ values in the micromolar range, whereas the faster

association-dissociation kinetics may have hindered the resolution of distinct protein-RNA complex in EMSA, especially given that the affinity difference between *ODC1* 279-303 and the non-GQ fragment *ODC1* 449-473 appears to be rather modest.

## Characterization of the GQ-unfolding activity of DDX3X

For DDX3X to effectively stimulate the translation of mRNA transcripts containing GQ structures in their 5'-UTRs, the helicase core of DDX3X needs to unwind the GQ structure following the binding of the N-IDR to the GQ segment. The unfolding of GQ structures has been characterized in the yeast counterpart of DDX3X, Ded1p[68], and in other human DDX family members, such as DDX1[69], DDX5[70], and DDX21[71], although the GQ-unfolding activity of DDX3X itself has not yet been demonstrated, to our knowledge. To test the GQ-unfolding activity of DDX3X, we turned to a fluorescence assay using dually labeled NRAS GQ, particularly focusing on the functional role of the N-IDR.

The GQ-unfolding activity was measured via fluorescence quenching of the 6-carboxyfluorescein (FAM) fluorophore at the 5'-end by the BHQ1 quencher at the 3'-end of the NRAS GQ sequence[70]. When the GQ structure forms, the 5'- and 3'-termini come into close proximity, effectively quenching the fluorescence of the FAM group. Unfolding of the GQ structure by DDX3X can be monitored as an increase in FAM fluorescence (Fig. 5a). As a positive control to confirm that FAM fluorescence is sensitive to structural reorganization of the

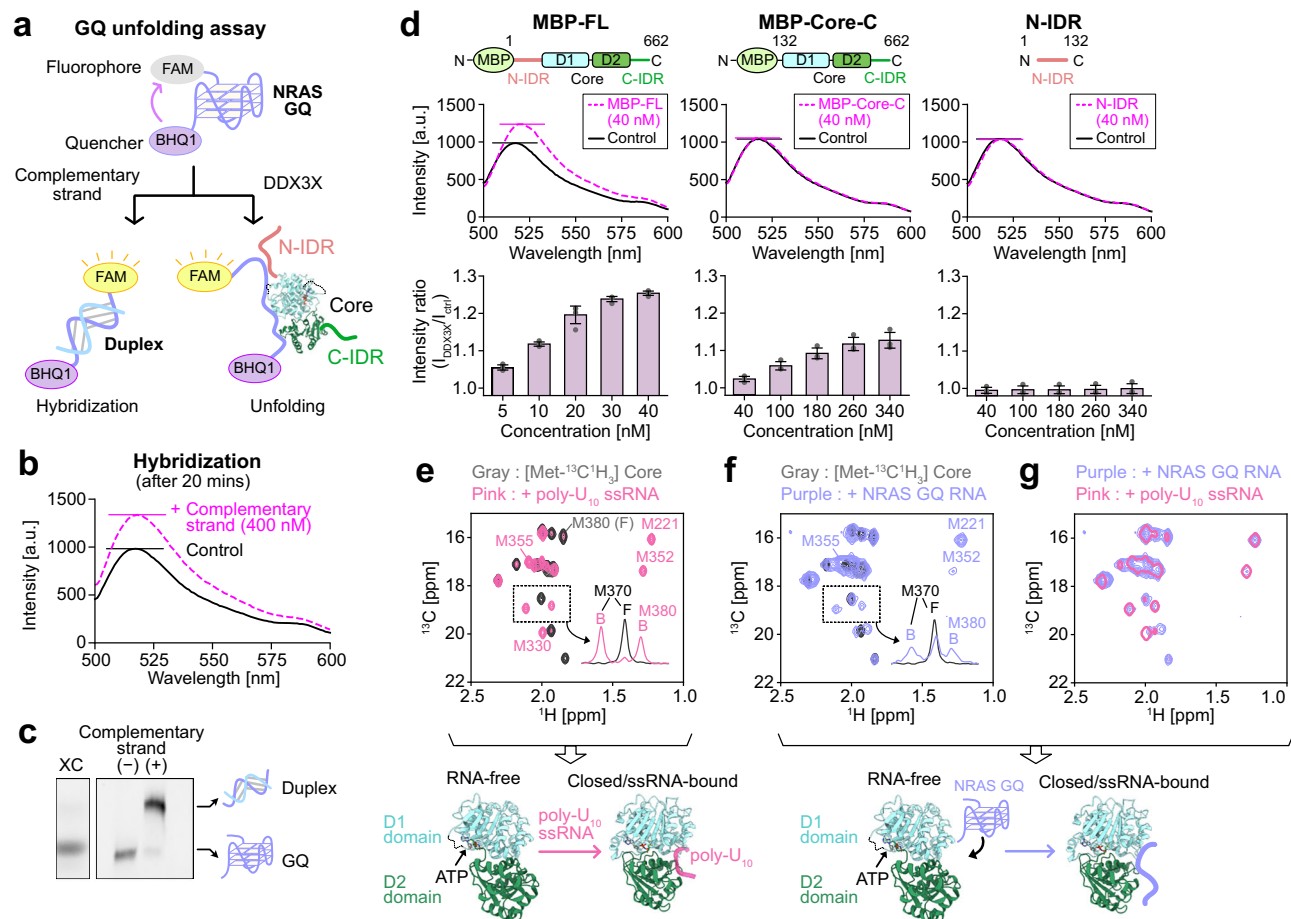

**Fig. 5 | GQ-unfolding activity of DDX3X. a** Schematic representation of the fluorescence-based NRAS GQ-unfolding assay. **b** Fluorescence emission spectra of FAM/BHQ1-labeled NRAS GQ recorded without (solid black line) and with (dotted magenta line) 400 nM complementary RNA after 20 min of incubation. The peak intensities are indicated by horizontal solid lines. **c** Native gel analyses of FAM/BHQ1-labeled NRAS GQ with and without the complementary RNA. The band of the Xylene Cyanol FF (XC) was included on the left as a standard. The analyses were independently repeated three times with consistent results. **d** Fluorescence emission spectra of FAM/BHQ1-labeled NRAS GQ recorded without (solid black line) and with (dotted magenta line) the MBP-FL (left), MBP-Core-C (middle), and N-IDR (right) proteins. The y-axis is labeled in arbitrary units (a.u.). The intensity ratios of the maximum fluorescence intensities of FAM/BHQ1-labeled NRAS GQ at each protein concentration are plotted below. Error bars represent the standard deviation of three (for MBP-Core-C and N-IDR) or four (for MBP-FL) independent measurements, with center indicating mean values. **e** Overlay of the spectra of [Metε-$^{13}$C$^1$H$_3$]-labeled E348Q DDX3X Core measured without (gray) and with 4 equimolar poly-U$_{10}$ ssRNA (pink). **f** Overlay of the spectra measured without (gray) and with 6 equimolar NRAS GQ (purple). **g** Overlay of the spectra measured with 6 equimolar NRAS GQ (purple, multiple contours) and with 4 equimolar poly-U$_{10}$ ssRNA (pink, single contour). Schematic cartoons describing the interactions are shown below the spectra (PDB 5E7M and 2DB3)[26,32]. In panels (**e**) and (**f**), the projections of the dotted region are shown in the inset. Free (F) and bound (B) signals are indicated for representative residues. All NMR measurements were performed at 35 °C and 1 GHz, with a protein concentration of 50 μM. Source data are provided as a Source Data file.

dually labeled NRAS GQ RNA, we measured fluorescence changes upon the addition of the complementary RNA strand[72]. In the presence of an excess amount of the complementary strand, an increase in FAM fluorescence was observed, reflecting the increased distance between the 5′- and 3′-termini upon duplex formation (Fig. 5b). The duplex formation was further confirmed by a band shift in the native PAGE analysis of NRAS GQ with and without the complementary strand (Fig. 5c).

We then measured the fluorescence of the modified NRAS GQ (20 nM) in the presence of varying concentrations of MBP-FL, MBP-Core-C, and N-IDR proteins (Fig. 5d). Upon the addition of MBP-FL, we observed an increase in fluorescence, with the signal nearly reaching a plateau at 2 equimolar concentrations (40 nM) of MBP-FL, suggesting that DDX3X can unfold the NRAS GQ structure. The apparent binding affinity of MBP-FL toward the dually labeled NRAS GQ was estimated to be on the order of 10 nM, consistent with the apparent $K_d$ for MBP-FL-NRAS GQ interaction ($K_d = 43 \pm 4$ nM) obtained from EMSA (Supplementary Fig. 11). In contrast, almost no increase in fluorescence was observed upon the addition of the same concentration of MBP-Core-C or N-IDR (Fig. 5d). However, at a much higher concentration (~340 nM), MBP-Core-C showed a smaller but significant increase in fluorescence, while the N-IDR alone produced little to no effect (Fig. 5d, Supplementary Fig. 23a). These results indicate that the Core-C region, rather than the N-IDR, is responsible for the GQ-unfolding activity, and that this activity is significantly enhanced in the presence of N-IDR. Combined with EMSA results showing that the N-IDR significantly increases binding affinity to GQ RNA substrates (Supplementary Fig. 11), these findings indicate that the robust GQ-unfolding activity of MBP-FL arises from the synergistic actions of the N-IDR-mediated binding and the helicase activity of the folded core. Time-course analysis showed that MBP-FL unfolded the NRAS GQ structure almost instantaneously, within the ~14-s dead time following MBP-FL addition (Supplementary Fig. 23b), which is consistent with previous results for DDX5, which rapidly unfolded GQ DNA with a half-time of a few seconds[70]. Supporting the results of the fluorescence quenching assay, the GQ-unfolding activity of MBP-FL was further confirmed by a GQ

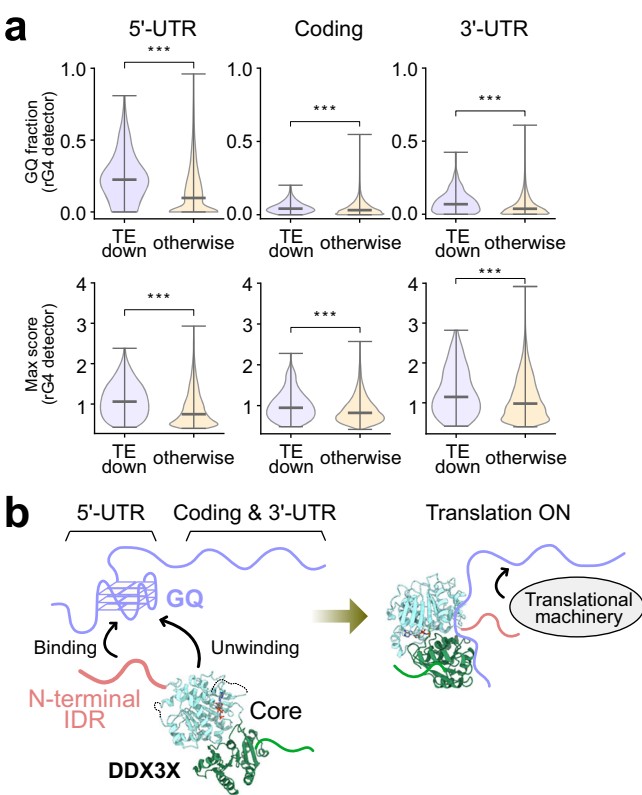

**Fig. 6 | GQ propensity of DDX3X's targets. a** Violin plots comparing the distributions of the predicted GQ fractions (top) and maximum GQ propensity scores (bottom) between DDX3X's targets (TE-down; 208 transcripts) and non-targets (otherwise; 8783 transcripts). Three regions, 5′-UTR (left), protein coding (center), and 3′-UTR (right), were analyzed separately. The horizontal bars in the violin plots indicate the median values, while the top and bottom error bars represent the maximum and minimum values, respectively. Statistical significance between the two groups was evaluated using the two-sided Mann-Whitney U test (***: $p < 0.001$). All relevant data and exact $p$ values are included in the Source Data file. **b** Cartoon representations of the interaction between an mRNA transcript containing the GQ structure and DDX3X.

hybridization assay using a complementary RNA strand[68] (Supplementary Fig. 24).

To further validate the GQ-unfolding activity of the folded core, we investigated the structural changes in the core region using NMR. In our previous study, we showed that the folded core in the ATP-bound state preferentially binds to the ssRNA form of structured RNA, adopting a well-defined domain-closed structure, which underlies its unwinding activity toward a diverse set of RNAs[34]. Based on this model, we hypothesized that the core region can unfold the GQ structure to form the ssRNA-bound closed conformation. To test this, we analyzed the $^{13}C$-$^{1}H$ heteronuclear multiple quantum coherence (HMQC) spectra of the [Metε-$^{13}C^{1}H_3$]-labeled core region (residues 132–607) in the ATP-bound state, both with and without GQ RNA. In these analyses, we used the core region without the N- and C-IDRs, due to the instability of the IDR-containing DDX3X proteins at the high concentrations required for NMR analyses. We also introduced the E348Q mutation in the conserved DEAD motif to stabilize the ATP-bound state by slowing down ATP hydrolysis, as described in our previous study[34]. As a reference, we measured the $^{13}C$-$^{1}H$ HMQC spectra of the [Metε-$^{13}C^{1}H_3$]-labeled core region with and without a model ssRNA substrate, poly-$U_{10}$. Methionine methyl signals from M221, M330, M352, M355, and M380 in the D1 domain showed significant chemical shift changes, reflecting poly-$U_{10}$ binding and the accompanying structural

rearrangement of the core region (Fig. 5e). We then measured the $^{13}C$-$^{1}H$ HMQC spectrum of the [Metε-$^{13}C^{1}H_3$]-labeled core region in the presence of an excess amount of NRAS GQ (300 μM) to promote binding (Fig. 5f). Upon addition of NRAS GQ, we observed a set of bound-state signals in a slow exchange manner, with an apparent bound population of ~30–50%, reflecting the inherently weak binding affinity of the core region for NRAS GQ without the N-IDR. Notably, comparison of the bound-state signals between poly-$U_{10}$ and NRAS GQ revealed that their chemical shifts were nearly identical, suggesting a close similarity in the bound conformation (Fig. 5g). In our previous study[34], we proposed that this bound state corresponds to the closed conformation, where an elongated ssRNA binds both the D1 and D2 domains in a bipartite manner. Given that NRAS GQ adopts a compact folded conformation in its GQ topology (Supplementary Fig. 4), it is reasonable to expect that the GQ structure must be, at least partially, unwound to participate in the bipartite interaction in an elongated form. Although direct NMR observation of the bound NRAS GQ RNA was not feasible due to structural heterogeneity, modest binding affinity, and the large molecular weight of the complex, the close similarity in the bound conformation between poly-$U_{10}$ and NRAS GQ strongly supports our model that the core region binds to NRAS GQ by adopting the ssRNA-bound, closed conformation, accompanied by the unfolding of the GQ structure (Fig. 5f, g).

## GQ propensity of DDX3X's target transcripts

The NMR experiments and biochemical assays conducted in vitro supported the importance of the GQ structure in target recognition by DDX3X. We then asked whether this GQ-binding specificity defines the translational regulation by DDX3X under physiological conditions. Previous studies have shown that the putative GQ motif, $(G_2-L_{12})_4$, is more prevalent in RNA fragments that bind to the RG motifs in the N-IDR, and that GQ recognition by DDX3X is involved in the translational regulation of ribosomal protein synthesis[36,73]. Motivated by these findings, we extended our analysis to a global set of DDX3X-regulated translational targets to explore the general functional implications of GQ-specific binding by DDX3X.

Several comprehensive studies have identified sets of genes translationally regulated by DDX3X using CLIP-seq and ribosome-profiling[7,10,21]. Among these studies, we focused on the set of genes identified by Calviello et al.[7], where DDX3X-dependent changes in the translational levels of genes such as *RAC1* and *ODC1* are confidently supported by both in vitro and in cell assays. Given the general role of DDX3X in stimulating the translation of a subset of genes, we selected transcripts categorized as having downregulated translational efficiency (TE-down) upon DDX3X knockdown by siRNA (a total of 208 transcripts) and compared the GQ propensity of these transcripts with that of non-target genes (a total of 8783 transcripts). To analyze DDX3X's impact on different mRNA regions, we performed the comparisons separately for the 5′-UTR, protein-coding, and 3′-UTR regions.

We calculated the fraction of nucleotides with a prediction score greater than the threshold value (GQ fraction) and the maximum score (Max Score) for each transcript using the rG4detector software[63]. We then compared the distributions of these parameters between DDX3X targets (TE-down) and non-targets (otherwise) (Fig. 6a). Notably, the distributions of both the GQ fraction and Max Score values were significantly higher in DDX3X target genes compared to non-targets, with the difference being more prominent in the 5′-UTR region (median GQ fractions: 0.22 vs 0.097 in the 5′-UTR, $p < 0.001$; 0.041 vs 0.031 in the coding region, $p < 0.001$; and 0.069 vs 0.037 in the 3′-UTR, $p < 0.001$). Focusing on the 5′-UTR region, 86% of DDX3X target genes (179/208) contain at least one putative GQ-forming region with an rG4detector score above the threshold of 0.6, and this number increases to 95% (197/208) when a slightly lower threshold of 0.5 is used. Given that DDX3X has been proposed to bind to long transcripts[5,9], we also compared the transcript lengths, considering that longer transcripts

are expected to have many more potential binding sites, thus exhibiting higher affinity. While we found that the 5′-UTR regions were longer in DDX3X targets, the difference was modest (median values of 231 nt vs 202 nt, $0.01 < p < 0.05$), suggesting that DDX3X binding is not solely dependent on 5′-UTR length (Supplementary Fig. 25a). The enrichment of GQ segments was further confirmed using other predictors, G4Hunter[64] and pqsfinder[65] (Supplementary Fig. 25b, c). These observations suggest that the high content and stability of GQ structures, particularly in the 5′-UTR, are key factors that determine whether a gene is translationally regulated by DDX3X.

## GQ-recognition in IRES-mediated translational regulation

Although we have so far focused on the target specificity of DDX3X in the context of canonical cap-dependent translational regulation, it is reasonable to expect that such GQ-mediated target recognition by DDX3X also plays a role in other modes of translational control. To further strengthen the physiological relevance of GQ-mediated DDX3X recognition, we included the transcript of the microphthalmia-associated transcription factor (MITF) gene in our analysis. Phung and Cieśla et al.[22] previously proposed that DDX3X regulates non-canonical translation of *MITF* via an internal ribosome entry site (IRES) located within its 5′-UTR, which is closely linked to melanoma progression through its effects on cell proliferation and migration. The direct binding of DDX3X to the 5′-UTR of the *MITF* transcript and its functional consequences on cell proliferation have been well characterized.

Given that the 5′-UTR of the *MITF* transcript contains a predicted GQ-forming sequence around nucleotide 70, as identified by rG4detector (Supplementary Fig. 26a), we monitored N-IDR binding to this GQ-forming segment (*MITF* 62−86) as well as to a negative control fragment (*MITF* 7−31). The GQ structure of the *MITF* 62−86 fragment was confirmed by [1]H NMR analysis (Supplementary Fig. 26b). Notably, a marked intensity reduction of the N-IDR signals was observed upon addition of the GQ-forming *MITF* 62−86 fragment, while no such change was seen with the *MITF* 7−31 fragment, suggesting that the GQ segment in *MITF* is responsible for DDX3X recognition (Supplementary Fig. 26c). Importantly, the identified GQ-forming region (*MITF* 62−86) overlaps with the DDX3X-binding SL3B motif (*MITF* 68−93), which constitutes part of the IRES element in the *MITF* transcript. Our results, together with extensive cellular evidence reported by Phung and Cieśla et al.[22], strongly support the physiological significance of GQ-recognition by DDX3X, and such selective recognition plays a key role in various modes of translational regulation including both canonical cap-dependent and non-canonical IRES-mediated pathways.

## Discussion

The DDX RNA helicases are an important class of proteins that regulate gene expression at the translational level, participating in various physiological processes. While numerous studies have identified genes translationally regulated by DDX helicases, the molecular mechanism by which these helicases selectively recognize specific transcripts has largely remained unclear. Building on the study by Herdy et al., which identified DDX3X as a physiological binder of NRAS GQ[36,73], we set out to explore the potential role of the GQ binding in the selective recruitment of mRNA substrates and to investigate the structural basis underlying this process, focusing primarily on the role of the N-IDR in substrate RNA binding and its influence on helicase activity. We demonstrated that the N-IDR, despite its lack of a well-defined structure, can selectively recognize the GQ structure of RNA. This selective interaction is primarily mediated by arginine- and aromatic-rich segments of the N-IDR, which preferentially bind to the characteristic G-tetrad structure through a combination of electrostatic and multivalent π-interactions. The physiological importance of this interaction is further supported by findings that the N-IDR binds to the GQ-forming segment of the 5′-UTR in DDX3X targets, *RAC1*, *ODC1*, and

*MITF*, and that the putative GQ-forming sequences are more highly enriched in the 5′-UTRs of the mRNA transcripts translationally regulated by DDX3X. Our results highlight the key functional role of the GQ structure as a marker for the selective translational regulation by DDX3X in both canonical cap-dependent and non-canonical IRES-mediated mechanisms, wherein the two synergistic functions of DDX3X, the specific binding of the N-IDR to GQ-containing segments and the subsequent unfolding of the GQ structure by the helicase core, are at play at the molecular level (Fig. 6b). Our NMR results with the N-Core-MBP construct are in line with a model in which the N-IDR retains its flexible structure within the full-length context and contributes to GQ-specific recognition through its binding activity (Supplementary Fig. 10). Nonetheless, we note that the interactions between the N-IDR and the helicase core, which are likely transient in nature, were also suggested, and such N-IDR-core interplay might play a role in the intramolecular transfer of the RNA substrate or in fine-tuning of helicase activity as recently proposed for the yeast counterpart Ded1p[53]. While our model assumes that DDX3X directly participates in unfolding GQ structures based on the NRAS GQ-unfolding assay (Fig. 5), we also underscore the importance of an alternative pathway in which DDX3X stimulates translation by efficiently recruiting ribosomes and translational initiation factors[6,7] through target RNA recognition by the N-IDR.

Our NMR binding analyses using the 5′-UTR RNA fragments from *RAC1* and *ODC1* transcripts demonstrated that the N-IDR binds to the GQ-forming segments in these physiological substrates. We acknowledge that these analyses used relatively short RNA fragments (25 nt), which may not fully reflect the dominant structural features of full-length mRNAs; however, our NMR results are highly consistent with the translational activity assays of *RAC1* and *ODC1* deletion mutants reported by Wilkins et al.[12]. The region of the *RAC1* fragment with the highest affinity toward the N-IDR, nucleotides 53−77, overlaps with the region where a deletion mutation (nucleotides 40−78) abolished DDX3X's translational regulation. Similarly, in the analyses by Wilkins et al.[12], the translational repression of *ODC1* was not abolished by a single deletion mutant, which can be explained by our finding that *ODC1* has two major GQ-forming segments, nucleotides 150−174 and 279−303, both of which showed comparable affinity for the N-IDR. Additionally, the functional role of the GQ structure is consistent with previous studies, which showed that DDX3X-sensitive transcripts are generally highly enriched in guanosine[9,10,16,22,36,74].

Although the detailed atomic structure of the GQ-bound state of the N-IDR is not available due to its inherently weak affinity and the inhomogeneous nature of the complex, our NMR analyses still provide some key structural insights into the interaction, highlighting the importance of electrostatic interactions and π-interactions involving the arginine-rich segments. While IDR-RNA interactions are often thought to be primarily driven by electrostatic interactions between the negatively charged RNA phosphate backbone and polycationic IDR chains, our results indicate that π-interactions also make significant contributions as commonly observed in interactions involving RNA-folded protein complexes[75]. In the sequence of the N-IDR, the arginine residues are clustered with one or two glycine residues (Supplementary Fig. 1), typically referred to as arginine-glycine (RG) or arginine-glycine-glycine (RGG) motifs[60,76]. Consistent with our results, a number of studies have shown that arginine-rich peptides or RG/RGG domains of proteins interact with DNA or RNA GQ structures[54,77−83]; however, high-resolution structural information for such complexes is only available in a few favorable cases[84−86]. For example, in the structure of the high-affinity complex between the RGG peptide of fragile X mental retardation protein (FMRP) and GQ-forming sc-1 RNA[84,85], arginine residues form highly complementary interactions, primarily with the junction of duplex and GQ regions. The side chains of the arginine residues do not directly interact with the G-tetrad core within the GQ structure[84,85]. This is in contrast with our NMR results, where marked

CSPs were observed in the G-tetrad region in the N-IDR and telomeric 12mer GQ complex (Fig. 3f, g), reflecting that the aromatic guanine base moieties are involved in the interaction. In this regard, we speculate that the binding mode of the N-IDR to GQ may differ from that found in the high-affinity FMRP RGG-sc-1 RNA complex. Instead, our NMR results suggest that the specificity for GQ RNAs may arise from a collection of weak π-interactions, such as π-π interactions and cation-π interactions, involving the arginine and aromatic side chains and guanine bases[60]. We propose that similar interactions define the binding between RG/RGG-rich IDRs and GQ RNAs. Since RG and RGG motifs are frequently found in RNA-binding proteins and play important roles in the substrate binding[60,76,78], our structural findings have broad implications for understanding the molecular interactions of these motifs.

A number of experimental observations, such as increased $R_2$ values in the bound state, low stoichiometry in ITC, and apparent Hill coefficients deviating from one in EMSA, suggest that the complex formed between the N-IDR and GQ RNA cannot be described as a homogeneous one-to-one complex, rather indicate that multiple N-IDR and GQ RNA molecules are involved in forming a heterogeneous complex with variable and not uniformly-defined stoichiometry. This is not unexpected, given that N-IDR contains multiple RR clusters, each of which can independently bind RNA with comparable affinity, thereby allowing for a variety of distinct binding modes. We also underscore the likely involvement of multiple DDX3X molecules in complex formation, potentially enabling self-oligomerization of DDX3X around the RNA ligand. Several studies have proposed that multiple DDX3X molecules are involved in the RNA unwinding process[26,27,87], and that the formation of nanometer-scale DDX3X-RNA clusters underlies its catalytic activity[88]. Additionally, such self-association may also represent a nucleation event that precedes the formation of phase-separated cellular condensates such as stress granules[28,30,31]. We therefore emphasize that the heterogeneous nature of the N-IDR-GQ RNA complex likely reflects a physiologically relevant interaction that facilitates efficient catalysis and/or precise and dynamic regulation of sub-cellular organization.

Since the formation of a stable GQ structure in the 5′-UTR negatively regulates translation[47,89], preferential binding to structured GQ RNAs would be beneficial for efficiently targeting such translationally repressed mRNA transcripts and participating in the unfolding of the GQ structure to stimulate their translation. We propose that the balance between translational repression by 5′-UTR GQ formation and the unfolding activity of DDX helicases enables nuanced control of the translation landscape depending on cellular conditions. For example, the enrichment of GQ structures in the 5′-UTR has been suggested in some oncogenes, which can be translationally controlled via DDX-dependent unfolding of the GQ structure[47,90]. By exploiting the GQ structure in mRNA transcripts as a marker, the expression level of these genes can be tightly regulated through control of DDX activity depending on cellular stress levels, cell cycle, and developmental stage[10,24,74]. Such balanced control, which exploits the interplay between 5′-UTR structure and helicase unfolding activity, has recently been proposed in yeast stress responses. In these processes, the inactivation of Ded1p, a yeast counterpart of DDX3X, plays a central role in translational regulation under stress conditions by localizing Ded1p to stress granules[91]. Intriguingly, recent studies have shown that the N-IDR of DDX3X is also directly involved in the formation of and localization to stress granules[21,28,30,31]. Therefore, the sequestration of DDX3X through the formation of stress granules, possibly with GQ RNAs, might be one of the underlying mechanisms that alter the translational landscape under stress conditions in humans as well. Consistent with this notion, Turner et al. reported that endogenous GQ RNAs colocalize with the stress granule marker G3BP1, as detected using GQ-binding molecules, suggesting that GQ structures are maintained within cellular condensates[63]. Several in vitro studies have also

supported the preservation of GQ structures within protein condensates[92,93]; however, Luo et al. recently reported conflicting evidence suggesting the melting of GQ structures within in vitro DDX4 condensates[94]. These observations indicate that the folding stability of GQ RNAs within condensates may vary depending on their sequence, the type of condensate, and the cellular context, potentially representing an additional layer of regulation in biological responses.

In summary, we demonstrated that the flexible N-IDR of DDX3X plays a pivotal role in substrate RNA recognition. We propose that the interaction between the N-IDR and the well-folded GQ structure of RNA is a key factor that defines the physiological selectivity of translational regulation by DDX3X. Together with our previous results[34], our NMR studies illustrate a remarkable molecular mechanism of DDX3X, where two distinct binding interactions synergistically contribute to its physiological activity: the folded core region preferentially binds to unstructured ssRNA for helicase activity, while the unstructured N-IDR specifically recognizes the well-folded GQ segment of the substrate RNA. These two distinct interplays between the ordered and disordered parts of the molecules enable DDX3X to simultaneously exert substrate specificity and efficient enzymatic activity. In this regard, our results highlight the critical role of solution NMR spectroscopy in characterizing the disordered nature of biomolecules, thereby advancing our understanding of the various physiological processes dynamically regulated by complex, elaborate molecular machineries.

## Methods

### Protein expression and purification

The human DDX3X (UniProt: O00571) gene was synthesized by Eurofins Genomics. The DDX3X gene with the N-terminal His$_6$-MBP or His$_6$-SUMO tag was cloned into a modified pET28a(+) vector (Novagen 69864) containing the T7pCONS promoter and the translation initiation region, TIR-2[95]. All DDX3X variants, except the RtoK variant, were introduced by inverse polymerase chain reaction[96] and verified by sequencing. The RtoK variant gene (residues 1–132) was synthesized by Eurofins Genomics and cloned into the pET28a(+) vector as described above. The sequences of all the DDX3X variants and DNA primers used in this study were summarized in Supplementary Data 1.

For producing the N-terminally His$_6$-MBP-tagged DDX3X proteins, transformed *Escherichia coli* BL21(DE3) cells (ThermoFisher Scientific C600003) were grown in LB medium at 37 °C. Cells were induced by adding 1 mM isopropyl β-ᴅ-1-thiogalactopyranoside (IPTG) (Nacalai Tesque 19742-07) at an OD$_{600}$ of ~1.0 and grown for ~18 h at 18 °C. The proteins were first purified on an Amylose resin (New England Biolabs E8021S). The resin was washed using a buffer containing 20 mM Tris-HCl (pH 8.0) (Nacalai Tesque 35434-21) and 1 M NaCl (Nacalai Tesque 31320-05), then the proteins were eluted in the buffer containing 10 mM maltose (Nacalai Tesque 21115-02). For preparing the DDX3X core without the N-terminal His$_6$-MBP tag, the tag was cleaved by adding TEV protease, then the cleaved tag was removed by passing through a Ni-NTA Agarose resin (QIAGEN 30230). The proteins were further purified by size exclusion chromatography on a Superdex™ 200 Increase column (Cytiva 28990944) in a buffer containing 20 mM Tris-HCl (pH 8.0), 1000 mM NaCl, and 1 mM dithiothreitol (DTT) (Nacalai Tesque 14112-94). The proteins were buffer exchanged using an Amicon® Ultra MWCO30K concentrator (Merck UFC903024). The protein concentration was estimated based on absorbance at 280 nm using molar extinction coefficients of 142,100 M$^{-1}$ cm$^{-1}$ for MBP-FL (residues 1–662), 125,140 M$^{-1}$ cm$^{-1}$ for MBP-N-Core (residues 1–607) and MBP-Core-C (residues 132–662), 108,180 M$^{-1}$cm$^{-1}$ for MBP-Core (residues 132–607), and 40,340 M$^{-1}$cm$^{-1}$ for Core without His$_6$-MBP tag.

For producing unlabeled DDX3X N-IDR (residues 1–132) proteins, transformed *E. coli* BL21(DE3) cells were grown in LB medium. [U-$^{15}$N]-labeled N-IDR proteins were produced by using H$_2$O M9 medium supplemented with 1.0 g/L [$^{15}$N] ammonium chloride (Cambridge Isotope Laboratories NLM-467), while [U-$^{13}$C, $^{15}$N]-labeled N-IDR proteins

were produced by using $H_2O$ M9 medium supplemented with 1.0 g/L [$^{15}$N] ammonium chloride and 3.0 g/L [U-$^{13}$C] D-glucose (Cambridge Isotope Laboratories CLM-1396). Cells were induced by adding 1 mM IPTG at an $OD_{600}$ of ~1.0 and grown for ~4 h at 37 °C. The proteins were purified under denaturing conditions to remove bound nucleic acids. The cells were lysed in a buffer containing 20 mM Tris-HCl (pH 8.0), 300 mM NaCl, and 6 M guanidine hydrochloride (GdHCl) (Nacalai Tesque 17318-95). The proteins were purified by Ni$^{2+}$-affinity chromatography using a Ni-NTA Agarose resin. The resin was washed with the lysis buffer containing 6 M GdHCl, followed by the buffer without GdHCl. The resin was washed with the buffer containing 20 mM imidazole, and then the proteins were eluted in the buffer containing 300 mM imidazole. The N-terminal His$_6$-SUMO tag was cleaved by the addition of Ulp1 protease. The proteins were further purified by size exclusion chromatography on a HiLoad® 16/600 Superdex™ 75 pg (Cytiva 28989333) or Superdex™ 75 Increase 10/300 GL (Cytiva 29148721) column in a buffer containing 20 mM Tris-HCl (pH 8.0), 300 mM NaCl, and 1 mM DTT. The proteins were buffer exchanged using an Amicon® Ultra MWCO3K concentrator (Merck UFC900396). The protein concentration was estimated based on absorbance at 280 nm using molar extinction coefficients of 16,960 M$^{-1}$ cm$^{-1}$ for the wild type, RtoK, and RKtoA variants, and 15,470 M$^{-1}$ cm$^{-1}$ for the FYtoA variant.

RR fragment proteins were purified using a similar protocol to that for the N-IDR, except that the size exclusion chromatography was performed on a Superdex™ 75 Increase 10/300 GL or Superdex peptide HR10/30 column (Pharmacia 17-1453-01) in a buffer containing 150–200 mM ammonium carbonate (Nacalai Tesque 02421-85). The purified fragment proteins were lyophilized and subsequently dissolved in the desired buffer. The protein concentration of RR-1 and RR-3 was estimated based on absorbance at 280 nm using molar extinction coefficients of 8480 M$^{-1}$ cm$^{-1}$ for RR-1 and 5500 M$^{-1}$ cm$^{-1}$ for RR-3. The concentration of RR-2 was estimated based on absorbance at 205 nm using a molar extinction coefficient of 115,870 M$^{-1}$ cm$^{-1}$ [97].

For producing the C-terminally MBP-tagged [U-$^{15}$N]-N-Core-MBP (residues 1–607) proteins, transformed *E. coli* BL21(DE3) cells were grown in $H_2O$ M9 medium supplemented with 1.0 g/L [$^{15}$N] ammonium chloride and 1.0 g/L [$^{15}$N] Celtone® base powder (Cambridge Isotope Laboratories CGM-1030P-N). Cells were induced by adding 1 mM IPTG at an $OD_{600}$ of ~1.0 and grown for ~20 h at 18 °C. Proteins were purified using the same protocol as for the N-terminally MBP-tagged proteins. The protein concentration was estimated based on absorbance at 280 nm using a molar extinction coefficient of 125,140 M$^{-1}$ cm$^{-1}$.

For producing [Metε-$^{13}$C$^1$H$_3$]-labeled E348Q DDX3X core (residues 132–607) proteins, transformed *E. coli* BL21(DE3) cells were grown in $H_2O$ M9 medium at 37 °C. For selective Metε methyl $^{13}$C$^1$H$_3$-labeling, 100 mg/L L-methionine-(methyl-$^{13}$C) (ISOTEC 299146) was added 1 h before induction of protein overexpression[98]. Cells were induced by adding 1 mM IPTG at an $OD_{600}$ of ~1.0 and grown for ~4 h at 37 °C. Proteins were purified as described previously[34]. The proteins were first purified on Ni$^{2+}$-affinity chromatography using a Ni-NTA Agarose resin. The resin was washed using a buffer containing 20 mM Tris-HCl (pH 8.0) and 1000 mM NaCl, then the proteins were eluted in a buffer containing 20 mM Tris-HCl (pH 8.0), 300 mM NaCl, and 300 mM imidazole. The N-terminal His$_6$-SUMO tag was cleaved by the addition of Ulp1 protease. The proteins were further purified by size exclusion chromatography on a Superdex™ 200 Increase column in a buffer containing 20 mM Tris-HCl (pH 8.0), 300 mM NaCl, and 1 mM DTT. The proteins were buffer-exchanged into 20 mM potassium phosphate (pH 7.0) (Nacalai Tesque 28726-05 and 28721-55), 300 mM KCl (Nacalai Tesque 28514-75), and 5 mM DTT using an Amicon® Ultra MWCO30K concentrator and diluted into a low-salt buffer containing ATP and RNA. The protein concentration was estimated based on absorbance at 280 nm using a molar extinction coefficient of 40,340 M$^{-1}$ cm$^{-1}$.

## RNA preparation
All RNA oligos were synthesized by Hokkaido System Science Co., Ltd. The GQ RNA was prepared by first heating it to 90 °C for 3 min at a concentration of ~200–300 μM in a buffer containing 20 mM potassium phosphate (pH 7.0) and 50 mM KCl, followed by cooling to 4 °C with a temperature ramp rate of 0.1 °C/s. The formation of the GQ structure was confirmed by recording a $^1$H NMR spectrum. The concentration of each RNA was estimated based on absorbance at 260 nm, using a molar extinction coefficient determined by the nearest-neighbor approach[99]. The concentration of RNA as a single strand is used throughout the paper, except for ITC analyses. The sequences of all the RNAs used in the NMR experiments are summarized in Supplementary Data 2.

## dsRNA unwinding assay
The RNA unwinding activity of DDX3X was monitored by following the displacement of the fluorescently labeled 18mer strand from the complementary 36mer, as described previously[34,38]. The dsRNA substrate was prepared by annealing a 5′-FAM (6-Carboxyfluorescein)-labeled 18mer RNA (5′-CCCAAGAACCCAAGGAAC-3′) and an unlabeled 36mer RNA (5′-ACCAGCUUUGUUCCUUGGGUUCUUGGGAGCAG CAGG-3′, wherein the underlined region is complementary to the labeled 18mer). An excess amount of unlabeled 18mer trap ssRNA was included in the reaction mixture to prevent reannealing of the labeled strand. The reaction mixture comprised 0 to 5 μM DDX3X protein, 10 nM dsRNA, 200 nM trap ssRNA, 20 mM HEPES-NaOH (pH 7.4) (Nacalai Tesque 17546-05), 50 mM NaCl, 5 mM ATP (Nacalai Tesque 08886-64), 5 mM $MgCl_2$ (Nacalai Tesque 20908-65), 5 mM DTT, and 5% glycerol (Nacalai Tesque 17017-93). Reactions were started by adding an ATP-Mg stock solution, followed by incubation at 37 °C for 30 min. The reaction was quenched by mixing with a sodium dodecyl sulfate solution (final 0.1%) (Nacalai Tesque 02873-75); then the samples were separated by native PAGE on a 20% polyacrylamide gel at 150 V for 90 min in Tris-glycine buffer (25 mM Tris, 192 mM glycine) (Nacalai Tesque 17141-95). The gel was imaged on LuminoGraph I equipped with WSE-5600 CyanoView (excitation wavelength at 505 ± 25 nm) (ATTO) and analyzed by using CS Analyzer4 (version 2.4.5) (ATTO).

## GQ RNA unfolding assay monitored by fluorescence quenching
The GQ-unfolding activity was monitored by recording the fluorescence spectra of 5′-FAM-labeled, 3′-BHQ1-labeled NRAS GQ (5′-UGUGGGAGGGGCGGGUCUGGGUGC-3′) using an RF-6000 spectrofluorometer (Shimadzu). The excitation wavelength was set to 490 nm, and fluorescence was measured in the range of 500 nm to 600 nm with a scan rate of 600 nm/min. The excitation and emission bandwidths were set to 5 nm. The reaction mixture comprised 0 to 340 nM DDX3X protein or 400 nM complementary RNA strand (5′-GCACCCA-GACCCGCCCCUCCCACA-3′), 20 nM fluorescently labeled NRAS GQ, 20 mM potassium phosphate (pH 7.0), 50 mM KCl, 1 mM ATP, 1 mM $MgCl_2$, and 1 mM DTT. All of the measurements were performed at 25 °C. We did not attempt to convert the increase in fluorescence into the unwound ratio of NRAS GQ as previously done[70], because NRAS GQ remains stably folded even in the absence of KCl[47], making it difficult to reliably measure the maximum fluorescence intensity corresponding to complete unfolding. The fluorescently labeled NRAS GQ samples, with and without the complementary strand, were separated by native PAGE on a 20% polyacrylamide gel as described in the dsRNA unwinding assay method.

## GQ RNA unfolding assay monitored by hybridization
The GQ-unfolding activity of DDX3X was assessed by monitoring accelerated hybridization to a complementary RNA strand[68]. The reaction mixture comprised 50 nM 5′-FAM-labeled NRAS GQ (5′-UGUGGGAGGGGCGGGUCUGGGUGC-3′), with or without 100 nM DDX3X protein (MBP-FL), and with or without 100 nM complementary

RNA strand (5′-GCACCCAGACCCGCCCCUCCCACA-3′) in a buffer consisting of 20 mM potassium phosphate (pH 7.0), 50 mM KCl, 1 mM ATP, 1 mM MgCl$_2$, and 1 mM DTT. After incubation at room temperature for 10 min, 1/5 volume of proteinase K solution (Nacalai Tesque 15679) was added to quench the reaction. The samples were then separated by native PAGE on a 20% polyacrylamide gel as described in the dsRNA unwinding assay method. Gels were imaged using the ChemiDoc Go system with a Blue Sample Tray (Bio-Rad) and analyzed using ImageLab (version 6.1.0) (Bio-Rad).

### Electrophoretic mobility shift assay (EMSA)
The binding affinity was assessed by monitoring the band shift of fluorescently labeled RNA upon complex formation with DDX3X[50]. Since the bands of DDX3X-bound RNA were not clearly observable in most cases, the bound fraction was estimated based on the intensity reduction of the free RNA probe bands. FAM labeling was introduced at the 3′-end for poly-U$_{12}$ ssRNA and GC-14mer dsRNA, and at the 5′-end for all other RNAs. The reaction mixture comprised 0 to 45 μM N-IDR or 0 to 1 μM MBP-FL/MBP-Core-C protein, 50 nM (as a single strand) fluorescently labeled RNA, 20 mM potassium phosphate (pH 7.0), 125 mM KCl, 1 mM DTT, 0.005% Tween-20 (MP Biomedicals, 103168), and 10% glycerol. 1 mM ATP-Mg was added to measure the affinity for MBP-FL/MBP-Core-C protein. Samples were separated using polyacrylamide gels: 8% for all RNAs except the 14mer-tetra-loop, which was run on a 12% gel. All gels were prepared with 0.5× TBE buffer (44.5 mM Tris-borate, 1 mM EDTA; Nacalai Tesque 35440-31). Electrophoresis was carried out at 150 V for 35–40 min (for 8% gel) or 50–60 min (12% gel) in 0.5 × TBE buffer at room temperature. Gels were imaged using the ChemiDoc Go system with a Blue Sample Tray (Bio-Rad) and analyzed using ImageLab (version 6.1.0, Bio-Rad). Apparent $K_d$ and Hill coefficient ($n$) values were estimated by fitting the bound fraction curves to a standard Hill-type equation using in-house written programs (Python 3.12.4, see the Code Availability section).

### ITC experiments
The N-IDR was dialyzed against a buffer containing 20 mM potassium phosphate (pH 7.0), 50 mM KCl, and 1 mM Tris(2-carboxyethyl)phosphine (Merck 646547) immediately prior to the experiments, using Slide-A-Lyzer™ MINI Dialysis Devices 10 K MWCO (Thermo Scientific 88401). Lyophilized RR fragment proteins were directly dissolved in a buffer containing 20 mM potassium phosphate (pH 7.0) and 50 mM KCl. For each ITC experiment, reaction heats were measured by titrating 3-μL aliquots of telomeric GQ or NRAS GQ into the N-IDR or RR solution at 25 °C using a Microcal PEAQ-ITC system (Malvern). The concentrations of the syringe and cell solutions are indicated in each figure legend. The titration data were analyzed using the standard binding model assuming one-set of sites with MicroCal PEAQ-ITC Analysis Software (version 1.41) (Malvern). During the fitting process for obtaining the $K_d$ and $N$ (number of binding sites) values, the dimer concentration was used for telomeric GQ RNA[45], while the monomer concentration was used for NRAS GQ RNA[48].

### NMR experiments
All NMR measurements were performed using Bruker AVANCE-III HD or Bruker AVANCE NEO spectrometers with a cryogenically cooled *z* pulsed-field gradient triple-resonance TCI/TXO probe or a cryogenically cooled *z* pulse-filed gradient quadruple-resonance QCI-P probe at 600 MHz, 800 MHz, or 1 GHz. All NMR data were acquired using Bruker TopSpin (version 3.5pl7, 4.1.4, or 4.2.0). All spectra were processed using the NMRPipe suite of programs[100] (version 11.1), analyzed by NMRFAM-SPARKY[101] (version 1.470), and visualized using the Python package nmrglue[102] (version 0.90).

Backbone resonance assignments of the wild-type, RtoK, FYtoA, and RKtoA variant N-IDR were obtained by standard 3D triple-

resonance experiments[103,104] including HNCO, HN(CA)CO, HNCACB, CBCA(CO)NH, HNN, (H)CC(CO)NH-TOCSY, and H(CC)(CO)NH-TOCSY at 600 MHz or 800 MHz and 10 °C, using a 90–700 μM [U-$^{13}$C, $^{15}$N]-labeled samples in an NMR buffer containing 20 mM potassium phosphate (pH 6.0 or 7.0), 250 mM KCl, 5 mM DTT, and 5% D$_2$O (for the wild type, RtoK, and RKtoA variants) or a buffer containing 20 mM MES (pH 6.0) (Nacalai Tesque 02442-44), 200 mM KCl, 1 mM EDTA (Nacalai Tesque 15111-45), 5 mM DTT, and 5% D$_2$O (for the FYtoA variant). All 129 observable signals in the $^{15}$N-$^{1}$H correlation spectrum (excluding 2 proline residues and the N-terminal residue) were successfully assigned in the wild-type N-IDR. The assignments of each RR fragment were obtained either by transferring the assignments from the full N-IDR, by recording $^{15}$N-edited $^{1}$H-$^{1}$H NOESY (mixing time of 200 ms) and $^{15}$N-edited $^{1}$H-$^{1}$H TOCSY using the 10 kHz DIPSI-2 mixing sequence[105,106] (mixing time of 69.5 ms), or by recording a set of 3D triple-resonance experiments (for RR-2). 3D datasets were recorded with non-uniform-sampled time-domain data based on Poisson-Gap sampling schedules[107], and the spectra were reconstructed using SMILE[108].

$^{15}$N-$^{1}$H HSQC spectra of the N-IDR/RR fragments were recorded using a gradient-enhanced sequence[109]. The measurements were performed at 1 GHz and 10 °C, using a 50 μM [U-$^{15}$N] or [U-$^{13}$C,$^{15}$N]-labeled N-IDR/RR fragment sample in an NMR buffer consisting of 20 mM potassium phosphate (pH 7.0), 200 mM KCl, 5 mM DTT, 260 units/mL RNAsin® Plus RNAase inhibitor (Promega N2611), and 5% D$_2$O, unless otherwise specifically indicated. $^{15}$N-$^{1}$H HSQC spectra of N-Core-MBP were measured at 1 GHz and 10 °C, using a 30 μM [U-$^{15}$N]-labeled N-Core-MBP sample in the same NMR buffer. The errors in the signal height ratio were calculated using the signal-to-noise ratios of the two spectra, with a minimum error of 3%, determined based on the standard deviation of three independent biological replicates in representative experiments.

$^{15}$N and $^{1}$H CPMG relaxation dispersion experiments were recorded using a pulse scheme employing (y, y, x, −x) phase cycling[57,58,110,111]. The constant-time relaxation delay was set to 20 ms. $R_{ex}$ contributions were estimated from the difference in the relaxation rates measured at 50 Hz and 1000 Hz CPMG fields. The measurements were performed at 1 GHz and 10 °C using the NMR samples containing 50 μM [U-$^{15}$N]-labeled N-IDR, 20 mM potassium phosphate (pH 7.0), 200 mM KCl, 1 mM EDTA, 5 mM DTT, 260 units/mL RNAsin® Plus RNAase inhibitor, and 5% D$_2$O, with or without 25 μM telomeric GQ RNA or 10 μM NRAS GQ RNA.

$^{13}$C-$^{1}$H HMQC spectra of [Metε-$^{13}$C$^{1}$H$_3$]-labeled E348Q DDX3X core were recorded using a sequence that exploits the methyl-TROSY effect[112,113]. The measurements were performed at 1 GHz and 35 °C, using a 50 μM [Metε-$^{13}$C$^{1}$H$_3$]-labeled sample in an NMR buffer consisting of 20 mM potassium phosphate (pH 7.0), 100 mM KCl, 5 mM DTT, 5 mM ATP, 5 mM MgCl$_2$, and 5% D$_2$O. Methionine methyl assignments of DDX3X core were established in our previous study[34].

$^{1}$H 1D NMR spectra of RNA samples were measured with solvent suppression using the excitation sculpting method[114]. $^{1}$H-$^{1}$H 2D TOCSY spectra of telomeric GQ RNA were measured using the MLEV-17 mixing sequence[115] (mixing time of 43.8 ms) with the excitation sculpting solvent suppression. The measurements were performed at 1 GHz and 10 °C, using a 50 μM unlabeled RNA sample in an NMR buffer consisting of 20 mM potassium phosphate (pH 7.0), 200 mM KCl, and 5% D$_2$O. DTT was added to a final concentration of 5 mM during the titration of the N-IDR. The signal assignments were transferred from the previous study without correction, as the difference in buffer composition was minimal[45].

### Estimation of the GQ propensity
The GQ propensity of the 5′-UTR of the *RAC1* (Ensembl Transcript ID: ENST00000356142) and *ODC1* (Ensembl Transcript ID: ENST00000234111) transcripts was analyzed using the rG4detector software[63] (https://github.com/OrensteinLab/rG4detector)

in detection mode. In the analyses using G4Hunter[64] (https://github.com/AnimaTardeb/G4Hunter) and pqsfinder[65] (version 2.18.0) software, we used a minimum length of 25 and a threshold value of 1.0 for G4Hunter, and a minimum threshold score of 26 for pqsfinder to identify putative GQ-forming segments. To be consistent with previous studies, we used the 5′-UTR sequences from the paper by Calviello et al.[7]. Note that the length of the 5′-UTR sequence of these transcripts may differ between Ensembl releases, and the sequences used in this study can be found in Ensembl Release 80. The GQ propensity of the *MITF* transcript (Ensembl Transcript ID: ENST00000314557) was analyzed using the rG4detector software. For consistency, the numbering of the 5′-UTR follows that reported by Phung and Cieśla et al.[22].

The GQ fraction and maximum score for transcripts of DDX3X targets were analyzed using the list of genes from Supplementary Table 1 of the paper by Calviello et al.[7], using in-house written programs (Python 3.11.7 and R 4.3.2, see the Code Availability section). We first split the set of genes into two groups: those categorized as TE down (DDX3X target genes) and all others (TE up, concordant up, concordant down, and mixed) as non-target genes. The 5′-UTR, coding, and 3′-UTR sequences of the canonical form of each transcript were obtained from the BioMart Ensembl database (Release 111)[116,117]. Sequences longer than 80 nt (the window size of the rG4detector) were included in the analyses. To normalize for the differences in transcript length, we compared the GQ fraction by calculating either the fraction of nucleotides with a GQ prediction score greater than the threshold (0.6) using the rG4detector in detection mode or by calculating the fraction of the total length of non-overlapping GQ-forming segments as predicted by G4Hunter (minimum length 25, threshold 1.0) or pqsfinder (minimum score 26). The maximum score was taken from the highest score value found in each region. The distributions of the GQ fractions and maximum scores were compared between the two groups, DDX3X targets (TE-down) and non-targets (otherwise), separately for the 5′-UTR, coding, and 3′-UTR regions. The results were visualized using a violin plot, and statistical significance between the two groups was evaluated using the two-sided Mann-Whitney U test.

### Structural modeling

The structure of DDX3X bound to poly-$U_{10}$ was modeled using the SWISS-MODEL web server (https://swissmodel.expasy.org/)[118] using the structure of the VASA/AMPPNP/poly-$U_{10}$ complex (PDB 2DB3)[32] as a template. The structure of GC-14mer dsRNA was modeled using the MacroMoleculeBuilder software (version 3.4)[119] with the crystal structure of 14-bp $[U(UA)_6A]_2$ RNA (PDB ID: 1RNA)[120] as a template. The structures presented in the figures were visualized using the UCSF ChimeraX software (version 1.6.1)[121].

### Reporting summary

Further information on research design is available in the Nature Portfolio Reporting Summary linked to this article.

## Data availability

NMR assignments for the N-IDR of DDX3X have been deposited in the BMRB database under accession numbers 52738 for the wild type, 52739 for the RtoK variant, 52740 for the FYtoA variant, 52741 for the RKtoA-1 variant, 52742 for the RKtoA-2 variant, and 52743 for the RKtoA-3 variant. Structure data used in this study are available in the Protein Data Bank under accession codes 1RNA[120], 2DB3[32], 2KBP[45], 2KOC[44], 3IBK[46], 5E7M[26], and 7SXP[48]. The protein sequence used in this study is available from UniProt with the accession code of O00571 (DDX3X). The transcript sequences used in this study are available from Ensembl with the accession IDs ENST00000356142 (*RAC1*), ENST00000234111 (*ODC1*), and ENST00000314557 (*MITF*). Source data are provided with this paper.

## Code availability

The R and Python scripts used in this study along with the relevant data are available on https://doi.org/10.6084/m9.figshare.27134787.v1[122].

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

## Acknowledgements
This work was supported by the Japan Agency for Medical Research and Development (AMED) Grant Number JP21ae0121028 for I.S. and JSPS KAKENHI Grant Numbers JP24K09408 and JP25H02244 for Y.T. The authors thank Dr. Stephen N. Floor for sharing the details of the 5′-UTR sequences of genes translationally regulated by DDX3X. The authors thank Dr. Robert W. Harkness for helpful discussions regarding the interpretation of the ITC results. The authors thank Drs. Yutaka Kofuku, Shunsuke Imai, Yuji Tokunaga, and Yutaro Shiraishi for valuable discussions. Molecular graphics and analyses performed with UCSF ChimeraX, developed by the Resource for Biocomputing, Visualization, and Informatics at the University of California, San Francisco, with support from National Institutes of Health R01-GM129325 and the Office of Cyber Infrastructure and Computational Biology, National Institute of Allergy and Infectious Diseases.

## Author contributions
Y.T. conceived the project, purified the proteins, performed the biochemical assays, prepared the NMR samples, conducted the NMR experiments, analyzed the NMR data, and performed the bioinformatic analyses. Y.T., K.T., and I.S. designed the research and wrote the manuscript.

## Competing interests
The authors declare no competing interests.
