## [Transparent Peer Review file · Nature Communications]

Regulatory role of the N-terminal intrinsically disordered region of the DEAD-box RNA helicase DDX3X in selective RNA recognition

Corresponding Author: Professor Ichio Shimada

Version 0:

Reviewer comments:

Reviewer #1

(Remarks to the Author)

This study investigates the regulatory role of the N-terminal intrinsically disordered region (IDR) of the DEAD-box RNA helicase DDX3X in selective RNA recognition, addressing a topic of significant biological relevance. Using techniques such as nuclear magnetic resonance (NMR), the study demonstrates that the N-IDR preferentially recognizes structured RNAs, particularly G-quadruplex (GQ) structures. These findings provide valuable insights into the molecular mechanisms by which DDX3X regulates translation and highlight the crucial role of disordered regions in RNA recognition. While the study makes meaningful contributions, it could be further improved to enhance the manuscript's quality. For instance, the description of experimental findings primarily emphasizes observational results (e.g., changes in NMR signals) without sufficient quantitative analysis. The characterization of RNA G4 unwinding activity is simplistic and could be expanded to provide a more detailed understanding. Furthermore, the proposed mechanism of N-IDR function remains insufficiently detailed, and the potential synergy between the N-IDR and the helicase core remains underexplored. To strengthen the manuscript, I recommend addressing these points before it is considered for publication in the journal.

1. Abstract. The introduction to the research background is overly detailed, which detracts from the focus on the study's main findings.
2. Results. The influence of N- and C-terminal IDRs on DDX3X function.
 - a. It would be beneficial to present an SDS-PAGE image of all the proteins used in this study to demonstrate their purity.
 - b. In Fig.1, MBP is a 42.5 kD tag. The authors noted that the addition of the N-terminal MBP tag resulted in a relatively small decrease in activity (~1.8-fold larger apparent K_m in MBP-Core compared to Core), supporting the notion that the MBP tag does not significantly affect the helicase activity of DDX3X. However, this observation actually indicates that MBP attenuated the unwinding activity. Additionally, it is important to consider whether MBP could influence the assembly state of DDX3X.
 - c. The authors noted that "there was a nearly 6-fold reduction in activity by truncating the N-IDR and a 53-fold reduction by truncating both the N- and C-IDRs." The term "truncating the C-IDR" should be corrected to "truncating the N-IDR." Furthermore, it is unclear how these parameters were obtained. Additionally, the conclusion that the N-IDR is more significant than the C-IDR is not strongly supported by the results, as a direct comparison between truncating the C-IDR alone and truncating the N-IDR alone is lacking.
3. Results: Structural characterization of the interaction between the N-IDR and structured RNAs.
 - a. The results lack quantitative analysis of the binding affinity between the N-IDR and different RNA structures. Methods such as ITC, SPR, FA, or EMSA could be employed to provide more robust and detailed insights.
 - b. The NMR results suggest that the N-IDR shows the strongest binding to RNA G4. However, it remains unclear which substrate DDX3X, as a whole, binds to most effectively. Is the binding preference of DDX3X consistent with that of the N-IDR? Addressing this question would provide a more comprehensive understanding of the relationship between the N-IDR and the helicase core.
4. Results. Structural basis for the interaction between the N-IDR and GQ RNA.
 - a. The proposed multivalent binding model of the N-IDR is based solely on signal changes around residues 40, 90, and 120, without additional support from structural models. It is recommended to expand the discussion on the functional roles of

these binding sites.

- b. Are the RR-1, RR-2, and RR-3 motifs specific to RNA G4 binding, or do they also mediate similar multivalent interactions with other RNA substrates? Further investigation would help clarify their substrate specificity.
 - c. The authors stated: Although exchange broadening effects complicate precise estimation of the bound fraction, this result aligns reasonably well with the expected 24-fold decrease in binding affinity (e.g., a decrease in the bound fraction from ~64% to ~15% with a K_d increase from 10 μM to 240 μM at protein and RNA concentrations of 50 μM at 10 °C). However, it is unclear how these K_d values were determined. Providing additional details on the methodology or calculations used to derive these values would enhance the clarity and reproducibility of the results.
5. Results: Binding of the N-IDR to GQ segments in physiological mRNA substrates. While NMR is valuable for characterizing protein-substrate interactions, are there additional methods that could be employed to further validate these findings? For example, quantifying the binding affinities by comparing dissociation constants (K_d) across different substrate structures and various G4s using techniques such as ITC, SPR, or EMSA would provide a more comprehensive understanding of the binding properties.
6. Results: Characterization of the GQ-unfolding activity of DDX3X.
- a. The characterization of RNA G4 unwinding is relatively simplistic and could be improved. Incorporating additional methods commonly used for G4 unwinding studies, such as gel electrophoresis, could provide more robust and detailed insights.
 - b. Quantitative analysis is lacking. For instance, what is the unwinding ratio? Additionally, analyzing the unwinding kinetics over time would provide a more comprehensive understanding of the process.
 - c. There appears to be a discrepancy between the fluorescence assay in Fig. 5D, which shows no unwinding activity in the core, and the NMR assay in Fig. 5F, which suggests that RNA G4 is converted to ssRNA by the core. This potential contradiction should be clarified to ensure consistency and accuracy in the interpretation of the results.
7. Results: GQ propensity of DDX3X's target transcripts. The authors stated that the high content and stability of G-quadruplex (GQ) structures, particularly in the 5'-UTR, are key factors that determine whether a gene is translationally regulated by DDX3X. However, there is a lack of direct evidence within cells to support this claim. It remains unclear whether the regulation of these genes is truly mediated through RNA G-quadruplexes.
8. Discussion. It would be beneficial to discuss the differences between this work and the research by Herdy et al. [Reference 36].

Reviewer #2

(Remarks to the Author)

This outstanding manuscript describes a biophysical and cellular study of the mechanism of DDX3X on selective RNA recognition. Using NMR, EMSA, ITC, bioinformatics, and biochemical unwinding activity assays to unravel the molecular mode of interaction of the protein DDX3X that is an essential protein.

The authors show that the intrinsically disordered N- and C-terminal domains of the protein bind preferentially to RNA G quadruplexes, while they do not recognize single stranded, double stranded or stem-loop RNA structures. The interaction is entropically and (!) enthalpically driven. The authors show that this binding is localized to RG-repeat structures of in the IDRs of DDX3X. The enthalpic part stems from Arginine cation- π -interaction with the top and bottom tetrads of the RNA G4s. This is an outstanding manuscript that I strongly recommend to be published.

There is a single point that I would recommend the authors to consider. It would be interesting to see whether DDX3X can differentiate between RNA and DNA G4. Maybe, DDX3X is strictly cytosolic, and thus this might be irrelevant due to the non-nuclear localisation. But it is probably worthwhile to at least mention this. Further, the stability of RNA G4 in condensates is worthwhile to discuss.

Reviewer #3

(Remarks to the Author)

In "Regulatory role of the N-terminal intrinsically disordered region of the DEAD-box RNA helicase DDX3X in selective RNA recognition", Toyama et al explore the role of the intrinsic disordered regions (IDRs) of DDX3X, a member of the DEAD-box RNA helicase family that stimulates the initiation of translation of several mRNAs by binding to and unwinding their 5'-untranslated region (UTR). DDX3X is composed of two conserved folded domains, D1 and D2, which are responsible for the ATP binding and helicase activity, surrounded by IDRs. Using NMR and biochemical assays, the authors here investigate the role of the N-terminal IDR of DDX3X and how it contributes to the RNA selection of the enzyme targets.

DDX3X is an important helicase, with clear correlation to disease. Its mutations can cause neurological syndromes, and its perturbation has been found to cause cancer onset and progression, making this enzyme surely important to study. The mechanism of action of the catalytic domains (D1D2) has been extensively investigated in the past, also by the authors of this manuscript, but a detailed characterisation of DDX3X's IDRs has lagged significantly. The role of the N-terminal IDR of DDX3X in phase separation upon stress was reported (Shen et al, Mol Cell 2022, <https://doi.org/10.1016/j.molcel.2022.04.022>), but a mechanistic characterisation of the IDRs of DDX3X is still missing. This is not surprising, considering the challenges of studying IDRs. The characterisation provided by the authors enables a somehow better understanding of the overall mechanism of action of an enzyme that is paramount for the cellular physiology during development and adult life. Moreover, it attempts to describe the role of the IDRs in the general mechanism of DDX3X, which will be of use not only to the DEAD-box RNA helicase field, but more in general, to the whole field of scientists working on the mechanistic characterisation of proteins (most of which carry IDRs!).

Although this work is noteworthy, I found some of the discoveries redundant with previous work. The findings of the

contribution of the N-terminal region of DDX3X on the helicase activity were already previously reported by other colleagues in the field (See Figure 4 of He et al, *Biochemical and Biophysical Research Communications* 2024, <https://doi.org/10.1016/j.bbrc.2024.149964>), therefore the whole Figure 1, although clean and rigorous, states known facts. Moreover, the contribution of electrostatic interactions and π -interaction in IDR–RNA (and more generally protein–RNA) interaction is well known, as reviewed by Corley et al, *Molecular Cell* 2020, doi: 10.1016/j.molcel.2020.03.011. This makes the data reported in Figure 2 and Supplementary Figure 3 of low impact.

Before publication in *Nat Comm*, I recommend major reviews and clarifications from the authors.

1. The authors use NMR to characterise the interaction between the N-terminal IDR of DDX3X and a set of RNA targets, some single stranded—polyU—some structured—G-quadruplexes. Although the authors do provide a zoom-in of the 1H-15N correlation NMR spectra throughout the manuscript, they never provide full spectra. All the full spectra should be included; the editor can advise in what form this should be done (supplementary data or supplementary figures), but they should be surely included in the manuscript.
2. In their interaction analysis using 1H-15N correlation NMR, the authors state that very small shifts are observed, and the binding is claimed by mapping the change in peak intensity. However, in similar studies on intrinsically disordered proteins/regions, the interaction results in chemical shift perturbations. Can the author discuss why? Do they think that this is due to the formation of a fuzzy complex in solution or because of the weak interaction? Can this be due to aggregation? Have the authors tried to perform relaxation NMR or transverse-correlation experiments to corroborate the dynamics and nature of these interactions?
3. The authors perform the NMR experiments on IDR-RNA complexes at a salt concentration that is above the physiological conditions, as themselves explain at page 10, to push the interaction between protein and RNA. What is the landscape of the interaction at physiological salt concentration (~120 mM NaCl)? In what way is the interaction affected? Do the overall observed trends change?
4. The binding curves calculated by ITC are quite concerning. The authors state that the stoichiometry observed is off due to the bad temperament of the samples, but they still trust (up to a certain extent) the K_d values obtained. I am afraid that this must be repeated using an orthogonal technique (BLI, MST, EMSA...) to make sure that the results obtained describe the system correctly. Moreover, the fact that the authors observe aggregation raises a massive concern regarding all sets of NMR data presented. Can the authors state confidently that the reduction of signal observed in their NMR data is due to interaction rather than aggregation? Can the authors provide hard evidence of this (DLS or similar)? This also connects to point 2. The small shifts observed versus the change in peak intensity could be due to overall (micro or macro)aggregation phenomena happening in the NMR tube, and this must be investigated better.
5. The authors state that there are three binding sites in the N-IDR, around residues 40, 90 and 120, indicating that the binding cannot be described by a conventional one-site binding model with a single defined K_d value. I am afraid that the authors cannot state this confidently, as they do not present enough biophysical and structural data to support this claim. First, 15N-correlation data are never indicative of a direct interaction, as the change in intensity or shift of the amide peaks can be due to an indirect effect. The fact that the authors see three main binding regions in their NMR plots does not mean that the three sites are independent. In fact, unclear stoichiometry values from the ITC experiments make this data interpretation even weaker (see point 4). The observed NMR data are compatible with the N-IDR decorating the RNA, maintaining part of the protein in a fuzzy state. This is quite evident from the titration with NRAS GQ, where, at 200 mM salt, pretty much the whole region between residues 40 and 120 seems to participate to RNA binding. To prove their claim, the authors must take a different approach. They could either perform SAXS data to see whether the N-IDR goes from an extended to a globular state upon RNA-binding, in which case most likely the three sites are not independent. Moreover, they should test the RNA-binding activity of the three regions carrying the three different RGG boxes and see whether they can independently bind to the RNA targets (using any interaction technique of their choice).
6. From all their NMR data, it seems that the RGG box around site 40 is predominant. The authors have not discussed and thoroughly investigated this, but they should. A better description of this must be provided, to clarify whether the three sites are independent or not, as well as the perturbation of the RGG boxes around 90 and 120 is direct or indirect.
7. Figure 5e seems to be identical to what reported by the same authors in their previous work Toyama et al, *Nat Comm* 2024 15, Article number: 3303 at Figure 2d. The panel from the previous work states that the molar ratio is 2, while here it is 4, but they seem identical. Can the authors explain?
8. The interpretation of Figure 5e and 5f is quite confusing to me. Figure 5e shows the set of Methionine shifts upon binding of D1/D2 (core) to polyU10 (single stranded RNA), overlaid to the core in its apo state. Figure 5f shows the overlay of the complexes of core with polyU10 and core with NRAS GQ (folded) and the authors state that a “distinct set of bound signals was observed”. To me, the bound signals are virtually identical, only M330 shifts very minorly less, but this could be due to lower affinity towards NRAS GQ. The core-NRAS GQ was recorded at a very high excess of NRAS GQ. Now the authors state that the fact they see these shifts means that NRAS GQ binds in single-stranded state to the core, although the authors stated that the core does not unwind GQ (see Figure 5d middle panel). This, to me, seems to be quite an overclaim. If the core does not unwind the GQ, how is this RNA in a single stranded state when bound to the core? The same shifts observed between polyU10 and NRAS GQ are not enough to substantiate this claim. The authors must perform a more detailed characterisation of this system (using 1H-NMR of NRAS GQ in its bound state to the core to claim single stranding of the GQ, as well as a better NMR characterisation of the overall bound states).
9. The model in Figure 6b seems to indicate that, upon binding, the N-IDR fold onto the core, however, the authors do not provide evidence for this model. To confirm this, they must perform SAXS experiments with either the FL or the construct spanning residues 1-607 to see the claimed conformational change upon binding. Alternatively, NMR could be of use here too, looking at how the core is perturbed by the N-IDR in its apo and RNA-bound states.
10. The authors must confirm some of the statements on the role of the N-IDR in its tethered state to the D1/D2 core. The interaction between the N-IDR and RNA could be affected by the presence of the core (as somewhat the data in Figure 5f would suggest – see also point 8). Since the author wants to provide a wholesome description of this system, they must observe the interaction of the RGG boxes when these are tethered to the folded domains. Without this, the whole synergistic

description in the Discussion at page 22 is way too speculative to be included.

Very minor typos were identified.

1. Page 3: there is an extra space between that and unwinds in the sentence “an enzyme that unwinds structured RNA...”.
2. Page 11: there is an extra space between first and conducted in the sentence “we first conducted NMR binding experiments under....”.
3. Page 26: reference missing for inverse PCR cloning method.
4. Page 27: D-glc has been reported with the D in superscript.
5. En dash should be preferred to hyphen when intervals are reported (for example, page 28: 132–607 instead of 132-607).
6. Page 28: 1,000 mM should be corrected into 1000 or 1'000.
7. Page 29: reference missing for the nearest-neighbour approach.
8. Page 33: 10 °C should not carry a space.
9. Page 33: the word paper is misspelled in “paepr by Caliello et al”.

Version 1:

Reviewer comments:

Reviewer #1

(Remarks to the Author)

The authors have addressed my concerns in the revision. The manuscript is now ready for publication.

Reviewer #3

(Remarks to the Author)

I am absolutely thrilled with the thorough and precise improvements the authors have made to the manuscript. The additional work they have undertaken to address my concerns is impressive, and I am now fully satisfied with the revised version. Therefore, I enthusiastically recommend this manuscript for publication.

Special kudos to the authors for their honest and thoughtful interpretation of the complex formation, particularly their careful discussion of the potential presence of higher-order complexes based on the EMSA and ITC results. I also greatly appreciated the restructuring of the section on the interplay between the helicase domains and the disordered N-terminal region. In its revised form, this section is much clearer and has successfully convinced me of the functional relevance of the synergy between structured and disordered domains.

One minor issue I noted is that Supplementary Figure 2 is cited before Supplementary Figure 1 in the main text. I recommend correcting this for consistency.

Finally, I would like to sincerely thank the authors for their rigorous and commendable work. This manuscript will be of significant value to several scientific communities, including researchers working on DDX3X and biophysicists studying intrinsically disordered regions. I wish the authors all the best.

Point-by-point response to the reviewers' comments

We would like to express our sincere gratitude to all three referees for their valuable comments and suggestions on our manuscript. Based on the reviewers' insightful comments, we have carefully revised the manuscript. The reviewer's comments and our responses are shown in light blue and black text, respectively. The revisions made to the manuscript are highlighted in red font. We hope that these changes have satisfactorily addressed all of the concerns raised by the reviewers.

Reviewer #1:

This study investigates the regulatory role of the N-terminal intrinsically disordered region (IDR) of the DEAD-box RNA helicase DDX3X in selective RNA recognition, addressing a topic of significant biological relevance. Using techniques such as nuclear magnetic resonance (NMR), the study demonstrates that the N-IDR preferentially recognizes structured RNAs, particularly G-quadruplex (GQ) structures. These findings provide valuable insights into the molecular mechanisms by which DDX3X regulates translation and highlight the crucial role of disordered regions in RNA recognition. While the study makes meaningful contributions, it could be further improved to enhance the manuscript's quality. For instance, the description of experimental findings primarily emphasizes observational results (e.g., changes in NMR signals) without sufficient quantitative analysis. The characterization of RNA G4 unwinding activity is simplistic and could be expanded to provide a more detailed understanding. Furthermore, the proposed mechanism of N-IDR function remains insufficiently detailed, and the potential synergy between the N-IDR and the helicase core remains underexplored. To strengthen the manuscript, I recommend addressing these points before it is considered for publication in the journal.

We appreciate the reviewer's careful reading of our manuscript and their constructive suggestions. To address these points, we have included additional experimental data to strengthen our claims: (i) quantitative evaluation of RNA binding affinity of the N-IDR using electrophoretic mobility shift assay (EMSA); (ii) RNA GQ unfolding assay monitored by gel electrophoresis; (iii) EMSA analyses

of the interactions between full-length and N-IDR-truncated DDX3X with various RNA ligands; and (iv) structural analyses of the N-IDR when attached to the helicase core. The details of these results are presented in our point-by-point responses below. We sincerely thank the reviewer for their valuable critiques, which have helped us improve the strength of our conclusions and the overall clarity of the manuscript.

1. Abstract. The introduction to the research background is overly detailed, which detracts from the focus on the study's main findings.

We appreciate the reviewer's comment on the abstract. We have revised it to focus more clearly on the main findings of our study, rather than overstating the research background as shown below.

(Abstract)

DDX3X, a member of the DEAD-box RNA helicase family, plays a central role in the translational regulation of gene expression through its unwinding activity toward complex RNA structures in messenger RNAs (mRNAs). Although DDX3X is known to selectively stimulate the translation of a subset of genes, a specific sequence motif has not been identified; thus, the molecular mechanism underlying this selectivity remains elusive. Using solution nuclear magnetic resonance (NMR) spectroscopy, we demonstrate that the N-terminal intrinsically disordered region (IDR) of DDX3X plays a critical role in the binding and unwinding of structured RNAs. We propose that the selectivity toward target transcripts is mediated by its preferential binding to structured motifs, particularly the G-quadruplex structure, through arginine-rich segments within the N-terminal IDR. Our results provide a molecular basis for understanding translational regulation by DDX3X and highlight the remarkable role of the flexible IDR in controlling the cellular translational landscape.

2. Results. The influence of N- and C-terminal IDRs on DDX3X function.

a. It would be beneficial to present an SDS-PAGE image of all the proteins used in this study to demonstrate their purity.

We appreciate the reviewer's suggestion. In the revised manuscript, we have added an SDS-PAGE summary of the proteins used in the activity assays as follows.

(Page 7, line 1)

The effect of the N-terminal MBP tag was tested by comparing the activity of Core (residues 132-607) without the N-terminal MBP tag as a control. Protein purity was assessed by sodium dodecyl sulfate-polyacrylamide gel electrophoresis (Supplementary Fig. 2a, b).

(Supplementary Fig. 2b)

Supplementary Figure 2 Characterization of DDX3X proteins. (b) SDS-PAGE analysis of purified proteins.

b. In Fig.1, MBP is a 42.5 kD tag. The authors noted that the addition of the N-terminal MBP tag resulted in a relatively small decrease in activity (~1.8-fold larger apparent K_m in MBP-Core compared to Core), supporting the notion that the MBP tag does not significantly affect the helicase activity of DDX3X. However, this observation actually indicates that MBP attenuated the unwinding activity.

We thank the reviewer for pointing this out. We have revised the text to clearly indicate that the presence of the MBP-tag actually reduces helicase activity. While we acknowledge the effect of the MBP tag, the impact of the N- and C-IDRs can be reliably assessed by comparing among the MBP-tagged proteins, thus we believe this difference does not affect our overall conclusions.

(Page 7, line 22)

The apparent $K_{1/2}$ value of ~30 nM for MBP-FL was consistent with the previous report²⁶. We note that the addition of the N-terminal MBP tag led to a slight decrease in activity (~1.8-fold larger apparent $K_{1/2}$ in MBP-Core compared to Core), likely reflecting the tag's potential influence on ligand binding. While we acknowledge the effect of the MBP tag, the impact of the N- and C-IDRs can be reliably assessed by comparing among the MBP-tagged proteins. These results indicate that the N- and C-IDRs play regulatory roles in enhancing the helicase activity of DDX3X toward dsRNA, presumably by promoting the binding of dsRNA via these regions. Hereafter, we focus on the molecular interaction involving the N-IDR, as it had the greater effect on helicase activity.

Additionally, it is important to consider whether MBP could influence the assembly state of DDX3X.

We appreciate the reviewer's suggestion to address the potential effect of the MBP tag on the assembly state. We have included analytical size-exclusion chromatography results for the MBP-tagged proteins in the supplementary information, confirming that all constructs are predominantly monomeric. Consequently, the activity of each protein construct is compared within a common monomeric background, allowing us to evaluate the effects of the N- and C-IDRs on helicase activity as monomers, rather than through potential differences in oligomerization.

(Page 7, line 4)

We also performed analytical size exclusion chromatography and confirmed that the purified proteins were predominantly monomeric. This allows for a comparison of activity differences attributable to the presence or absence of the N- and C-IDRs in the monomeric state, without the

need to account for potential effects on oligomerization (Supplementary Fig. 2c).

(Supplementary Fig. 2b)

Supplementary Figure 2 Characterization of DDX3X proteins. (a) Domain architecture of the DDX3X variants used in this study. (b) SDS-PAGE analysis of purified proteins. (c) Analytical size-exclusion chromatography of purified proteins. Proteins were separated using a Superdex™ 200 increase column equilibrated with buffer containing 20 mM Tris (pH 8.0), 500 mM NaCl, and 2 mM DTT. For each run, 100 μ L of 20 μ M protein sample was injected and analyzed at a flow rate of 0.5 mL/min. Molecular weight standards included aldolase (158 kDa), conalbumin (75 kDa), ovalbumin (43 kDa), carbonic anhydrase (29 kDa), and ribonuclease A (13.7 kDa), provided by the Gel Filtration Calibration Kits LMW and HMW (Cytiva 28403841 and 28403842).

c. The authors noted that “there was a nearly 6-fold reduction in activity by truncating the N-

IDR and a 53-fold reduction by truncating both the N- and C-IDRs.” The term “truncating the C-IDR” should be corrected to “truncating the N-IDR.” Furthermore, it is unclear how these parameters were obtained. Additionally, the conclusion that the N-IDR is more significant than the C-IDR is not strongly supported by the results, as a direct comparison between truncating the C-IDR alone and truncating the N-IDR alone is lacking.

We thank the reviewer for catching the inconsistency in our description and for pointing out the lack of activity data for the construct lacking only the C-IDR. We have now included helicase activity data for this construct (MBP-N-Core). From the comparison of the half-maximum concentration ($K_{1/2}$) at which the ssRNA fraction reaches 50%, we found that truncation of the C-IDR resulted in an approximately 1.4-fold reduction in activity, truncation of the N-IDR led to a 6-fold reduction, and removal of both the N- and C-IDRs caused a 53-fold reduction in activity. These results clearly indicate that truncation of the N-IDR has a greater impact. We also clarify that the activity was assessed on the basis of the ratio of $K_{1/2}$ values, which were obtained by fitting the DDX3X concentration dependence of the ssRNA fraction assuming a sigmoidal relationship.

(Page 6, line 19)

We compared the helicase activity of constructs with or without the IDRs: MBP-full length (FL) (residues 1-662), MBP-N-Core (residues 1-607), MBP-Core-C (residues 132-662), and MBP-Core (residues 132-607) (Fig. 1b). MBP-Core consists of the minimum functional region of DDX3X²⁶.

(Page 7, line 13)

The assays were conducted with 50 mM added salt according to the previous studies, where low salt conditions were preferred (no added salt to 50 mM) to highlight the helicase activity^{27,37-39}. Relative activity was assessed by comparing the apparent half-maximal concentration, $K_{1/2}$, at which the ssRNA fraction reaches 50%, assuming a sigmoidal relationship. The $K_{1/2}$ values for each protein construct were 33 nM for MBP-FL, 47 nM for MBP-N-Core, 190 nM MBP-Core-C, 1700 nM for MBP-Core, and 970 nM for Core. Consistent with the previous studies^{26,37}, truncation of the N- and/or C-IDR attenuated the helicase activity toward the 36mer/18mer dsRNA, with a 1.4-fold reduction by truncating the C-IDR, a 6-fold reduction by truncating the

N-IDR, and a 53-fold reduction by truncating both the N- and C-IDRs, as determined by the ratio of $K_{1/2}$ values. The apparent $K_{1/2}$ value of ~ 30 nM for MBP-FL was consistent with the previous report²⁶.

Figure 1 Structure of DDX3X and its helicase activity toward dsRNA. (a) The domain architecture of DDX3X. The model structure of full-length DDX3X is shown below (PDB ID: 5E7M)²⁶. (b) dsRNA unwinding assays for IDR-truncated DDX3X variants. The domain architecture of each construct is shown on the left, and representative results of the dsRNA unwinding assays are shown on the right. In the gels, the upper band corresponds to the 18mer/36mer dsRNA, while the lower band corresponds to the 18mer ssRNA displaced from dsRNA by the unwinding activity of DDX3X. Protein concentrations were varied with a 0.6-

fold serial dilution, with the maximum and minimum concentrations indicated at the top of each gel. The reaction mixture was incubated for 30 mins at 37 °C. (c) Plots of ssRNA fractions as a function of DDX3X concentration [M] after 30 mins of incubation, using MBP-FL (circle), MBP-N-Core (square), MBP-Core-C (cross), MBP-Core (triangle), and Core (inverted triangle) proteins. Error bars represent the standard deviation of three independent measurements.

3. Results: Structural characterization of the interaction between the N-IDR and structured RNAs.

a. The results lack quantitative analysis of the binding affinity between the N-IDR and different RNA structures. Methods such as ITC, SPR, FA, or EMSA could be employed to provide more robust and detailed insights.

We appreciate the reviewer's comment regarding the lack of quantitative evaluation of the binding affinity between the N-IDR and various RNA ligands. As an orthogonal approach, we conducted electrophoretic mobility shift assays (EMSA) to monitor the interactions between the N-IDR and these RNA ligands (see Supplementary Fig. 6 below). Consistent with the NMR results, little to no band-shift was observed for poly-U₁₂ ssRNA, GC-14mer dsRNA, and 14mer tetraloop RNA, whereas clear binding was observed for GQ RNAs. The apparent dissociation constants for telomeric and NRAS GQ RNAs were estimated to be $1.0 \pm 0.2 \mu\text{M}$ and $54 \pm 12 \text{ nM}$, respectively. The apparent binding affinity for the NRAS mutant RNA ($K_d = 6.0 \pm 0.8 \mu\text{M}$) was about two orders of magnitude weaker than that for the NRAS GQ RNA, further confirming the GQ-specific recognition by the N-IDR. These EMSA results have been incorporated into the revised manuscript as follows.

(Page 11, line 7)

As an orthogonal test, we conducted electrophoretic mobility shift assays (EMSA)⁵⁰ using fluorescently labeled RNAs to verify the binding preference for GQ RNAs and to obtain rough estimates of the apparent dissociation constant (K_d) values (Supplementary Fig. 6). Consistent with the NMR results, little to no band-shift was observed for poly-U₁₂ ssRNA, GC-14mer dsRNA, and 14mer tetraloop RNA, whereas clear binding was observed for GQ RNAs. The

apparent affinities for telomeric and NRAS GQ RNAs were estimated to be $1.0 \pm 0.2 \mu\text{M}$ and $54 \pm 12 \text{ nM}$, respectively, at 125 mM KCl. Although binding to the NRAS mutant RNA was also observed, its apparent K_d ($6.0 \pm 0.8 \mu\text{M}$) was about two orders of magnitude weaker than that of the NRAS GQ RNA, further confirming the GQ-specific recognition by the N-IDR.

(Supplementary Fig. 6)

Supplementary Figure 6 EMSA analysis of the interaction between the N-IDR and various RNA ligands. EMSA binding experiments were performed for N-IDR using poly-U₁₂-FAM ssRNA, GC-14mer-FAM dsRNA, FAM-14mer tetraloop RNA, FAM-NRAS GQ RNA, FAM-telomeric GQ RNA, and FAM-NRAS mutant RNA. Gel images and fitted binding curves used to estimate K_d values are shown. Unlike in the NMR experiment, poly-U₁₂ was used instead of poly-U₁₀ in EMSA for synthetic reasons. FAM labeling was introduced at the 3'-end for poly-U₁₂ ssRNA and GC-14mer dsRNA, and at the 5'-end for all other RNAs. N-IDR protein was titrated from 0 to 45 μM . The bound fraction was estimated from the intensity of the free RNA probe and fitted to a standard Hill-type equation to obtain apparent K_d values. Free and bound probes were separated using polyacrylamide gels: 8% for all RNAs except the 14mer tetraloop, which was run on a 12% gel. All gels were prepared with 0.5 \times TBE buffer. Error bars represent the standard deviation ($n = 3-4$), with center indicating mean values.

b. The NMR results suggest that the N-IDR shows the strongest binding to RNA G4. However, it remains unclear which substrate DDX3X, as a whole, binds to most effectively. Is the binding preference of DDX3X consistent with that of the N-IDR? Addressing this question would provide a more comprehensive understanding of the relationship between the N-IDR and the helicase core.

We thank the reviewer for raising this important point. To address whether the binding preference of the N-IDR is preserved in the context of the full-length protein, we conducted additional EMSA experiments using full-length DDX3X proteins and NMR experiments on the N-IDR attached to the helicase core.

In EMSA, we monitored the binding between the full-length DDX3X with an N-terminal MBP tag (MBP-FL) and various RNA ligands (see Supplementary Fig. 9 below). Consistent with NMR and EMSA results from the isolated N-IDR, MBP-FL exhibited high-affinity binding to telomeric GQ RNA ($K_d = 54 \pm 14$ nM) and NRAS GQ RNA ($K_d = 43 \pm 4$ nM), while showing little to no band-shift for poly-U₁₂ ssRNA or 14mer tetraloop RNA, and significantly weaker binding to GC-14mer dsRNA ($K_d = 665 \pm 31$ nM). The apparent affinity for NRAS mutant RNA ($K_d = 222 \pm 9$ nM) was about 5-fold weaker than that for NRAS GQ RNA, supporting a structural preference for the GQ moiety in the full-length construct. To confirm that this binding is mediated by the N-IDR, we additionally performed EMSA using an N-IDR-truncated construct (N-terminally MBP-tagged N-IDR-truncated DDX3X, referred to as MBP-Core-C). MBP-Core-C exhibited significantly reduced affinity for both telomeric GQ RNA ($K_d = 579 \pm 69$ nM, ~11-fold reduction) and NRAS GQ RNA ($K_d = 201 \pm 5$ nM, ~5-fold reduction), strongly suggesting that the N-IDR plays a major role in GQ-specific binding even in the full-length context. These results are summarized in the revised manuscript as follows.

Additionally, we conducted NMR experiments observing the N-IDR attached to the helicase core and monitored its binding to RNA ligands. To this end, we prepared [U-¹⁵N] N-Core-MBP, in which an MBP tag was fused to the C-terminus of DDX3X residues 1–607 (N-IDR + Core) to maintain solubility while minimizing structural interference with N-IDR-RNA interactions. In the ¹⁵N-¹H HSQC spectrum of [U-¹⁵N] N-Core-MBP, signals from the N-IDR region could be observed (see

Supplementary Fig. 8 below). The chemical shifts of the observed N-IDR signals were consistent with those in the isolated N-IDR, supporting that the N-IDR retains a flexible conformation when attached to the helicase core. Although only a limited number of NMR probes from the N-IDR were available due to line broadening caused by potential N-IDR-core interactions, we were still able to monitor the N-IDR signals in [U-¹⁵N] N-Core-MBP to evaluate binding to GQ RNA. As in the isolated N-IDR, signal intensity reductions were consistently observed upon the addition of NRAS GQ RNA, indicating that the interaction persists when the N-IDR is part of the larger protein construct. Notably, these reductions in intensity were less pronounced with the NRAS mutant RNA, suggesting that the structural selectivity is preserved even in the full-length background.

These NMR and EMSA results strongly support our conclusion that the N-IDR plays a significant role in GQ-specific recognition in the full-length protein and that the binding preference toward GQ RNA is consistent with the results obtained with the isolated N-IDR. To clarify these points, we have added the following explanations and supplementary figures in the revised manuscript.

(Page 12, line 13)

Before investigating the molecular interaction between the N-IDR and GQ RNAs in detail, we confirmed that the interaction observed with the isolated N-IDR is retained in the full-length protein context. To assess the binding of the N-IDR to GQ RNA when tethered to the folded helicase core, we prepared [U-¹⁵N] N-Core-MBP, in which an MBP tag was fused to the C-terminus of DDX3X residues 1–607 (N-IDR + Core) to maintain solubility while minimizing structural interference with N-IDR-RNA interactions. In the ¹⁵N-¹H HSQC spectrum of [U-¹⁵N] N-Core-MBP, signals from the N-IDR region could be observed due to its inherent structural flexibility, while signals from the helicase core and the MBP-tag were broadened beyond detection (Supplementary Fig. 8a). Notably, the chemical shifts of the observed N-IDR signals were consistent with those in the isolated N-IDR, except for residues in the middle to C-terminal portion (after residue 34), whose signals were severely broadened, similar to those from the core and MBP regions. This suggests that the middle-to-C-terminal portion of the N-IDR forms some interactions with the helicase core, restricting its structural flexibility. Similar interactions have recently been proposed for the yeast counterpart Ded1p as well⁵³. Although it remains challenging to characterize these potential N-IDR-core interactions, they are likely weak and transient in nature. This is supported by a clear difference in elution volume (~0.5 mL) in

analytical size exclusion chromatography between DDX3X constructs with and without the N-IDR, indicating a substantial difference in hydrodynamic radius, which would be expected when the N-IDR retains a flexible conformation rather than being tightly associated with the core (Supplementary Fig. 2c).

Although only a limited number of NMR probes from the N-IDR were available due to line broadening from these potential N-IDR-core interactions, we were still able to monitor the N-IDR signals in [U-¹⁵N] N-Core-MBP to evaluate binding to GQ RNA. As in the isolated N-IDR, signal intensity reductions were consistently observed upon the addition of NRAS GQ RNA, indicating that the interaction persists when the N-IDR is part of the larger protein construct, while retaining its disordered nature upon binding (Supplementary Fig. 8b). Notably, these reductions in intensity were less pronounced with the NRAS mutant RNA, suggesting that the structural selectivity is preserved even in the full-length background (Supplementary Fig. 8c). To further validate the binding preference, we performed EMSA experiments using MBP-FL as an orthogonal test. Consistent with results from the isolated N-IDR, MBP-FL exhibited high-affinity binding to telomeric GQ RNA ($K_d = 54 \pm 14$ nM) and NRAS GQ RNA ($K_d = 43 \pm 4$ nM), while showing little to no band-shift for poly-U₁₂ ssRNA or 14mer tetraloop RNA, and significantly weaker binding to GC-14mer dsRNA ($K_d = 665 \pm 31$ nM) (Supplementary Fig. 9a). The apparent affinity for NRAS mutant RNA ($K_d = 222 \pm 9$ nM) was about 5-fold weaker than that for NRAS GQ RNA, supporting a structural preference for the GQ moiety in the full-length construct. Notably, the N-IDR-truncated construct, MBP-Core-C, exhibited significantly reduced affinity for both telomeric GQ RNA ($K_d = 579 \pm 69$ nM) and NRAS GQ RNA ($K_d = 201 \pm 5$ nM), strongly suggesting that the N-IDR plays a major role in GQ-specific binding (Supplementary Fig. 9b). Collectively, these NMR and EMSA results provide strong evidence that DDX3X preferentially recognizes GQ RNA, and that this selectivity is primarily mediated through N-IDR-RNA interactions.

(Supplementary Fig. 8)

Supplementary Figure 8 NMR characterization of the N-IDR in the full-length protein context. (a) ^{15}N - ^1H HSQC spectrum of $[\text{U}-^{15}\text{N}]$ -labeled N-Core-MBP in the absence of RNA (navy, multiple contours). The spectrum of the isolated $[\text{U}-^{15}\text{N}]$ -labeled N-IDR is overlaid (coral, single contour). (b) 1D slices of the L21, N22, and N26 signals measured in the free state (left), in the presence of 0.33 eq. NRAS GQ RNA (center), or in the presence of 0.33 eq. NRAS mutant RNA (right). (c) Plots of peak height ratios in the presence and absence of 0.33 eq. NRAS GQ RNA (left) or NRAS mutant RNA (right). The average peak height ratio across all residues is indicated by a dashed horizontal line. All NMR measurements were performed at 10 °C and 1 GHz in a buffer containing 20 mM potassium phosphate (pH 7.0), 200 mM KCl, 5 mM DTT, 260 units/mL RNasin® Plus RNAase inhibitor, and 5% D_2O . The protein concentration was 30 μM , and the RNA concentration was 10 μM .

(Supplementary Fig. 9)

Supplementary Figure 9 EMSA analysis of the interaction between MBP-FL/MBP-Core-C and various RNA ligands. EMSA binding experiments were performed for MBP-FL (a) or MBP-Core-C (b) using poly-U₁₂-FAM ssRNA, GC-14mer-FAM dsRNA, FAM-14mer tetraloop RNA, FAM-NRAS GQ RNA, FAM-telomeric GQ RNA, and FAM-NRAS mutant RNA. Gel images and fitted binding curves used to estimate K_d values are shown. MBP-FL protein was titrated from 0 to 1 μ M. The bound fraction was estimated from the intensity of the free RNA probe and fitted to a standard Hill-type equation to obtain apparent K_d values. Free and bound probes were separated using polyacrylamide gels: 8% for all RNAs except the 14mer tetraloop, which was run on a 12% gel. All gels were prepared with 0.5 \times TBE buffer. Error bars represent the standard deviation ($n = 3-5$), with center indicating mean values.

4. Results. Structural basis for the interaction between the N-IDR and GQ RNA.

a. The proposed multivalent binding model of the N-IDR is based solely on signal changes around residues 40, 90, and 120, without additional support from structural models. It is recommended to expand the discussion on the functional roles of these binding sites.

b. Are the RR-1, RR-2, and RR-3 motifs specific to RNA G4 binding, or do they also mediate similar multivalent interactions with other RNA substrates? Further investigation would help clarify their substrate specificity.

We thank the reviewer for the suggestion to expand the discussion on the functional roles of the three arginine-rich segments and the specificity of each RR motif for GQ RNA. To address these points, we have conducted additional NMR and ITC binding experiments using the individual RR segments (RR-1, RR-2, and RR-3) with various RNA ligands.

Notably, for all RR fragment proteins, significant intensity reductions and chemical shift changes were observed only with telomeric and NRAS GQ RNAs, while little to no spectral changes were seen for poly-U₁₀ ssRNA, GC-14mer dsRNA, and NRAS mutant RNA, demonstrating that each RR region independently exhibits GQ-specific binding (see Supplementary Figs. 15-17 below). The intensity reduction profiles were overall similar among the three RR regions, although minor differences were observed in their preferences between telomeric and NRAS GQ RNAs. For example, RR-2 showed comparable intensity reductions upon the addition of 1 equimolar telomeric GQ and 0.5 equimolar NRAS GQ (Supplementary Fig. 16), whereas RR-3 exhibited larger intensity reductions for telomeric GQ than for NRAS GQ (Supplementary Fig. 17). These subtle differences may reflect distinct structural preferences or binding modes toward GQ RNAs with different topologies; however, we did not observe clear functional differences among these three RR regions, at least in the context of the N-IDR's GQ-specific recognition. The direct binding was further confirmed by ITC analyses, which yielded apparent K_d values ranging from 2 to 11 μ M, comparable to those of the full N-IDR ($K_d \sim 1-3 \mu$ M) (Supplementary Fig. 18).

These results demonstrate that all RR motifs have comparable affinities and exhibit similar GQ-specific binding, while small differences in binding preference toward telomeric or NRAS GQ were

observed among the motifs. Based on these results, we conclude that the functional differences among the three RR motifs are minimal, at least in the context of the GQ-specific binding, and that the structural coupling between the three RR clusters is relatively weak.

To clarify these points, we have added the following section discussing the binding of each RR motif, along with supplementary figures in the revised manuscript.

(Page 19, line 20)

To further investigate the independent binding activity and individual contributions of each RR cluster to GQ RNA recognition, we employed an alternative approach by monitoring the interaction using isolated RR fragments. To this end, we prepared three RR fragment proteins, RR-1 (residues 31–61), RR-2 (residues 73–102), and RR-3 (residues 106–127), and performed NMR binding experiments with various RNA ligands (Supplementary Figs. 15, 16, and 17). Most of the signal assignments for each RR region could be readily transferred from those of the full N-IDR, indicating that the disordered nature of the N-IDR is preserved in each isolated RR construct. Notably, for all RR fragment proteins, significant intensity reductions and chemical shift changes were observed only with telomeric and NRAS GQ RNAs, while little to no spectral changes were seen for poly-U₁₀ ssRNA, GC-14mer dsRNA, and NRAS mutant RNA, demonstrating that each RR region independently exhibits GQ-specific binding. The intensity reduction profiles were overall similar among three RR regions, although minor differences were observed in their preferences between telomeric and NRAS GQ RNAs. For example, RR-2 showed comparable intensity reductions upon the addition of 1 equimolar telomeric GQ and 0.5 equimolar NRAS GQ (Supplementary Fig. 16), whereas RR-3 exhibited larger intensity reductions for telomeric GQ than for NRAS GQ (Supplementary Fig. 17). These subtle differences may reflect distinct structural preferences or binding modes toward GQ RNAs with different topologies⁶¹; however, we did not observe clear functional differences among these three RR regions, at least in the context of the N-IDR's GQ-specific recognition. Direct binding of RR-1 and RR-2 to GQ RNAs was further confirmed by ITC analyses, which yielded apparent K_d values ranging from 2 to 11 μ M, comparable to those of the full N-IDR ($K_d \sim 1$ -3 μ M) (Supplementary Fig. 18). These results further support the notion that there is no strong structural coupling or binding cooperativity among the three RR clusters. In the NMR binding experiment with telomeric GQ RNA, we observed slightly larger intensity reductions around residue 40 of

the N-IDR (Fig. 3c). However, we did not detect stronger affinity for the RR-1 fragment protein that contains this region. This suggests that the observed larger intensity reductions are more likely due to restricted motion involving the characteristic successive proline residues (Pro40-Pro41) in the bound state, rather than the preferential binding of GQ RNA to this region.

(Supplementary Figs. 15-18)

Supplementary Figure 15 NMR characterization of the interaction between the RR-1 fragment and various RNA molecules. (a) NMR spectra and peak height ratios of RR-1 signals obtained with and without each RNA molecule. For each sub-panel, the overlay of ¹⁵N-¹H HSQC spectra of the [U-¹⁵N]-labeled RR-1 in the absence (navy) and presence (orange-red) of RNA, and the plot of peak height ratios are shown. The ratio was calculated by dividing the peak height in the presence of RNA by that in the absence of RNA. Residues that were not analyzed are indicated by gray backgrounds. All NMR measurements were performed at 10 °C and 1 GHz in a buffer containing 20 mM potassium phosphate (pH 7.0), 50 mM KCl, 5 mM DTT, 260 units/mL RNasin® Plus RNAase inhibitor, and 5% D₂O. The protein concentration was 50 μM, and the RNA concentration was 50 μM (for poly-U10 ssRNA, GC-14mer dsRNA, and telomeric GQ RNA) or 25 μM (for NRAS GQ RNA and NRAS GQ mutant RNA). The 1D slices of the

labeled signals are shown in each spectrum. (b) Full ^{15}N - ^1H spectrum of $[\text{U}-^{15}\text{N}]$ -labeled RR-1 with signal assignments.

Supplementary Figure 16 NMR characterization of the interaction between the RR-2 fragment and various RNA molecules. The experimental details and plots are the same as in Supplementary Fig. 15.

Supplementary Figure 17 NMR characterization of the interaction between the RR-3 fragment and various RNA molecules. The experimental details and plots are the same as in Supplementary Fig. 15.

Supplementary Figure 18 ITC experiments for RR-1/RR-2 and GQ RNA interactions. (a) ITC isotherm (top) and integrated heats (bottom) for the titration of telomeric GQ RNA (250 μM as a dimer) into solutions of the RR-1 (left) or RR-2 (right) fragment protein (100 μM). (b) ITC isotherm (top) and integrated heats (bottom) for the titration of NRAS GQ RNA (300 μM as a single strand) into solutions of the RR-1 (left) or RR-2 (right) fragment protein (50 μM). The titration data were analyzed using the standard binding model assuming one set of sites, and the fitted parameters were displayed in the insets. All measurements were performed at 25 $^\circ\text{C}$. Experiments on RR-3 were not included due to poor reproducibility, likely due to a tendency to form small amounts of aggregates.

c. The authors stated: Although exchange broadening effects complicate precise estimation of the bound fraction, this result aligns reasonably well with the expected 24-fold decrease in binding affinity (e.g., a decrease in the bound fraction from ~64% to ~15% with a K_d increase from 10 μM to 240 μM at protein and RNA concentrations of 50 μM at 10 °C). However, it is unclear how these K_d values were determined. Providing additional details on the methodology or calculations used to derive these values would enhance the clarity and reproducibility of the results.

We thank the reviewer for pointing out the lack of details about the estimation of the bound fraction. In the previous version of our manuscript, we used the intensity reduction ratio of the NMR signal as an estimate for the bound population, which is a crude assumption given the complexity of the binding kinetics and the heterogeneity of the bound state. In the revised manuscript, we instead use the apparent K_d value for the N-IDR-telomeric GQ interaction obtained from EMSA and ITC analyses (K_d range of 1-10 μM). Then, we estimate the bound fractions, using a simple one-site binding model and assuming the N-IDR and RNA concentrations of 50 μM . We believe that the details of the methodology and calculations are now described with reasonable clarity.

(Page 16, line 21)

The salt concentration dependence of electrostatic interactions between negatively charged RNA and positively charged peptides has been extensively characterized. For example, the interaction between a 19-residue box B RNA hairpin and a 22-residue arginine-rich peptide from the N protein of phage λ (containing 5 arginine and 2 lysine residues, theoretical pI of 11.44) exhibited strong dependence on KCl concentration. This system, comparable in length and size to the interaction between the GQ RNA and the arginine-rich segments of the N-IDR studied here, showed that the electrostatic contribution to the binding free energy ($\Delta G^\circ_{\text{dock}}$) varied significantly with KCl concentration, with $\partial(\Delta G^\circ_{\text{dock}})/\partial \log[\text{KCl}]$ of ~ 6 kcal/mol⁵⁹. This corresponds to a ~ 24 -fold decrease in binding affinity when the KCl concentration is doubled at 10 °C. **Assuming a K_d range of 1-10 μM as estimated from the EMSA and ITC experiments with telomeric GQ RNA, this ~ 24 -fold increase in K_d (i.e. a shifted K_d of 24-240 μM) results in a change in bound fraction**

from ~64-87% to ~15-51%, assuming 50 μM concentrations for both protein and RNA and a simple one-site binding model. Based on this, we monitored GQ RNA binding at 400 mM KCl and compared the results with those at 200 mM KCl (Fig. 3d, Supplementary Fig. 10c). At 400 mM KCl, intensity reductions were ~40-60% less pronounced than those observed at 200 mM KCl. Although it is difficult to estimate the exact bound fraction from intensity reductions due to the heterogeneity of the bound state, this result aligns reasonably well with the expected decrease in binding as noted above.

5. Results: Binding of the N-IDR to GQ segments in physiological mRNA substrates. While NMR is valuable for characterizing protein-substrate interactions, are there additional methods that could be employed to further validate these findings? For example, quantifying the binding affinities by comparing dissociation constants (K_d) across different substrate structures and various G4s using techniques such as ITC, SPR, or EMSA would provide a more comprehensive understanding of the binding properties.

We appreciate the reviewer's suggestion to employ additional methods to demonstrate the binding of the N-IDR to GQ segments in mRNA substrates. To this end, we conducted EMSA to estimate the dissociation constants for the mRNA substrates (see Supplementary Fig. 20 below). Stronger binding was observed for the GQ-forming fragments, *RAC1* 53–77 ($K_d = 0.51 \pm 0.08 \mu\text{M}$) and *ODC1* 150–174 ($K_d = 1.97 \pm 0.23 \mu\text{M}$), compared to the non-GQ fragments ($K_d > 10 \mu\text{M}$). We note one exception: *ODC1* 279–303 exhibited stronger binding in the NMR experiments but behaved similarly to the non-GQ fragments in EMSA. We speculate that this discrepancy most likely arises from the higher sensitivity of NMR for detecting relatively weak interactions with K_d values in the micromolar range. Alternatively, the faster association-dissociation kinetics of this ligand may have hindered the resolution of distinct protein-RNA complexes in EMSA. Given that EMSA is typically employed for detecting protein-nucleic acid interactions with K_d values in the nM range, we believe that the NMR results more accurately capture the binding preferences in this case and that this discrepancy does not affect our overall conclusion.

In the revised manuscript, we have added the EMSA results for the different mRNA substrates as follows.

(Page23, line 8)

The binding of these mRNA fragments was also validated by EMSA (Supplementary Fig. 20), which showed stronger binding for the GQ-forming fragments, *RAC1* 53–77 ($K_d = 0.51 \pm 0.08 \mu\text{M}$) and *ODC1* 150–174 ($K_d = 1.97 \pm 0.23 \mu\text{M}$), compared to the non-GQ fragments ($K_d > 10 \mu\text{M}$). One exception was *ODC1* 279–303, which exhibited binding in the NMR experiments but behaved similarly to the non-GQ fragments in EMSA. We speculate that this discrepancy most likely arises from the higher sensitivity of NMR for detecting relatively weak interactions with K_d values in the micromolar range, whereas the faster association-dissociation kinetics may have hindered the resolution of distinct protein-RNA complex in EMSA, especially given that the affinity difference between *ODC1* 279-303 and the non-GQ fragment *ODC1* 449-473 appears to be rather modest.

(Supplementary Fig. 20)

Supplementary Figure 20 EMSA analysis of the interaction between the N-IDR and mRNA fragments. EMSA binding experiments were performed using the N-IDR and the following FAM-labeled mRNA fragments: *RAC1* 53–77, *RAC1* 129–153, *ODC1* 150–174, *ODC1* 279–303, and *ODC1* 449–473. Gel images and fitted binding curves used to estimate K_d values are shown. FAM labeling was introduced at the 5'-end for all mRNA fragments. N-IDR protein was titrated from 0 to 45 μM . The bound fraction was estimated from the intensity of the free RNA probe and fitted to a standard Hill-type equation to obtain apparent K_d values. Samples were separated using 8% polyacrylamide gels prepared with $0.5\times$ TBE buffer. Error bars represent the standard deviation ($n = 3-4$), with center indicating mean values.

6. Results: Characterization of the GQ-unfolding activity of DDX3X.

a. The characterization of RNA G4 unwinding is relatively simplistic and could be improved. Incorporating additional methods commonly used for G4 unwinding studies, such as gel electrophoresis, could provide more robust and detailed insights.

We thank the reviewer's suggestion to incorporate an additional method commonly used for measuring GQ unfolding activity. In response, we have included a gel electrophoresis-based assay to monitor the GQ unfolding activity, as described by Gao *et al.* (*Chem. Commun.*, 55, 4467-4470 (2019)). Briefly, this assay monitors the accelerated hybridization of NRAS GQ to a complementary RNA strand as a result of GQ unfolding by DDX proteins (see Supplementary Fig. 22a below). In our preliminary experiments, we found that it was technically difficult to kinetically trap the reaction intermediates by adding the capture RNA/DNA strand, which hindered a quantitative evaluation of the reaction kinetics as performed in the study by Gao *et al.* Nevertheless, we were able to detect an increase in the dsRNA fraction in the presence of both DDX3X and the complementary RNA strand at a fixed time point, clearly indicating GQ-unfolding activity of DDX3X toward NRAS GQ.

In the revised manuscript, we have added the results of the GQ-unfolding assay monitored by hybridization to a complementary strand as follows.

(Page 25, line 17)

Supporting the results of the fluorescence quenching assay, the GQ-unfolding activity of MBP-FL was further confirmed by a GQ hybridization assay using a complementary RNA strand⁶⁸ (Supplementary Fig. 22).

(Supplementary Fig. 22)

Supplementary Figure 22 GQ-unfolding assay monitored by hybridization to a complementary RNA strand. (a) Schematic representation of the NRAS GQ-unfolding assay, in which GQ-unfolding is detected by hybridization to a complementary RNA strand. (b) Native gel analysis of reaction mixtures containing 50 nM FAM-labeled NRAS GQ RNA, with or without 100 nM complementary RNA, and with or without 100 nM DDX3X (MBP-FL). The two rightmost lanes show samples incubated at 95 °C for 3 minutes prior to electrophoresis. Samples were separated using 20% polyacrylamide gels prepared with Tris-Glycine buffer. (c) Plots of the intensity ratio between the duplex and NRAS GQ bands for lanes containing 100 nM complementary strand. Error bars represent the standard deviation ($n = 4$), with center indicating mean values.

b. Quantitative analysis is lacking. For instance, what is the unwinding ratio? Additionally, analyzing the unwinding kinetics over time would provide a more comprehensive understanding of the process.

We thank the reviewer for pointing out the lack of quantitative interpretation of the GQ-unfolding

assay monitored by fluorescence quenching. We did not include the unwinding ratio in our results because it was technically difficult to estimate reliably in our system due to the high stability of NRAS GQ.

Wu *et al.* originally reported the fluorescence-based GQ-unfolding assay using GQ DNA dually labeled with FAM and BHQ1 and DDX5 proteins. In their study, the unwinding ratio was estimated by comparing fluorescence intensity in the presence and absence of KCl, exploiting the fact that DNA GQ structures are almost completely unwound under KCl-free conditions. When an excess amount of DDX5 was added to the DNA GQ solution in the presence of 50 mM KCl, the fluorescence increased to the same level as observed in the 0 mM KCl condition, indicating that DDX5 nearly completely unwound the DNA GQ (see Figure 1 adapted from Wu *et al.*).

[REDACTED]

Adapted from Figure 1 in Wu *et al.*, *PNAS*, 116, 41, 20453-20461 (2019) DDX5 actively unfolds MycG4 and is a potent G4 unfolding protein. [REDACTED]

We applied the same strategy to estimate the maximum fluorescence difference of the dually labeled NRAS GQ (Supplementary Fig. R1). In the absence of KCl, the fluorescence intensity was higher than in the presence of 50 mM KCl, consistent with the idea that the unfolded (elongated) conformation of NRAS GQ is more populated under KCl-free conditions. However, when 40 nM MBP-FL was added in the KCl-free buffer, we observed a further increase in fluorescence intensity, suggesting that NRAS GQ is not fully unfolded even without KCl. This indicates that the fluorescence intensity at 0 mM KCl does not represent a fully unfolded state of NRAS GQ, making it difficult to estimate the unwound fraction under specific conditions, as the reference value for the fully unwound state is not available. The high stability of NRAS GQ even in the absence of stabilizing cations has been previously reported by Kumari *et al.* (*Nat. Chem. Biol.* 3 (4) 218-221, (2007)), where the apparent T_m of NRAS GQ RNA was estimated to be ~ 43 °C even without added cation.

Supplementary Figure R1 Fluorescence emission spectra of FAM/BHQ1-labeled NRAS GQ recorded without (gray line) and with (magenta line) 40 nM MBP-FL in the presence (left) or absence (right) of KCl. The buffer condition was 20 nM FAM-NRAS GQ-BHQ1, 20 mM Tris-HCl (pH 7.6), and 0 or 50 mM KCl. The excitation wavelength was set to 490 nm. The measurements were performed at 25°C.

Still, if we assume that the fluorescence intensity observed at 40 nM MBP-FL in 0 mM KCl represents the maximum value corresponding to fully unfolded NRAS GQ, we can estimate the unwound fraction to be around ~35% at 40 nM MBP-FL in 50 mM KCl. Although we expect the actual unwound fraction to be close to this value, we chose not to include it in our results due to uncertainty in the validity of this assumption. Importantly, even without estimating the absolute unwound fraction, the relative differences in GQ-unfolding activity among MBP-FL, MBP-Core-C, and N-IDR can be reliably assessed by comparing the relative increases in fluorescence in the presence and absence of each protein. Thus, the lack of an absolute unwound ratio does not affect the interpretation of our results.

Regarding the unfolding kinetics, we monitored the time course of the fluorescence signal of NRAS GQ upon addition of MBP-FL (see Supplementary Fig. 21 below). However, the unfolding occurred faster than the time resolution of our instrument. The fluorescence increase was completed within the ~14 sec dead time following MBP-FL addition, making it difficult to determine the unwinding rate. The observed rapid unfolding, with a half time of a few seconds, is consistent with previous results reported by Wu *et al.* (*PNAS*, 116 (41), 20453–20461 (2019)).

We have added the corresponding text and a supplementary figure to the revised manuscript.

(Page 25, line 14)

Time-course analysis showed that MBP-FL unfolded the NRAS GQ structure almost instantaneously, within the ~14-second dead time following MBP-FL addition (Supplementary Fig. 21b), which is consistent with previous results for DDX5, which rapidly unfolded GQ DNA with a half-time of a few seconds⁷⁰. Supporting the results of the fluorescence quenching assay, the GQ-unfolding activity of MBP-FL was also confirmed via an accelerated hybridization assay using a complementary RNA strand⁶⁸ (Supplementary Fig. 22).

(Page 41, line 18)

We did not attempt to convert the increase in fluorescence into the unwound ratio of NRAS GQ as previously done⁷⁰, because NRAS GQ remains stably folded even in the absence of KCl⁴⁷, making it difficult to reliably measure the maximum fluorescence intensity corresponding to complete unfolding.

Supplementary Figure 21 GQ-unfolding activity measured via fluorescence quenching. (b) Time-course analysis of GQ-unfolding upon addition of 40 nM MBP-FL to FAM/BHQ1-labeled NRAS GQ. At the time point indicated by the arrow, 100 μ L of MBP-FL solution was rapidly added to a 3 mL sample containing 20 nM FAM/BHQ1-labeled NRAS GQ. The gray background indicates the dead time during MBP-FL addition. The excitation and emission wavelengths were set to 490 nm and 520 nm, respectively.

c. There appears to be a discrepancy between the fluorescence assay in Fig. 5D, which shows no unwinding activity in the core, and the NMR assay in Fig. 5F, which suggests that RNA G4 is converted to ssRNA by the core. This potential contradiction should be clarified to ensure consistency and accuracy in the interpretation of the results.

We thank the reviewer for pointing out the apparent discrepancy between the fluorescence quenching assay in Fig. 5d and the NMR assay in Fig. 5f. Our interpretation is that the helicase core region, even in the absence of IDRs, does possess GQ-unfolding activity; however, this activity was not detected in the fluorescence quenching experiment with MBP-Core-C (N-IDR-truncated DDX3X) simply

because the protein concentration used was too low. Note that, in our previous version, the fluorescence quenching assays used 40-160 nM MBP-Core-C and 20 nM fluorescently labeled NRAS GQ, whereas the NMR experiments used 50 μ M core protein and 300 μ M NRAS GQ to promote binding.

To more directly test the helicase activity of MBP-Core-C and the N-IDR, we repeated the fluorescence quenching experiments using a higher concentration range, up to 340 nM. At concentrations above 200 nM, MBP-Core-C exhibited clear GQ-unfolding activity toward NRAS GQ, although the activity was significantly weaker than that of the full-length protein (MBP-FL) (see revised Fig. 5 and Supplementary Fig. 21 below). In contrast, the N-IDR alone showed little to no unfolding activity at the same concentration, demonstrating that the Core-C region, rather than the N-IDR, is responsible for GQ-unfolding activity, and that the presence of the N-IDR significantly enhances this activity. The ability of the core region to unfold GQ is fully consistent with the NMR results shown in Fig. 5f and 5g.

Combined with the EMSA results showing that the N-IDR significantly increases the binding affinity for GQ RNA substrates (see our response to point 3b above), these findings indicate that the robust GQ-unfolding activity of MBP-FL arises from the synergistic actions of the N-IDR-mediated RNA binding and the helicase activity of the folded core.

To clarify these points, we have revised Fig. 5d and included the additional data in Supplementary Fig. 21 as follows.

(Page 24, line 19)

We then measured the fluorescence of the modified NRAS GQ (20 nM) in the presence of varying concentrations of MBP-FL, MBP-Core-C, and N-IDR proteins (Fig. 5d). Upon the addition of MBP-FL, we observed an increase in fluorescence, with the signal nearly reaching a plateau at 2 equimolar concentrations (40 nM) of MBP-FL, suggesting that DDX3X can unwind the NRAS GQ structure. The apparent binding affinity of MBP-FL toward the dually labeled NRAS GQ was estimated to be on the order of 10 nM, consistent with the apparent K_d for MBP-FL-NRAS GQ interaction ($K_d = 43 \pm 4$ nM) obtained from EMSA (Supplementary Fig. 9). In contrast, almost no increase in fluorescence was observed upon the addition of the same

concentration of MBP-Core-C or N-IDR (Fig. 5d). However, at a much higher concentration (~340 nM), MBP-Core-C showed a smaller but significant increase in fluorescence, while the N-IDR alone produced little to no effect (Fig. 5d, Supplementary Fig. 21a). These results indicate that the Core-C region, rather than the N-IDR, is responsible for the GQ-unfolding activity, and that this activity is significantly enhanced in the presence of N-IDR. Combined with EMSA results showing that the N-IDR significantly increases binding affinity to GQ RNA substrates (Supplementary Fig. 9), these findings indicate that the robust GQ-unfolding activity of MBP-FL arises from the synergistic actions of the N-IDR-mediated binding and the helicase activity of the folded core.

(Figure 5)

Figure 5 GQ-unfolding activity of DDX3X. (d) Fluorescence emission spectra of FAM/BHQ1-labeled NRAS GQ recorded without (solid black line) and with (dotted magenta line) the MBP-FL (left), MBP-Core-C (middle), and N-IDR (right) proteins. The intensity ratios of the maximum fluorescence intensities of FAM/BHQ1-labeled NRAS GQ at each protein concentration are plotted below. Error bars represent the standard deviation of three independent measurements.

(Supplementary Fig. 21)

Supplementary Figure 21 GQ-unfolding activity measured via fluorescence quenching. (a) Representative fluorescence emission spectra of FAM/BHQ1-labeled NRAS GQ recorded in the absence (gray line) and presence (colored lines) of varying concentrations of MBP-FL (left), MBP-Core-C (middle), and N-IDR (right) proteins. Protein concentrations ranged from 0 to 40 nM for MBP-FL and from 0 to 340 nM for MBP-Core-C and N-IDR. The excitation wavelength was set to 490 nm.

7. Results: GQ propensity of DDX3X's target transcripts. The authors stated that the high content and stability of G-quadruplex (GQ) structures, particularly in the 5'-UTR, are key factors that determine whether a gene is translationally regulated by DDX3X. However, there is a lack of direct evidence within cells to support this claim. It remains unclear whether the regulation of these genes is truly mediated through RNA G-quadruplexes.

We thank the reviewer for pointing out the lack of direct evidence within cells. We acknowledge that our results are mainly derived from *in vitro* experiments, and the bioinformatic analysis of GQ propensity in physiological mRNA substrates of DDX3X serves only as indirect support for physiological relevance. While it would be ideal to perform direct cellular imaging of mRNA substrates using GQ-specific probes, such analyses remain extremely challenging, especially considering the complexity of cellular mRNA distributions/compositions and the technical difficulty of specifically visualizing DDX3X target transcripts, which we believe is beyond the scope of the

current study.

Nevertheless, we have provided some supporting evidence that the observed interaction between the N-IDR and GQ structures is physiologically relevant. Notably, the binding segment we identified in the *RAC1* transcript is fully consistent with prior cellular data by Wilkins *et al.* as discussed in our manuscript. In this study, translational activity assays showed that deletion of nucleotides 40-78 in *RAC1* abolished DDX3X-dependent translational regulation. This region overlaps with the GQ-forming segment we identified (nucleotides 53-77), which exhibited the highest binding affinity to the N-IDR in our assays. This concordance between *in vitro* and cellular data strongly supports the physiological relevance of the N-IDR–GQ interaction.

(Page 31, line 20)

Our NMR binding analyses using the 5'-UTR RNA fragments from *RAC1* and *ODC1* transcripts demonstrated that the N-IDR binds to the GQ-forming segments in these physiological substrates. We acknowledge that these analyses used relatively short RNA fragments (25 nt), which may not fully reflect the dominant structural features of full-length mRNAs; however, our NMR results are highly consistent with the translational activity assays of *RAC1* and *ODC1* deletion mutants reported by Wilkins *et al.*¹². The region of the *RAC1* fragment with the highest affinity toward the N-IDR, nucleotides 53-77, overlaps with the region where a deletion mutation (nucleotides 40-78) abolished DDX3X's translational regulation. Similarly, in the analyses by Wilkins *et al.*¹², the translational repression of *ODC1* was not abolished by a single deletion mutant, which can be explained by our finding that *ODC1* has two major GQ-forming segments, nucleotides 150-174 and 279-303, both of which showed comparable affinity for the N-IDR.

To further support the physiological relevance of our findings, we included an additional DDX3X target transcript in the revised manuscript. The *MITF* transcript has been extensively characterized in a previous study by Phung and Cieřla *et al.* (*Cell Rep.* **27**, 3573-3586.e7 (2019)), which demonstrated that DDX3X tightly regulates the translation of *MITF* through an internal ribosome entry site (IRES) element located within its 5'-UTR. The authors showed that DDX3X function affects protein synthesis and melanoma phenotype through the IRES-mediated translational regulation, and they clearly identified the region of the *MITF* IRES responsible for this regulation. This study provides compelling evidence, both *in cells* and *in vivo*, that the translational efficiency of this transcript is

highly dependent on DDX3X-IRES interactions.

We conducted a GQ-propensity analysis of the *MITF* 5'-UTR and identified a GQ-forming segment (*MITF* 62–86). NMR binding experiments were performed using this GQ-forming RNA fragment and a negative control fragment (*MITF* 7–31) (see Supplementary Fig. 24a below). The formation of a GQ structure in *MITF* 62–86 was confirmed by ¹H NMR (Supplementary Fig. 24b). Notably, a marked reduction in N-IDR signal intensity was observed upon the addition of the GQ-forming *MITF* 62–86 fragment, whereas no such change was seen with the control *MITF* 7–31 fragment, suggesting that the GQ segment mediates DDX3X recognition (Supplementary Fig. 24c). Importantly, the identified GQ-forming region (*MITF* 62–86) overlaps with the previously reported DDX3X-binding SL3B motif (*MITF* 68–93), which constitutes a part of the *MITF* IRES. This strongly supports the physiological significance of GQ recognition by DDX3X and suggests that such selective recognition plays a key role in diverse modes of translational regulation.

We believe this additional evidence further underscores the physiological relevance of our findings, and that the GQ recognition by the N-IDR of DDX3X contributes, at least in part, to its functional roles in cells. These results have been included in the revised manuscript.

(Page 29, line 5)

GQ-recognition in IRES-mediated translational regulation

Although we have so far focused on the target specificity of DDX3X in the context of canonical cap-dependent translational regulation, it is reasonable to expect that such GQ-mediated target recognition by DDX3X also plays a role in other modes of translational control. To further strengthen the physiological relevance of GQ-mediated DDX3X recognition, we included the transcript of the microphthalmia-associated transcription factor (*MITF*) gene in our analysis. Phung and Cieřla *et al.*²² previously proposed that DDX3X regulates non-canonical translation of *MITF* via an internal ribosome entry site (IRES) located within its 5'-UTR, which is closely linked to melanoma progression through its effects on cell proliferation and migration. The direct binding of DDX3X to the 5'-UTR of the *MITF* transcript and its functional consequences on cell proliferation have been well characterized.

Given that the 5'-UTR of the *MITF* transcript contains a predicted GQ-forming sequence around nucleotide 70, as identified by rG4 detector (Supplementary Fig. 24a), we monitored N-

IDR binding to this GQ-forming segment (*MITF* 62–86) as well as to a negative control fragment (*MITF* 7–31). The GQ structure of the *MITF* 62-86 fragment was confirmed by ^1H NMR analysis (Supplementary Fig. 24b). Notably, a marked intensity reduction of the N-IDR signals was observed upon addition of the GQ-forming *MITF* 62-86 fragment, while no such change was seen with the *MITF* 7–31 fragment, suggesting that the GQ segment in *MITF* is responsible for DDX3X recognition (Supplementary Fig. 24c). Importantly, the identified GQ-forming region (*MITF* 62–86) overlaps with the DDX3X-binding SL3B motif (*MITF* 68–93), which constitutes part of the IRES element in the *MITF* transcript. Our results, together with extensive cellular evidence reported by Phung and Cieřła *et al.*²², strongly support the physiological significance of GQ-recognition by DDX3X, and such selective recognition plays a key role in various modes of translational regulation including both canonical cap-dependent and non-canonical IRES-mediated pathways.

(Supplementary Fig. 24)

Figure 24 GQ propensity of *MITF* and its interaction with the N-IDR. (a) Plots of the prediction scores for GQ propensity of the *MITF* 5'-UTR sequence obtained using the rG4 detector software⁶. In the inset, the sequences of the segment with high prediction scores (*MITF* 62–86, purple) and the negative control (*MITF* 7–31, gold) are highlighted. (b) The imino ¹H 1D NMR spectra of the *MITF* 7–31 (top) and *MITF* 62–86 fragments. (c) NMR spectra and peak height ratios of the N-IDR signals obtained with and without each RNA fragment in the presence of 200 mM KCl. In the top row, overlays of ¹⁵N-¹H HSQC spectra of the [U-¹⁵N]-labeled N-IDR in the absence (navy) and presence (orange-red) of RNA are shown. The 1D slices of the labeled signals are shown in each spectrum. In the bottom row, plots of peak height ratios are shown. The ratio was calculated by dividing the peak height in the presence of 0.5 equimolar RNA by that in the absence of RNA. Residues that were not analyzed are indicated with gray backgrounds. All NMR measurements were performed at 10°C and 1 GHz, and protein and RNA concentrations were 50 μM and 25 μM, respectively.

8. Discussion. It would be beneficial to discuss the differences between this work and the research by Herdy et al. [Reference 36].

We appreciate the reviewer's suggestion. We acknowledge that our study builds upon the study by Herdy *et al.* (Nucleic Acids Res. 46, 11592–11604 (2018)), which identified DDX3X as a binder of NRAS GQ using an unbiased affinity proteomics approach. While this study discussed the potential role of GQ structures in DDX3X-mediated mRNA recruitment, it did not address the molecular mechanisms underlying this recognition or its broader physiological implications. In our study, we investigated the molecular basis underlying this selectivity and discussed its functional relevance in the context of selective translational regulation of gene expression by DDX3X, showing that molecular interactions between the flexible IDR and folded GQ RNA play a central role in this process.

To clarify our contributions regarding the molecular mechanism and its functional significance, we have added a brief explanation in the discussion section as follows.

(Page 30, line 11)

The DDX RNA helicases are an important class of proteins that regulate gene expression at the translational level, participating in various physiological processes. While numerous studies have identified genes translationally regulated by DDX helicases, the molecular mechanism by which these helicases selectively recognize specific transcripts has largely remained unclear. Building on the study by Herdy *et al.*, which identified DDX3X as a physiological binder of NRAS GQ^{36,73}, we set out to explore the potential role of the GQ binding in the selective recruitment of mRNA substrates and to investigate the structural basis underlying this process, focusing primarily on the role of the N-IDR in substrate RNA binding and its influence on helicase activity. We demonstrated that the N-IDR, despite its lack of a well-defined structure, can selectively recognize the GQ structure of RNA. This selective interaction is primarily mediated by arginine- and aromatic-rich segments of the N-IDR, which preferentially bind to the characteristic G-tetrad structure through a combination of electrostatic and multivalent π -interactions. The physiological importance of this interaction is further supported by findings that the N-IDR binds to the GQ-forming segment of the 5'-UTR in DDX3X targets, *RAC1*, *ODC1*, and *MITF1*, and that the putative GQ-forming sequences are more highly enriched in the 5'-UTRs of the mRNA transcripts translationally regulated by DDX3X. Our results highlight the key functional role of the GQ structure as a marker for the selective translational regulation by DDX3X in both canonical cap-dependent and non-canonical IRES-mediated mechanisms, wherein the two synergistic functions of DDX3X, the specific binding of the N-IDR to GQ-containing segments and the subsequent unfolding of the GQ structure by the helicase core, are at play at the molecular level (Fig. 6b).

Reviewer #2:

This outstanding manuscript describes a biophysical and cellular study of the mechanism of DDX3X on selective RNA recognition. Using NMR, EMSA, ITC, bioinformatics, and biochemical unwinding activity assays to unravel the molecular mode of interaction of the protein DDX3X that is an essential protein. The authors show that the intrinsically disordered N- and C-terminal domains of the protein bind preferentially to RNA G quadruplexes, while they do not recognize single stranded, double stranded or stem-loop RNA structures. The interaction is entropically and (!) enthalpically driven. The authors show that this binding is localized to RG-repeat structures of in the IDRs of DDX3X. The enthalpic part stems from Arginine cation-Pi-interaction with the top and bottom tetrads of the RNA G4s. This is an outstanding manuscript that I strongly recommend to be published.

We appreciate the reviewer's positive and valuable feedback on our manuscript. In response to the reviewer's comments, we have revised the manuscript to include several additional points of discussion. Our responses are summarized below.

There is a single point that I would recommend the authors to consider. It would be interesting to see whether DDX3X can differentiate between RNA and DNA G4. Maybe, DDX3X is strictly cytosolic, and thus this might be irrelevant due to the non-nuclear localisation. But it is probably worthwhile to at least mention this.

We thank the reviewer for the insightful comment. To examine whether the N-IDR can differentiate between RNA and DNA, we compared its interactions with the RNA and DNA versions of NRAS GQ (see Supplementary Fig. 5 below). The intensity reduction patterns observed in the NMR spectra were highly similar for both, suggesting that the N-IDR binds to RNA and DNA in a comparable manner.

We have included this comparison of RNA and DNA recognition in the revised manuscript as follows.

(Page 10, line 16)

As a control, we also analyzed the interaction with an NRAS mutant (5'-UGUAGAAAGAGCAGAUCUAGAUGC-3', where base substitutions were introduced at the underlined G-tetrad positions). In the NRAS mutant, GQ formation is abolished by multiple guanosine-to-adenosine substitutions to G-tetrads³⁶. The intensity reduction was largely suppressed in the NRAS mutant, further validating that the N-IDR specifically recognizes the GQ structure. We note that very similar structural preferences were observed in a set of NMR experiments conducted under more physiologically relevant salt conditions (120 mM KCl) (Supplementary Fig. 4). These results demonstrate that the N-IDR preferentially recognizes and binds to the GQ structure of RNA, while it does not strongly interact with ssRNA, dsRNA, or hairpin structures. **We note that, although the physiological substrate of the N-IDR is expected to be RNA, very similar binding was observed with the DNA counterpart, suggesting that the N-IDR does not specifically recognize RNA (*i.e.*, the ribose 2'-OH moiety) (Supplementary Fig. 5).**

(Supplementary Fig. 5)

Supplementary Figure 5 NMR characterization of the interaction between the N-IDR and NRAS RNA or DNA. Plots of peak height ratios of the N-IDR signals obtained in the presence and absence of 0.5 eq. NRAS GQ RNA (left) or NRAS GQ DNA (right). The ratio was calculated by dividing the peak height in the presence of RNA or DNA by that in the absence. NMR measurements were performed at 10°C and 1 GHz using [U-¹³C, ¹⁵N]-labeled N-IDR NMR samples containing 20 mM potassium phosphate (pH 7.0), 250 mM KCl, 5 mM DTT, 260 units/mL RNasin® Plus RNAase inhibitor, and 5% D₂O. The protein concentration was 50 μM,

and the RNA/DNA concentration was 25 μ M.

Further, the stability of RNA G4 in condensates is worthwhile to discuss.

We appreciate the reviewer's comment regarding the stability of GQ structures within condensates. Several *in vitro* and *in cell* studies have suggested that GQ structures are preserved within cellular condensates, supporting the idea that GQ segments can serve as markers for subcellular localization under stress conditions. However, a recent study by Luo *et al.* (*Chembiochem* 1, 26(3), e202400791 (2025)) reported conflicting evidence, showing the melting of GQ structures within *in vitro* DDX4 condensates. These observations suggest that the folding stability of GQ RNAs within condensates may vary depending on their sequence, the type of condensate, and the cellular context. Such dynamic nature and variability in GQ stability may represent an additional layer of regulation in biological responses.

In the revised manuscript, we have added a discussion on the stability of RNA GQ structures in condensates as follows.

(Page 35, line 2)

Intriguingly, recent studies have shown that the N-IDR of DDX3X is also directly involved in the formation of and localization to stress granules^{21,28,30,31}. Therefore, the sequestration of DDX3X through the formation of stress granules, possibly with GQ RNAs, might be one of the underlying mechanisms that alter the translational landscape under stress conditions in humans as well. Consistent with this notion, Turner *et al.* reported that endogenous GQ RNAs colocalize with the stress granule marker G3BP1, as detected using GQ-binding molecules, suggesting that GQ structures are maintained within cellular condensates⁶³. Several *in vitro* studies have also supported the preservation of GQ structures within protein condensates^{92,93}; however, Luo *et al.* recently reported conflicting evidence suggesting the melting of GQ structures within *in vitro* DDX4 condensates⁹⁴. These observations indicate that the folding stability of GQ RNAs within condensates may vary depending on their sequence, the type of condensate, and the cellular context, potentially representing an additional layer of regulation in biological responses.

Reviewer #3:

We sincerely thank the reviewer for their valuable and insightful comments, which have significantly contributed to strengthening our conclusions and enhancing the overall clarity and readability of the manuscript. To address all of the reviewer's comments, we have conducted a number of additional experiments to strengthen our conclusions. Our point-by-point responses are summarized below.

Although this work is noteworthy, I found some of the discoveries redundant with previous work. The findings of the contribution of the N-terminal region of DDX3X on the helicase activity were already previously reported by other colleagues in the field (See Figure 4 of He et al, Biochemical and Biophysical Research Communications 2024, <https://doi.org/10.1016/j.bbrc.2024.149964>), therefore the whole Figure 1, although clean and rigorous, states known facts. Moreover, the contribution of electrostatic interactions and π -interaction in IDR–RNA (and more generally protein–RNA) interaction is well known, as reviewed by Corley et al, Molecular Cell 2020, doi: 10.1016/j.molcel.2020.03.011. This makes the data reported in Figure 2 and Supplementary Figure 3 of low impact.

We appreciate the reviewer's critical comments regarding the novelty of our work. We acknowledge that the roles of the N- and C-IDRs in the helicase activity of DDX3X have been reported previously, and we believe these papers are appropriately cited in our manuscript (refs 26 and 37).

(page 5, line 6)

Additionally, *in vitro* characterizations of DDX3X helicase activity have shown that the N- and C-terminal IDRs greatly stimulate the helicase activity of the folded core, with the N-terminal IDR providing a larger contribution^{26,37}.

The originality of our study lies more in elucidating the structural features of the interaction between DDX3X and its RNA targets, particularly the specificity for RNA recognition. The helicase assays using various DDX3X constructs were performed as part of the characterization of our purified

proteins, which we consider essential for establishing the foundation of our study. Importantly, in the revised version of the manuscript, we now provide additional insights that were not reported in previous studies, *i.e.* the characterization of the oligomerization state of each construct and the influence of the N-IDR on RNA binding affinity. We believe these new findings significantly strengthen the novelty of our work.

Regarding the contributions of electrostatic interactions and π -interactions, we acknowledge that these contributions have been well documented in structural studies of RNA-protein complexes, mostly involving folded proteins. However, to the best of our knowledge, there has been no thorough experimental evaluation of these contributions in the context of IDR-GQ RNA interactions. For example, while the suggested reference by Corley *et al.* (*Molecular Cell*, 78(1), 9–29, (2020)) discusses electrostatic interactions in IDR–RNA interactions, it does not address π -interactions, which are an important aspect highlighted by our data.

[Corley *et al.*, page 22 line 21]

“IDRs show little RNA sequence dependence, however, suggesting that these regions’ high affinity for RNA is predominantly driven by electrostatic attraction to the phosphodiester backbone.”

Given that NMR allows residue-level structural analysis of IDRs, we believe our study makes a significant contribution to the understanding of IDR-RNA interactions. We have added the suggested reference by Corley *et al.* (ref. 75) and expanded our discussion on the role of π -interactions in IDR-RNA interaction in the revised manuscript as follows.

(Page 32, line 11)

Although the detailed atomic structure of the GQ-bound state of the N-IDR is not available due to its inherently weak affinity and the inhomogeneous nature of the complex, our NMR analyses still provide some key structural insights into the interaction, highlighting the importance of electrostatic interactions and π -interactions involving the arginine-rich segments. **While IDR-RNA interactions are often thought to be primarily driven by electrostatic interactions between the negatively charged RNA phosphate backbone and polycationic IDR chains, our results indicate that π -interactions also make significant contributions as commonly observed in interactions involving RNA-folded protein complexes⁷⁵.**

1. The authors use NMR to characterise the interaction between the N-terminal IDR of DDX3X and a set of RNA targets, some single stranded—polyU—some structured—G-quadruplexes. Although the authors do provide a zoom-in of the ^1H - ^{15}N correlation NMR spectra throughout the manuscript, they never provide full spectra. All the full spectra should be included; the editor can advise in what form this should be done (supplementary data or supplementary figures), but they should be surely included in the manuscript.

We thank the reviewer for pointing this out. We have now included the full spectra from all NMR experiments in the Source Data file.

(Source Data [extracted])

(This Source Data includes all full spectra displayed in the main text and Supplementary Figures.)

2. In their interaction analysis using ^1H - ^{15}N correlation NMR, the authors state that very small shifts are observed, and the binding is claimed by mapping the change in peak intensity. However, in similar studies on intrinsically disordered proteins/regions, the interaction results in chemical shift perturbations. Can the author discuss why? Do they think that this is due to the formation of a fuzzy complex in solution or because of the weak interaction? Can this be due to aggregation? Have the authors tried to perform relaxation NMR or transverse-correlation experiments to corroborate the dynamics and nature of these interactions?

We appreciate the reviewer's comment. As the reviewer pointed out, the chemical shift perturbation (CSP) values observed in our study are relatively small compared to those observed in conventional NMR experiments. We believe this is mainly because the N-IDR is an intrinsically disordered protein lacking secondary structure in both the free and bound states. As a result, RNA binding does not induce large changes in chemical environment, such as ring current effects, formation of secondary structures, or reorganization of hydrogen-bond networks, that typically lead to large CSPs in folded proteins. We note that the magnitude of CSPs observed here is comparable to or even larger than those reported in similar studies on IDR-DNA interactions. For example, in the data shown below, a large excess of DNA (4-6 equivalent) was added when analyzing CSPs. Our experiments were performed using only 0.5–1 equivalent of RNA, suggesting that the magnitude of CSPs in our study is actually larger than in those previous reports when normalized for RNA concentration.

Adapted from Figure 3 in Ghosh and Singh, *Nucleic Acids Research*, 46, 19 10246–10261 (2018)

(B) 2D ^{15}N - ^1H HSQC spectrum of the RGG-box (black) and in complex with Tel22 at 1:6 protein to DNA molar ratio (red). Specific chemical shift perturbations were observed for several residues (marked with green arrows). A representative cartoon of monomeric G-quadruplex form of Tel22 is shown (only one conformation in K^+ ion is shown). (D) The titration curves showing chemical shift change plotted as a function of increasing DNA:protein ratio for 14 interacting residues of the RGG-box is shown.

Adapted from Fig.3 in Papageorgiou et al., *Nat Commun* 14, Article number: 6751 (2023)

a ^1H - ^{15}N HSQC spectra of 50 μM RSM titrated with double-stranded 10mer DNA (ds10). Some chemical shift perturbations (CSP) are indicated with arrows. Inset shows the crosspeaks of tryptophan sidechains. **b** CSPs of RSM residues induced by 4 \times molar addition of ds10.

While we have mainly used intensity reductions to identify interaction sites due to their high-intrinsic sensitivity, site-specific chemical shift perturbations also consistently indicate that GQ RNA primarily binds to the arginine-rich segments of the N-IDR (see Supplementary Fig. 11 below). In the revised manuscript, we have added an explanation regarding the small but significant chemical shift changes and included summary plots of chemical shift changes in ^{15}N and ^1H directions as follows.

(Page 14, line 14)

The titration series of telomeric GQ and NRAS GQ RNA showed that the chemical shift changes upon binding to GQ RNA were small in both the ^1H (< 0.04 ppm) and ^{15}N (< 0.12 ppm)

dimensions, and the effect of binding was mainly observed as an intensity reduction in a subset of residues (Fig. 3a, Supplementary Fig. 10a). Such small chemical shift changes have been reported in other IDR-nucleic acid interactions^{54,55}, likely reflecting the modest change in the local chemical environment due to the absence of drastic rearrangement in secondary or tertiary structures around the spin probe or hydrogen bond network. We note that, although the absolute chemical shift change values are small, they still provide site-specific information about the interaction consistent with that obtained from intensity changes (Supplementary Fig. 11).

(Supplementary Fig. 11)

Supplementary Figure 11 Chemical shift perturbations (CSPs) of the N-IDR upon the addition of RNA ligands. Plots of the ^{15}N (left) or ^1H (right) absolute chemical shift perturbation values of the [U- ^{15}N]-labeled N-IDR observed upon the addition of 1 equivolar (eq.) polyU₁₀ ssRNA, 1 eq. GC-14mer dsRNA, 1eq. 14mer tetraloop RNA, 1 eq. telomeric GQ RNA, 0.5 eq. NRAS GQ RNA, and 0.5 eq. NRAS mutant RNA. The three RR clusters in the sequence, defined as RR-1, RR-2, and RR-3, are indicated above each plot. The KCl concentration was 200 mM. All NMR measurements were performed at 10°C and 1 GHz, with a protein concentration of 50 μM .

Have the authors tried to perform relaxation NMR or transverse-correlation experiments to corroborate the dynamics and nature of these interactions?

We thank the reviewer for the suggestion. To better characterize the origin of the intensity reductions, we performed ^{15}N and ^1H Carr-Purcell-Meiboom-Gill (CPMG) relaxation dispersion experiments in the presence of GQ RNA. We first analyzed the ^{15}N or ^1H R_{ex} contributions, which are sensitive to microsecond-to-millisecond exchange processes. If the intensity reductions were mainly due to exchange between the free and bound states accompanied by chemical shift differences, we would expect to observe large R_{ex} contributions in residues showing marked intensity reductions upon GQ RNA titration. Notably, we did not detect significant R_{ex} contributions in these residues, either in the presence or absence of GQ RNAs, suggesting that the intensity reductions are not simply attributable to millisecond-to-microsecond exchange between free and bound states (see Supplementary Fig. 12 below). Instead, we observed marked increases in the apparent R_2 rates, measured at a 1000 Hz CPMG field, upon addition of GQ RNA. These increases were evident in the both ^1H and ^{15}N dimensions: 1.3-fold and 1.2-fold increases for ^{15}N and ^1H , respectively, in the presence of 0.5 eq. telomeric GQ RNA; and 1.5-fold and 1.3-fold increases, respectively, for 0.2 eq. NRAS GQ RNA, averaged over all residues. These elevated R_2 rates quantitatively account for the observed signal height reductions in the titration experiments. For example, in the presence of 0.5 equimolar telomeric GQ RNA, the expected average signal height reduction is ~ 0.64 ($= 1/1.3 \times 1/1.2$), considering broadening in both the ^{15}N and ^1H dimensions. This value closely matches the observed average decrease of ~ 0.67 (Fig. 3c), suggesting that the signal reductions are not due to irreversible aggregation or precipitation of the complex which would escape detection in the NMR sample tube.

Our current interpretation is that the free and bound states are in fast-exchange and the R_2 of the GQ RNA bound state is markedly larger than that of the free state, presumably due to the heterogeneity of the bound ensemble. As the reviewer noted, such heterogeneity can be viewed as the formation of a fuzzy complex, which may involve rapid exchange between multiple different bound conformations with distinct chemical shifts, restriction of the NH bond vector as a result of binding, and/or the formation of transient higher-order assemblies involving multiple N-IDR and GQ RNA molecules (*i.e.*, non 1:1 interactions) that increase the apparent molecular weight. These interpretations are

consistent with the apparently low Hill coefficient observed in our EMSA analyses and the reduced apparent stoichiometry in our ITC experiments. We will revisit this point in our response to point 4 below.

In the revised manuscript, we have added a discussion of the intensity reductions and included the results of the CPMG analyses as follows.

(Page 15, line 3)

To gain further insight into the interaction, we investigated the origin of the signal intensity reduction observed upon the addition of GQ RNA. The simplest explanation is that the free and bound states are exchanging on the microsecond-to-millisecond timescale, resulting in exchange broadening⁵⁶. This is a common feature of relatively weak binding interactions with K_d values in the micromolar range, as is the case here. To test this, we performed ¹⁵N and ¹H Carr-Purcell-Meiboom-Gill (CPMG) relaxation dispersion experiments^{57,58} to extract such potential exchange-induced broadening (R_{ex}) contributions in the presence of GQ RNA (Supplementary Fig. 12a, b). Intriguingly, the R_{ex} values measured in the presence of 0.5 equimolar telomeric GQ RNA or 0.2 equimolar NRAS GQ RNA were generally smaller than 5 s^{-1} , suggesting that the exchange broadening between free and bound states is not the major cause of the observed intensity reductions. Instead, we observed an overall increase in the apparent ¹⁵N and ¹H R_2 rates measured at a 1 kHz CPMG field (1.3-fold and 1.2-fold increases for ¹⁵N and ¹H, respectively, in the presence of telomeric GQ RNA; and 1.5-fold and 1.3-fold increases, respectively, for NRAS GQ RNA, averaged over all residues). These increases in apparent R_2 rates are broadly consistent with the observed signal intensity reductions. For example, the average decrease in signal height in the presence of 0.5 equimolar telomeric GQ RNA is expected to be ~ 0.64 ($= 1/1.3 \times 1/1.2$), considering the broadening in both the ¹⁵N and ¹H dimensions. This value closely matches the observed average decrease of ~ 0.67 (Fig. 3c), suggesting that the signal reduction is unlikely to result from irreversible aggregation or precipitation of the complex that escape detection in the NMR sample tube. Given that exchange between free and bound states occurs in the fast exchange regime, the increase in R_2 rates likely reflects elevated apparent R_2 in the GQ RNA-bound state, which may be attributed to heterogeneity within the bound ensemble, restricted picosecond-to-nanosecond motions of the amide ¹H-¹⁵N bond vector upon binding, and/or an increase in the apparent molecular weight due to complex formation involving multiple

N-IDR and GQ RNA molecules. In what follows, we focus on signal intensity reduction as a hallmark of GQ binding owing to its high intrinsic sensitivity.

(Supplementary Fig. 12)

Supplementary Figure 12 ^{15}N and ^1H Carr-Purcell-Meiboom-Gill (CPMG) relaxation dispersion analyses of the N-IDR with GQ RNA ligands. Plots of ^{15}N R_{ex} contributions (*top left*), ^1H R_{ex} contributions (*top right*), ^{15}N R_2 values measured at a 1,000 Hz CPMG field (*bottom left*), and ^1H R_2 values measured at a 1,000 Hz CPMG field (*bottom right*), recorded in the

absence (navy) or presence (pink) of GQ RNA. Results with GQ RNA were obtained in the presence of 0.5 eq. telomeric GQ RNA (a) or 0.2 eq. NRAS GQ RNA (b). The KCl concentration was 200 mM. All NMR measurements were performed at 10°C and 1 GHz, with a protein concentration of 50 μ M.

3. The authors perform the NMR experiments on IDR-RNA complexes at a salt concentration that is above the physiological conditions, as themselves explain at page 10, to push the interaction between protein and RNA. What is the landscape of the interaction at physiological salt concentration (~120 mM NaCl)? In what way is the interaction affected? Do the overall observed trends change?

We thank the reviewer for their comment. To address this point, we conducted the NMR binding experiments under more physiologically relevant salt conditions (120 mM KCl) (see Supplementary Fig. 4 below). The interaction profiles were very similar to those observed at 200 mM KCl, indicating that the interaction landscape is not strongly affected under physiological salt concentrations.

In the revised manuscript, we have added the results obtained at 120 mM KCl as follows.

(Page 10, line 21)

We note that very similar structural preferences were observed in a set of NMR experiments conducted under more physiologically relevant salt conditions (120 mM KCl) (Supplementary Fig. 4). These results demonstrate that the N-IDR preferentially recognizes and binds to the GQ structure of RNA, while it does not strongly interact with ssRNA, dsRNA, or hairpin structures.

(Supplementary Fig. 4)

Supplementary Figure 4 NMR characterization of the interaction between the N-IDR and various RNA molecules under physiological salt conditions. The experimental details and plots are the same as in Supplementary Fig. 3, except that all NMR measurements were performed in a buffer containing 20 mM potassium phosphate (pH 7.0), 120 mM KCl, 5 mM DTT, 260 units/mL RNasin[®] Plus RNAase inhibitor, and 5% D₂O.

4.The binding curves calculated by ITC are quite concerning. The authors state that the stoichiometry observed is off due to the bad temperament of the samples, but they still trust (up to a certain extent) the K_d values obtained. I am afraid that this must be repeated using an orthogonal technique (BLI, MST, EMSA...) to make sure that the results obtained

describe the system correctly.

We appreciate the reviewer's critical comment. As an orthogonal approach, we performed electrophoretic mobility shift assays (EMSA) to assess the interactions between the N-IDR and various RNA ligands (see Supplementary Fig. 6 below). Consistent with the NMR results, little to no band shift was observed for poly-U₁₂ ssRNA, GC-14mer dsRNA, and 14mer tetraloop RNA, whereas clear binding was observed for GQ RNAs. The apparent dissociation constants for telomeric and NRAS GQ RNAs were estimated to be $1.0 \pm 0.2 \mu\text{M}$ and $54 \pm 12 \text{ nM}$, respectively. The apparent binding affinity for the NRAS mutant RNA ($K_d = 6.0 \pm 0.8 \mu\text{M}$) was about two orders of magnitude weaker than that of the NRAS GQ RNA, further confirming the GQ-specific recognition by the N-IDR. In the revised manuscript, we have included these EMSA results as follows.

(Page 11, line 7)

As an orthogonal test, we conducted electrophoretic mobility shift assays (EMSA)⁵⁰ using fluorescently labeled RNAs to verify the binding preference for GQ RNAs and to obtain rough estimates of the apparent dissociation constant (K_d) values (Supplementary Fig. 6). Consistent with the NMR results, little to no band-shift was observed for poly-U₁₂ ssRNA, GC-14mer dsRNA, and 14mer tetraloop RNA, whereas clear binding was observed for GQ RNAs. The apparent affinities for telomeric and NRAS GQ RNAs were estimated to be $1.0 \pm 0.2 \mu\text{M}$ and $54 \pm 12 \text{ nM}$, respectively, at 125 mM KCl. Although binding to the NRAS mutant RNA was also observed, its apparent K_d ($6.0 \pm 0.8 \mu\text{M}$) was about two orders of magnitude weaker than that of the NRAS GQ RNA, further confirming the GQ-specific recognition by the N-IDR.

(Supplementary Fig. 6)

Supplementary Figure 6 EMSA analysis of the interaction between the N-IDR and various RNA ligands. EMSA binding experiments were performed for N-IDR using poly-U₁₂-FAM ssRNA, GC-14mer-FAM dsRNA, FAM-14mer tetraloop RNA, FAM-NRAS GQ RNA, FAM-telomeric GQ RNA, and FAM-NRAS mutant RNA. Gel images and fitted binding curves used to estimate K_d values are shown. Unlike in the NMR experiment, poly-U₁₂ was used instead of poly-U₁₀ in EMSA for synthetic reasons. FAM labeling was introduced at the 3'-end for poly-U₁₂ ssRNA and GC-14mer dsRNA, and at the 5'-end for all other RNAs. N-IDR protein was titrated from 0 to 45 μM. The bound fraction was estimated from the intensity of the free RNA probe and fitted to a standard Hill-type equation to obtain apparent K_d values. Free and bound probes were separated using polyacrylamide gels: 8% for all RNAs except the 14mer tetraloop, which was run on a 12% gel. All gels were prepared with 0.5× TBE buffer. Error bars represent the standard deviation ($n = 3-4$), with center indicating mean values.

While these EMSA results provide orthogonal evidence for the direct binding of the N-IDR and its selectivity toward GQ RNA, which is the main focus of our study, the affinity values obtained from EMSA and ITC do not perfectly match (1.0 ± 0.2 μM for telomeric GQ and 54 ± 12 nM for NRAS GQ in EMSA at 125 mM KCl, versus 3.4 μM for telomeric GQ and 1.4 μM for NRAS GQ in ITC at 50 mM KCl). We believe that this discrepancy most likely arises from the complex nature of the N-IDR-GQ interactions, where multiple N-IDR and GQ RNA molecules form heterogeneous complexes

with variable and not uniformly defined stoichiometry. In such scenarios, all three methods (NMR, EMSA, and ITC) are sensitive to distinct microscopic processes, which can potentially lead to deviations in apparent K_d values when analyzed with oversimplified binding models. For example, NMR might be sensitive to both IDR-IDR and IDR-GQ RNA interactions with fast dissociation and association kinetics, EMSA mainly detects slowly dissociating IDR-RNA complexes, and ITC is sensitive to heat changes arising from potential self-association of either the N-IDR or GQ RNA in addition to direct binding. If such oligomerization/self-association of N-IDR or GQ RNA processes are involved, signals from these methods can depend on whether the N-IDR or GQ RNA is used as the titrant. Similar discrepancies have been reported in recent studies of IDR-IDR interactions (H1.0 and Pro α), where picomolar K_d values from single-molecule FRET, sub-micromolar K_d from ITC, and fast-exchange behavior in NMR (typical of high-micromolar affinity) were observed (Borgia *et al.*, *Nature* (2018), 555 (7694), 61-66; Feng *et al.*, *Biochemistry*, 57, 48, 6645–6648). Chowdhury *et al.* recently proposed that such discrepancies can be resolved by taking into account the presence of higher-order ternary complexes (Chowdhury *et al.* *PNAS* 2023 120 (41) e2304036120). While it may be possible to develop a unified model including N-IDR and GQ RNA oligomerization to comprehensively explain all of our results; however, the presence of phase-separated states (discussed in our response to the next point below) significantly complicates such analysis. Given that the main conclusion of our study is the preferential binding of the N-IDR to specific structured RNAs, reasonably supported by both EMSA and NMR, we believe that an extensive thermodynamic characterization underlying this process lies beyond the scope of this study.

In the revised manuscript, we have added the following explanations regarding the apparent K_d values obtained from EMSA and ITC as follows.

(Page 11, line 15)

Additionally, we performed isothermal titration calorimetry (ITC) to analyze the binding of the N-IDR to these GQ RNA molecules (Supplementary Fig. 7). In the presence of 50 mM KCl, the apparent K_d values were calculated to be 3.4 μ M for telomeric GQ and 1.4 μ M for NRAS GQ, based on fitting to the standard n -site binding model, confirming the direct binding of N-IDR to GQ RNAs. We note that, whereas the apparent K_d values for telomeric GQ were comparable between EMSA and ITC, those for NRAS GQ differed significantly. We believe this discrepancy reflects the complex nature of the binding interaction, in which multiple N-IDR molecules may

be involved in binding a single GQ RNA molecule, supported by Hill coefficients deviating from one in EMSA and the number of binding sites, N , being less than one in ITC analyses. In addition, ITC is sensitive to heat changes arising from potential self-association of either the N-IDR or GQ RNA, further complicating interpretation. Indeed, apparent discrepancies between ITC and other methods when assuming an oversimplified binding model have recently been reported in other systems involving IDRs^{51,52}. Given the inherent complexity of the interaction between the N-IDR and GQ RNA, the apparent K_d values are highly dependent on the experimental method used, as well as on whether the N-IDR or GQ RNA is used as the titrant. Therefore, we do not attempt to derive a unified binding model that can comprehensively explain the NMR, ITC, and EMSA results, as this would be beyond the scope of the present study.

Moreover, the fact that the authors observe aggregation raises a massive concern regarding all sets of NMR data presented. Can the authors state confidently that the reduction of signal observed in their NMR data is due to interaction rather than aggregation? Can the authors provide hard evidence of this (DLS or similar)? This also connects to point 2. The small shifts observed versus the change in peak intensity could be due to overall (micro or macro)aggregation phenomena happening in the NMR tube, and this must be investigated better.

We appreciate the reviewer's critical comments regarding the potential aggregation formation in our system.

As mentioned in our response to point 2 above, the observed intensity reductions can be reasonably explained by elevated R_2 rates in the bound form. For example, in the presence of 0.5 equimolar telomeric GQ RNA, the expected average signal height reduction is ~ 0.64 ($= 1/1.3 \times 1/1.2$), considering broadening in both the ^{15}N and ^1H dimensions. This value closely matches the observed average decrease of ~ 0.67 (Fig. 3c), suggesting that the signal reductions are not due to irreversible aggregation or precipitation of the complex which would escape detection in the NMR sample tube. Although the intensity reductions appear substantial, the corresponding increases in R_2 rates are modest, on average 1.3-fold with 0.5 eq. telomeric GQ and 1.5-fold with 0.2 eq. NRAS GQ in the ^{15}N

dimension, suggesting that these are not caused by the formation of massively large aggregates typically observed with protein fibrils or denatured aggregates.

As the reviewer rightly noted, light-scattering techniques such as DLS or SAXS would be a method of choice for convincingly ruling out the possibility of aggregation. However, we found that, at the high protein concentrations required for these analyses, the N-IDR and GQ RNA co-phase separate to form liquid droplets (Supplementary Fig. R2), significantly hampering the application of these light-scattering methods as they are extremely sensitive to the presence of micro-meter sized droplets. Importantly, the formation of such liquid droplets containing DDX3X and RNA may be related to the formation of and localization to cellular stress granules, which are currently being investigated in our laboratory.

Supplementary Figure R2. Fluorescence image of the Alexa 555-labeled N-IDR (left) and FAM-labeled GQ RNA (middle). The merged images of the N-IDR and GQ RNA are shown in the right column. The images were measured in the presence of 100 μM telomeric GQ (top) or 75 μM NRAS GQ (bottom). The total N-IDR concentration was 100 μM, containing 2-3 % of Alexa 555-labeled N-IDR. The white scale bar corresponds to a length of 20 μm.

A number of experimental observations; such as increased R_2 values in the bound state, low stoichiometry in ITC, and apparent Hill coefficients deviating from one in EMSA, and the formation of liquid droplets by the N-IDR and GQ RNA, suggest that the complex formed between the N-IDR and GQ RNA cannot be described as a homogenous one-to-one complex, rather indicate that multiple N-IDR and GQ RNA molecules are involved in forming a heterogeneous complex with variable and

not uniformly-defined stoichiometry. Importantly, several previous studies have proposed that multiple DDX3X molecules are involved in the RNA unwinding process (Floor et al., *J. Biol. Chem.* **291**, 2412–2421 (2016); Song and Ji, *Nat. Commun.* **10**, 1–8 (2019); Sharma *J. Mol. Biol.* **429**, 3730–3742 (2017)) and that the formation of nanometer-scale DDX3X-RNA clusters underlies its catalytic activity (Yanas, et al., *Curr. Biol.* **34**, 5714-5727.e6 (2024)). Thus, the self-association or oligomerization of N-IDR may be functionally important, enabling the formation of catalytically active complexes during the RNA unwinding process. Furthermore, as noted above, self-association could represent a nucleation event that precedes the formation of phase-separated cellular condensates such as stress granules (Saito, M. *et al. Nat. Chem. Biol.* **15**, 51–61 (2019); Shen *et al. Mol. Cell* **82**, 2588-2603.e9 (2022)). We therefore emphasize that the heterogeneous nature of the N-IDR-GQ RNA complex likely reflects a physiologically relevant mode of interaction that facilitates efficient catalysis and/or the precise and dynamic regulation of sub-cellular organization, rather than being a consequence of experimental artifact or bad sample treatment.

To clarify these points, we have added the following discussion in the revised manuscript.

(Page 33, line 16)

A number of experimental observations, such as increased R_2 values in the bound state, low stoichiometry in ITC, and apparent Hill coefficients deviating from one in EMSA, suggest that the complex formed between the N-IDR and GQ RNA cannot be described as a homogenous one-to-one complex, rather indicate that multiple N-IDR and GQ RNA molecules are involved in forming a heterogeneous complex with variable and not uniformly-defined stoichiometry. This is not unexpected, given that N-IDR contains multiple RR clusters, each of which can independently bind RNA with comparable affinity, thereby allowing for a variety of distinct binding modes. We also underscore the likely involvement of multiple DDX3X molecules in complex formation, potentially enabling self-oligomerization of DDX3X around the RNA ligand. Several studies have proposed that multiple DDX3X molecules are involved in the RNA unwinding process^{26,27,87}, and that the formation of nanometer-scale DDX3X-RNA clusters underlies its catalytic activity⁸⁸. Additionally, such self-association may also represent a nucleation event that precedes the formation of phase-separated cellular condensates such as stress granules^{28,30,31}. We therefore emphasize that the heterogeneous nature of the N-IDR-GQ RNA complex likely reflects a physiologically relevant interaction that facilitates efficient

catalysis and/or precise and dynamic regulation of sub-cellular organization.

5. The authors state that there are three binding sites in the N-IDR, around residues 40, 90 and 120, indicating that the binding cannot be described by a conventional one-site binding model with a single defined K_d value. I am afraid that the authors cannot state this confidently, as they do not present enough biophysical and structural data to support this claim. First, ^{15}N -correlation data are never indicative of a direct interaction, as the change in intensity or shift of the amide peaks can be due to an indirect effect. The fact that the authors see three main binding regions in their NMR plots does not mean that the three sites are independent. In fact, unclear stoichiometry values from the ITC experiments make this data interpretation even weaker (see point 4). The observed NMR data are compatible with the N-IDR decorating the RNA, maintaining part of the protein in a fuzzy state. This is quite evident from the titration with NRAS GQ, where, at 200 mM salt, pretty much the whole region between residues 40 and 120 seems to participate to RNA binding. To prove their claim, the authors must take a different approach. They could either perform SAXS data to see whether the N-IDR goes from an extended to a globular state upon RNA-binding, in which case most likely the three sites are not independent. Moreover, they should test the RNA-binding activity of the three regions carrying the three different RGG boxes and see whether they can independently bind to the RNA targets (using any interaction technique of their choice).

6. From all their NMR data, it seems that the RGG box around site 40 is predominant. The authors have not discussed and thoroughly investigated this, but they should. A better description of this must be provided, to clarify whether the three sites are independent or not, as well as the perturbation of the RGG boxes around 90 and 120 is direct or indirect.

We thank the reviewer for the critical comments regarding our interpretation of the presence of multiple RNA binding segments within the N-IDR. We agree that NMR signal intensity reductions or chemical shift changes do not directly indicate binding, and that the signal changes observed in each Arg-rich region could, in principle, reflect indirect effects, such as compaction into a globular

conformation upon RNA binding. While SAXS would be a very powerful method to characterize such potential conformational changes in IDRs upon RNA binding, the formation of liquid droplets under high concentration conditions required for SAXS unfortunately limits its applicability in our system, as noted in our response to point 4. To more directly assess the role of each RR region in RNA binding, we therefore took the second option and performed binding experiments using individual RR fragment proteins. This set of data also addresses the reviewer's point 6, and we have summarized our responses within this section.

We prepared three RR fragment proteins, RR-1 (residues 31–61), RR-2 (residues 73–102), and RR-3 (residues 106–127), and performed NMR binding experiments with various RNA ligands (Supplementary Figs. 15, 16, and 17). For all RR fragment proteins, significant intensity reductions and chemical shift changes were observed only with telomeric and NRAS GQ RNAs, while little to no spectral changes were seen for poly-U₁₀ ssRNA, GC-14mer dsRNA, and NRAS mutant RNA, unequivocally showing that each RR region is capable of binding RNA and exhibits specificity for GQ structures.

The intensity reduction profiles were overall similar among the three RR regions, although minor differences were observed in their relative preferences between telomeric and NRAS GQ RNAs. For example, RR-2 showed comparable intensity reductions upon the addition of 1 equimolar telomeric GQ and 0.5 equimolar NRAS GQ (Supplementary Fig. 16), whereas RR-3 exhibited larger intensity reductions with telomeric GQ than with NRAS GQ (Supplementary Fig. 17). These subtle differences may reflect distinct structural preferences or binding modes toward GQ RNAs with different topologies; however, we did not observe clear functional differences among these three RR regions, at least in the context of the N-IDR's GQ-specific recognition.

Direct binding of RR-1 and RR-2 to GQ RNAs was further confirmed by ITC analyses, which yielded apparent K_d values ranging from 2 to 11 μM , comparable to those of the full N-IDR ($K_d \sim 1\text{--}3 \mu\text{M}$) (Supplementary Fig. 18). While we acknowledge that the K_d values derived from ITC should be interpreted with caution, these results, at least qualitatively, support the notion that each RR segment can independently bind to GQ RNA with comparable affinity and that there is no strong structural coupling or binding cooperativity among the three RR clusters.

As the reviewer pointed out, we observed slightly larger intensity reductions around residue 40 in the NMR binding experiment with the full N-IDR and telomeric GQ RNA. However, we did not detect a stronger binding affinity for the RR-1 fragment protein, which includes this region, in either NMR or ITC analyses. This suggests that the observed larger intensity reductions are more likely due to restricted motion in the bound state, particularly involving the characteristic successive proline residues (Pro40-Pro41), rather than preferential binding of GQ RNA to this region.

To clarify these points, we have added a section discussing the binding properties of each RR motif and supplementary figures in the revised manuscript. We note that these results do not necessarily exclude the possibility of the formation of globular conformation upon interaction between the N-IDR and GQ RNA, as suggested by the reviewer. Although such a conformation may indeed form, we currently lack sufficient structural data to support this hypothesis. Therefore, we chose not to discuss further in the manuscript, which we believe is not essential for supporting our main conclusions.

(Page 19, line 20)

To further investigate the independent binding activity and individual contributions of each RR cluster to GQ RNA recognition, we employed an alternative approach by monitoring the interaction using isolated RR fragments. To this end, we prepared three RR fragment proteins, RR-1 (residues 31–61), RR-2 (residues 73–102), and RR-3 (residues 106–127), and performed NMR binding experiments with various RNA ligands (Supplementary Figs. 15, 16, and 17). Most of the signal assignments for each RR region could be readily transferred from those of the full N-IDR, indicating that the disordered nature of the N-IDR is preserved in each isolated RR construct. Notably, for all RR fragment proteins, significant intensity reductions and chemical shift changes were observed only with telomeric and NRAS GQ RNAs, while little to no spectral changes were seen for poly-U₁₀ ssRNA, GC-14mer dsRNA, and NRAS mutant RNA, demonstrating that each RR region independently exhibits GQ-specific binding. The intensity reduction profiles were overall similar among three RR regions, although minor differences were observed in their preferences between telomeric and NRAS GQ RNAs. For example, RR-2 showed comparable intensity reductions upon the addition of 1 equimolar telomeric GQ and 0.5 equimolar NRAS GQ (Supplementary Fig. 16), whereas RR-3 exhibited larger intensity reductions for telomeric GQ than for NRAS GQ (Supplementary Fig. 17). These subtle

differences may reflect distinct structural preferences or binding modes toward GQ RNAs with different topologies⁶¹; however, we did not observe clear functional differences among these three RR regions, at least in the context of the N-IDR's GQ-specific recognition. Direct binding of RR-1 and RR-2 to GQ RNAs was further confirmed by ITC analyses, which yielded apparent K_d values ranging from 2 to 11 μM , comparable to those of the full N-IDR ($K_d \sim 1\text{-}3 \mu\text{M}$) (Supplementary Fig. 18). These results further support the notion that there is no strong structural coupling or binding cooperativity among the three RR clusters. In the NMR binding experiment with telomeric GQ RNA, we observed slightly larger intensity reductions around residue 40 of the N-IDR (Fig. 3c). However, we did not detect stronger affinity for the RR-1 fragment protein that contains this region. This suggests that the observed larger intensity reductions are more likely due to restricted motion involving the characteristic successive proline residues (Pro40-Pro41) in the bound state, rather than the preferential binding of GQ RNA to this region.

(Supplementary Figs. 15-18)

Supplementary Figure 15 NMR characterization of the interaction between the RR-1 fragment and various RNA molecules. (a) NMR spectra and peak height ratios of RR-1 signals obtained with and without each RNA molecule. For each sub-panel, the overlay of ^{15}N - ^1H HSQC spectra of the [^{15}N]-labeled RR-1 in the absence (navy) and presence (orange-red) of RNA,

and the plot of peak height ratios are shown. The ratio was calculated by dividing the peak height in the presence of RNA by that in the absence of RNA. Residues that were not analyzed are indicated by gray backgrounds. All NMR measurements were performed at 10 °C and 1 GHz in a buffer containing 20 mM potassium phosphate (pH 7.0), 50 mM KCl, 5 mM DTT, 260 units/mL RNasin® Plus RNAase inhibitor, and 5% D₂O. The protein concentration was 50 μM, and the RNA concentration was 50 μM (for poly-U10 ssRNA, GC-14mer dsRNA, and telomeric GQ RNA) or 25 μM (for NRAS GQ RNA and NRAS GQ mutant RNA). The 1D slices of the labeled signals are shown in each spectrum. (b) Full ¹⁵N-¹H spectrum of [U-¹⁵N]-labeled RR-1 with signal assignments.

Supplementary Figure 16 NMR characterization of the interaction between the RR-2 fragment and various RNA molecules. The experimental details and plots are the same as in Supplementary Fig. 15.

Supplementary Figure 17 NMR characterization of the interaction between the RR-3 fragment and various RNA molecules. The experimental details and plots are the same as in Supplementary Fig. 15.

Supplementary Figure 18 ITC experiments for RR-1/RR-2 and GQ RNA interactions. (a) ITC isotherm (top) and integrated heats (bottom) for the titration of telomeric GQ RNA (250 μM as a dimer) into solutions of the RR-1 (left) or RR-2 (right) fragment protein (100 μM). (b) ITC isotherm (top) and integrated heats (bottom) for the titration of NRAS GQ RNA (300 μM as a single strand) into solutions of the RR-1 (left) or RR-2 (right) fragment protein (50 μM). The titration data were analyzed using the standard binding model assuming one set of sites, and the fitted parameters are displayed in the insets. All measurements were performed at 25 $^{\circ}\text{C}$. Experiments on RR-3 were not included due to poor reproducibility, likely due to a tendency to form small amounts of aggregates.

7. Figure 5e seems to be identical to what reported by the same authors in their previous

work Toyama et al, Nat Comm 2024 15, Article number: 3303 at Figure 2d. The panel from the previous work states that the molar ratio is 2, while here it is 4, but they seem identical. Can the authors explain?

We thank the reviewer for the comment. The spectra shown in Fig. 5e are not identical to those reported in our previous study (Toyama and Shimada, *Nat. Commun.* 2024, 15, 3303). In the present study, we used [Met- ^{13}C]-labeled DDX3X, recorded spectra at 1 GHz, and added 4 eq. poly-U₁₀. In contrast, in our previous study, we used [Frac-/U- ^2H ; Ile δ 1- $^{13}\text{C}^1\text{H}_3$; Met ϵ - $^{13}\text{C}^1\text{H}_3$]-labeled DDX3X, recorded spectra at 600 MHz, and added 2 eq. poly-U₁₀.

8. The interpretation of Figure 5e and 5f is quite confusing to me. Figure 5e shows the set of Methionine shifts upon binding of D1/D2 (core) to polyU10 (single stranded RNA), overlaid to the core in its apo state. Figure 5f shows the overlay of the complexes of core with polyU10 and core with NRAS GQ (folded) and the authors state that a “distinct set of bound signals was observed”. To me, the bound signals are virtually identical, only M330 shifts very minorly less, but this could be due to lower affinity towards NRAS GQ.

We thank the reviewer for pointing out the apparent discrepancy between the fluorescence quenching assay and the NMR results and apologize for the confusion.

When we referred to a “distinct set of bound signals”, our intention was to indicate that the bound-state signals were observed in a slow exchange regime, and that the chemical shifts of the newly observed bound state were different from those of the RNA-free state. We did not mean to suggest that there was a significant difference between the NRAS GQ-bound and poly-U₁₀-bound states. In fact, the chemical shifts of the NRAS GQ-bound state are almost identical to those of the poly-U₁₀-bound state, which is the key conclusion of this section.

In the initial submission, we did not include an overlay of the RNA-free and NRAS GQ-bound DDX3X spectra, which likely contributed to the confusion. To clarify the spectral changes upon the

addition of NRAS GQ, we have included the overlay of the RNA-free and NRAS GQ-bound spectra in the revised manuscript (Fig. 5f). Then, we separately showed the overlay of the poly-U₁₀ bound and NRAS GQ-bound spectra to demonstrate the similarity between these two bound states (Fig. 5g). We have also revised the text accordingly.

(Figure 5)

Figure 5 GQ-unfolding activity of DDX3X. (e) Overlay of the spectra of [Metε-¹³C¹H₃]-labeled E348Q DDX3X Core measured without (gray) and with 4 equimolar poly-U₁₀ ssRNA (pink). (f) Overlay of the spectra measured without (gray) and with 6 equimolar NRAS GQ (purple). (g) Overlay of the spectra measured with 6 equimolar NRAS GQ (navy, multiple contours) and with 4 equimolar poly-U₁₀ ssRNA (pink, single contour). Schematic cartoons describing the interactions are shown below the spectra (PDB ID: 5E7M and 2DB3)^{26,32}. In panels (e) and (f), the projections of the dotted region are shown in the inset. Free (F) and bound (B) signals are indicated for representative residues. All NMR measurements were performed at 35°C and 1 GHz, with a protein concentration of 50 μM.

(Page 26, line 9)

As a reference, we measured the ¹³C-¹H HMQC spectra of the [Metε-¹³C¹H₃]-labeled core region with and without a model ssRNA substrate, poly-U₁₀. Methionine methyl signals from M221, M330, M352, M355, and M380 in the D1 domain showed significant chemical shift changes,

reflecting poly-U₁₀ binding and the accompanying structural rearrangement of the core region (Fig. 5e). We then measured the ¹³C-¹H HMQC spectrum of the [Metε-¹³C¹H₃]-labeled core region in the presence of an excess amount of NRAS GQ (300 μM) to promote binding (Fig. 5f). Upon addition of NRAS GQ, we observed a set of bound-state signals in a slow exchange manner, with an apparent bound population of ~30-50%, reflecting the inherently weak binding affinity of the core region for NRAS GQ without the N-IDR. Notably, comparison of the bound-state signals between poly-U₁₀ and NRAS GQ revealed that their chemical shifts were nearly identical, suggesting a close similarity in the bound conformation (Fig. 5g).

The core-NRAS GQ was recorded at a very high excess of NRAS GQ. Now the authors state that the fact they see these shifts means that NRAS GQ binds in single-stranded state to the core, although the authors stated that the core does not unwind GQ (see Figure 5d middle panel). This, to me, seems to be quite an overclaim. If the core does not unwind the GQ, how is this RNA in a single stranded state when bound to the core? The same shifts observed between polyU10 and NRAS GQ are not enough to substantiate this claim. The authors must perform a more detailed characterisation of this system (using 1H-NMR of NRAS GQ in its bound state to the core to claim single stranding of the GQ, as well as a better NMR characterisation of the overall bound states).

We again apologize for the confusion regarding the apparent discrepancy between the fluorescence quenching assay in Fig. 5d and the NMR results in Fig. 5f. Our interpretation is that the helicase core region without IDRs does retain the intrinsic ability to unfold GQ RNA. The apparent lack of activity observed in the fluorescence quenching experiments using MBP-Core-C (N-IDR-truncated DDX3X) was simply due to the low protein concentration employed. Note that, in our previous version, the fluorescence quenching assays used 40-160 nM MBP-Core-C and 20 nM fluorescently labeled NRAS GQ, whereas the NMR experiments used 50 μM core protein and 300 μM NRAS GQ to promote binding.

To more clearly evaluate the helicase activity of MBP-Core-C and the N-IDR, we repeated the

fluorescence quenching assays at higher protein concentrations up to 340 nM (see revised Fig. 5 and Supplementary Fig. 21 below). At concentrations above 200 nM, MBP-Core-C exhibited clear GQ-unfolding activity toward NRAS GQ, although the activity was weaker than that of the full-length protein (MBP-FL). In contrast, the N-IDR alone showed little to no unfolding activity at the same concentration, demonstrating that the Core-C region, rather than the N-IDR, is responsible for GQ-unfolding activity, and that the presence of the N-IDR significantly enhances this activity. The ability of the core region to unfold GQ is fully consistent with the NMR results shown in Fig. 5f and 5g.

To clarify these points, we have revised Fig. 5d, added Supplementary Fig. 21, and updated the corresponding text in the revised manuscript.

(Page 24, line 19)

We then measured the fluorescence of the modified NRAS GQ (20 nM) in the presence of varying concentrations of MBP-FL, MBP-Core-C, and N-IDR proteins (Fig. 5d). Upon the addition of MBP-FL, we observed an increase in fluorescence, with the signal nearly reaching a plateau at 2 equimolar concentrations (40 nM) of MBP-FL, suggesting that DDX3X can unwind the NRAS GQ structure. The apparent binding affinity of MBP-FL toward the dually labeled NRAS GQ was estimated to be on the order of 10 nM, consistent with the apparent K_d for MBP-FL-NRAS GQ interaction ($K_d = 43 \pm 4$ nM) obtained from EMSA (Supplementary Fig. 9). In contrast, almost no increase in fluorescence was observed upon the addition of the same concentration of MBP-Core-C or N-IDR (Fig. 5d). However, at a much higher concentration (~340 nM), MBP-Core-C showed a smaller but significant increase in fluorescence, while the N-IDR alone produced little to no effect (Fig. 5d, Supplementary Fig. 21a). These results indicate that the Core-C region, rather than the N-IDR, is responsible for the GQ-unfolding activity, and that this activity is significantly enhanced in the presence of N-IDR. Combined with EMSA results showing that the N-IDR significantly increases binding affinity to GQ RNA substrates (Supplementary Fig. 9), these findings indicate that the robust GQ-unfolding activity of MBP-FL arises from the synergistic actions of the N-IDR-mediated binding and the helicase activity of the folded core.

(Figure 5)

Figure 5 GQ-unfolding activity of DDX3X. (d) Fluorescence emission spectra of FAM/BHQ1-labeled NRAS GQ recorded without (solid black line) and with (dotted magenta line) the MBP-FL (left), MBP-Core-C (middle), and N-IDR (right) proteins. The intensity ratios of the maximum fluorescence intensities of FAM/BHQ1-labeled NRAS GQ at each protein concentration are plotted below. Error bars represent the standard deviation of three independent measurements.

(Supplementary Fig. 21)

Supplementary Figure 21 GQ-unfolding activity measured via fluorescence quenching. (a) Representative fluorescence emission spectra of FAM/BHQ1-labeled NRAS GQ recorded in the

absence (gray line) and presence (colored lines) of varying concentrations of MBP-FL (left), MBP-Core-C (middle), and N-IDR (right) proteins. Protein concentrations ranged from 0 to 40 nM for MBP-FL and from 0 to 340 nM for MBP-Core-C and N-IDR. The excitation wavelength was set to 490 nm.

As suggested by the reviewer, we conducted additional NMR experiments to probe the structural state of GQ RNA when bound to the DDX3X core region. To this end, we employed ^{19}F NMR using 2'- ^{19}F labeled GQ RNA and recorded ^{19}F spectra in the presence of and absence of DDX3X proteins, as we previously reported for characterizing RNA conformations bound to the DDX3X core (Toyama and Shimada, *Nat. Commun.* 2024, 15, 3303). We initially attempted to observe the ribose 2'- ^{19}F signal of NRAS GQ RNA (two labeling positions were tested). However, the ^{19}F signals were extremely broad, presumably due to structural polymorphism. Thus, we used telomeric GQ RNA where a ribose 2'- ^{19}F probe was introduced at the second adenosine (5'-UAGGGUUAGGGU-3', where the underlined A is labeled with 2'- ^{19}F), to monitor interactions with the DDX3X core. As a reference, we included previously reported data for the UA-12mer (5'-UUUAUUAAUAA-3', where the underlined A is labeled with 2'- ^{19}F), which binds to the DDX3X core with high affinity in its ssRNA form. Note that, under our experimental conditions, the UA-12mer exists in equilibrium between dsRNA and ssRNA forms.

Upon addition of 3 equimolar (as a monomer) E348Q DDX3X core in the ATP bound form, we did not observe a distinct bound-state signal; instead, only a small chemical shift change was observed (Supplementary Fig. R3). This suggests that the inherent affinity of the telomeric GQ RNA for the core region is weak, preventing clear observation of a tightly bound state by NMR. In addition, the telomeric 12mer exhibited significantly broader linewidths than the UA-12mer, likely due to conformational heterogeneity, making it further difficult to detect the even broader bound-state signal. This inherent weak binding of the core region to structured RNAs is consistent with our previous findings (Toyama and Shimada, *Nat. Commun.* 2024, 15, 3303) and was further supported by HMQC analyses of [Met- ^{13}C]-labeled core protein (data not shown). Increasing the concentration of DDX3X core to further push the equilibrium toward the bound state was not feasible due to the relatively poor solubility of the E348Q DDX3X core. We note that these results are not necessarily inconsistent with the unwinding assays described above, as those assays used FL-MBP or MBP-Core-C proteins, both

of which contain N- and/or C-IDRs and these IDRs greatly enhance RNA binding affinity.

Supplementary Figure R3. ¹⁹F NMR analyses of RNA. (a) ¹⁹F 1D spectra of UA-12mer in the absence (top) and presence or 2equimolar E348Q variant of DDX3X Core in the ATP-bound form. The experimentally obtained spectra are shown as navy lines, and deconvoluted lines of dsRNA (blue), ssRNA (turquoise), and bound (orange-red) signals are overlaid. Spectra were performed at 30 °C and 600 MHz. RNA concentration was 100 μ M (as a single strand). This figure was adapted from our previous paper (Toyama and Shimada, *Nat. Commun.* 2024, 15, 3303). (b) ¹⁹F 1D spectra of telomeric GQ in the absence (top) and presence or 3 equimolar E348Q variant of DDX3X Core in the ATP-bound form. Spectra were recorded at 35 °C and 600 MHz. RNA concentration was 50 μ M (as a single strand). Buffer condition was 20 mM potassium phosphate (pH 7.0), 100 mM KCl, 5 mM DTT, 5 mM ATP, 5 mM MgCl₂, 260 units/mL RNasin® Plus RNAase inhibitor, and 5% D₂O,

While direct NMR observation of GQ RNA bound to the core region proved technically challenging, mainly due to its inherently weak binding affinity, we believe that the NMR results obtained from the Met methyl signals of the core region (Fig. 5f, g) provide reasonably strong support for our conclusions. In our previous study, we proposed that the observed bound state corresponds to a closed conformation, where an elongated ssRNA binds both the D1 and D2 domains in a bipartite manner. Given that the NRAS GQ RNA adopts a compact folded conformation in its GQ topology

(Supplementary Fig. 1b), it is reasonable to expect that this GQ structure must be, at least partially, unwound to participate in such a bipartite interaction in an elongated form. To emphasize this point, we have revised the text as follows.

(Page 26, line 20)

In our previous study³⁴, we proposed that this bound state corresponds to the closed conformation, where an elongated ssRNA binds both the D1 and D2 domains in a bipartite manner. Given that NRAS GQ adopts a compact folded conformation in its GQ topology (Supplementary Fig. 1b), it is reasonable to expect that the GQ structure must be, at least partially, unwound to participate in the bipartite interaction in an elongated form. Although direct NMR observation of the bound NRAS GQ RNA was not feasible due to structural heterogeneity, modest binding affinity, and the large molecular weight of the complex, the close similarity in the bound conformation between poly-U₁₀ and NRAS GQ strongly supports our model that the core region binds to NRAS GQ by adopting the ssRNA-bound, closed conformation, accompanied by the unfolding of the GQ structure (Fig. 5f, g).

To further support the GQ unfolding activity of DDX3X, we included additional experimental evidence. In the revised manuscript, we have included a gel electrophoresis-based assay to monitor the GQ unfolding activity, as described by Gao *et al.* (*Chem. Commun.*, 55, 4467-4470 (2019)). Briefly, this assay monitors the accelerated hybridization of NRAS GQ to a complementary RNA strand as a result of GQ unfolding by DDX proteins (see Supplementary Fig. 22a below). We were able to detect an increase in the dsRNA fraction in the presence of both DDX3X and the complementary RNA strand at a fixed time point, clearly indicating GQ-unfolding activity of DDX3X toward NRAS GQ (Supplementary Fig. 22b, c).

Collectively, we have presented multiple lines of evidence to support our conclusions: (i) the GQ unfolding assay based on fluorescence quenching to monitor structural rearrangement of RNA, (ii) the GQ unfolding assay based on accelerated hybridization to probe structural destabilization of GQ RNA, and (iii) NMR experiments to examine the bound conformation of the DDX3X core. In addition, we have included new EMSA results demonstrating the effect of the N-IDR on the binding affinity of full-length DDX3X for GQ RNA using EMSA (see our response to point 10 below). We believe

these data provide reasonably strong support for our conclusion that the core and N-IDR of DDX3X synergistically contribute to the unwinding of GQ RNA.

In the revised manuscript, we have added the results of the GQ-unfolding assay monitored by hybridization to a complementary strand as follows.

(Page 25, line 17)

Supporting the results of the fluorescence quenching assay, the GQ-unfolding activity of MBP-FL was further confirmed by a GQ hybridization assay using a complementary RNA strand⁶⁸ (Supplementary Fig. 22).

(Supplementary Fig. 22)

Supplementary Figure 22 GQ-unfolding assay monitored by hybridization to a complementary RNA strand. (a) Schematic representation of the NRAS GQ-unfolding assay, in which GQ-unfolding is detected by hybridization to a complementary RNA strand. (b) Native gel analysis of reaction mixtures containing 50 nM FAM-labeled NRAS GQ RNA, with or without 100 nM complementary RNA, and with or without 100 nM DDX3X (MBP-FL). The two rightmost lanes show samples incubated at 95 °C for 3 minutes prior to electrophoresis. Samples were separated using 20% polyacrylamide gels prepared with Tris-Glycine buffer. (c)

Plots of the intensity ratio between the duplex and NRAS GQ bands for lanes containing 100 nM complementary strand. Error bars represent the standard deviation ($n = 4$), with center indicating mean values.

9. The model in Figure 6b seems to indicate that, upon binding, the N-IDR fold onto the core, however, the authors do not provide evidence for this model. To confirm this, they must perform SAXS experiments with either the FL or the construct spanning residues 1-607 to see the claimed conformational change upon binding. Alternatively, NMR could be of use here too, looking at how the core is perturbed by the N-IDR in its apo and RNA-bound states.

We thank the reviewer for the comment. In our original Fig. 6b, it was not our intention to claim that the N-IDR folds onto the core upon the interaction, rather the position of the N-terminus happened to be close to the core in the cartoon. To avoid any confusion, we have revised the cartoon so that the direct interaction between the N-IDR and the core is not implied.

(Figure 6b)

Figure 6 GQ propensity of DDX3X's targets. (b) Cartoon representations of the interaction between an mRNA transcript containing the GQ structure and DDX3X.

That being said, it is important to better characterize the structural state of the N-IDR when tethered to the core and to determine whether its conformation is affected by the RNA binding in the full-

length context. As the reviewer suggested, SAXS would be a powerful method in probing the conformational states of DDX3X; however, as noted above, the formation of the aggregates or liquid droplets precludes the application of SAXS in this case.

To this end, we conducted NMR experiments observing the N-IDR attached to the helicase core and monitored its binding to RNA ligands. We prepared [U-¹⁵N] N-Core-MBP, in which an MBP tag was fused to the C-terminus of DDX3X residues 1–607 (N-IDR + Core) to maintain solubility while minimizing structural interference with N-IDR-RNA interactions. In the ¹⁵N-¹H HSQC spectrum of [U-¹⁵N] N-Core-MBP, signals from the N-IDR region could be observed (see Supplementary Fig. 8 below). The chemical shifts of the observed N-IDR signals were consistent with those in the isolated N-IDR, supporting that the N-IDR retains a flexible conformation when attached to the helicase core.

Although only a limited number of NMR probes from the N-IDR were available due to line broadening of signals from the middle to C-terminal region, we were still able to monitor the N-IDR signals in [U-¹⁵N] N-Core-MBP to evaluate its binding to GQ RNA. As in the isolated N-IDR, signal intensity reductions were consistently observed upon the addition of NRAS GQ RNA, indicating that the interaction persists when the N-IDR is part of the larger protein construct. Importantly, no substantial chemical shift changes or emergence of folded-state signals were observed upon RNA binding, suggesting that the N-IDR remains flexible in both the free- and RNA-bound states, even within the full-length protein context.

To clarify these points, we have added the following explanations and supplementary figures in the revised manuscript. The revised Figure 6 above aligns with our view that the N-IDR retains its flexibility in both the free- and RNA-bound forms.

(Page 12, line 13)

Before investigating the molecular interaction between the N-IDR and GQ RNAs in detail, we confirmed that the interaction observed with the isolated N-IDR is retained in the full-length protein context. To assess the binding of the N-IDR to GQ RNA when tethered to the folded helicase core, we prepared [U-¹⁵N] N-Core-MBP, in which an MBP tag was fused to the C-terminus of DDX3X residues 1–607 (N-IDR + Core) to maintain solubility while minimizing structural interference with N-IDR-RNA interactions. In the ¹⁵N-¹H HSQC spectrum of [U-¹⁵N]

N-Core-MBP, signals from the N-IDR region could be observed due to its inherent structural flexibility, while signals from the helicase core and the MBP-tag were broadened beyond detection (Supplementary Fig. 8a). Notably, the chemical shifts of the observed N-IDR signals were consistent with those in the isolated N-IDR, except for residues in the middle to C-terminal portion (after residue 34), whose signals were severely broadened, similar to those from the core and MBP regions. This suggests that the middle-to-C-terminal portion of the N-IDR forms some interactions with the helicase core, restricting its structural flexibility. Similar interactions have recently been proposed for the yeast counterpart Ded1p as well⁵³. Although it remains challenging to characterize these potential N-IDR-core interactions, they are likely weak and transient in nature. This is supported by a clear difference in elution volume (~0.5 mL) in analytical size exclusion chromatography between DDX3X constructs with and without the N-IDR, indicating a substantial difference in hydrodynamic radius, which would be expected when the N-IDR retains a flexible conformation rather than being tightly associated with the core (Supplementary Fig. 2c).

Although only a limited number of NMR probes from the N-IDR were available due to line broadening from these potential N-IDR-core interactions, we were still able to monitor the N-IDR signals in [U-¹⁵N] N-Core-MBP to evaluate binding to GQ RNA. As in the isolated N-IDR, signal intensity reductions were consistently observed upon the addition of NRAS GQ RNA, indicating that the interaction persists when the N-IDR is part of the larger protein construct, while retaining its disordered nature upon binding (Supplementary Fig. 8b). Notably, these reductions in intensity were less pronounced with the NRAS mutant RNA, suggesting that the structural selectivity is preserved even in the full-length background (Supplementary Fig. 8c).

(Supplementary Fig. 8)

Supplementary Figure 8 NMR characterization of the N-IDR in the full-length protein context. (a) ^{15}N - ^1H HSQC spectrum of $[\text{U}-^{15}\text{N}]$ -labeled N-Core-MBP in the absence of RNA (navy, multiple contours). The spectrum of the isolated $[\text{U}-^{15}\text{N}]$ -labeled N-IDR is overlaid (coral, single contour). (b) 1D slices of the L21, N22, and N26 signals measured in the free state (left), in the presence of 0.33 eq. NRAS GQ RNA (center), or in the presence of 0.33 eq. NRAS mutant RNA (right). (c) Plots of peak height ratios in the presence and absence of 0.33 eq. NRAS GQ RNA (left) or NRAS mutant RNA (right). The average peak height ratio across all residues is indicated by a dashed horizontal line. All NMR measurements were performed at 10 °C and 1 GHz in a buffer containing 20 mM potassium phosphate (pH 7.0), 200 mM KCl, 5 mM DTT, 260 units/mL RNasin® Plus RNAase inhibitor, and 5% D_2O . The protein concentration was 30 μM , and the RNA concentration was 10 μM .

10. The authors must confirm some of the statements on the role of the N-IDR in its tethered state to the D1/D2 core. The interaction between the N-IDR and RNA could be affected by the presence of the core (as somewhat the data in Figure 5f would suggest – see also point 8). Since the author wants to provide a wholesome description of this system, they must observe the interaction of the RGG boxes when these are tethered to the folded domains.

Without this, the whole synergistic description in the Discussion at page 22 is way too speculative to be included.

We thank the reviewer for raising this important point. To address whether the interaction between the N-IDR and RNA is affected by the presence of the core, we performed NMR experiments to observe the N-IDR-GQ RNA interactions using the [U-¹⁵N] N-Core-MBP. As we have already shown some of the results in our response to point 9 above, the N-IDR interacts with NRAS GQ RNA in a manner similar to that observed with the isolated N-IDR, retaining its flexible structure in both the free and RNA-bound states in the full-length context. Additionally, we assessed the binding preference of the N-IDR to GQ RNA in the full-length form. Upon addition of NRAS mutant RNA, the observed intensity reductions were less pronounced than those seen with the NRAS GQ RNA, indicating that the structural selectivity is preserved even in the full-length background. These results strongly suggest that the N-IDR-RNA interactions are not significantly affected by the presence of the core.

(Supplementary Fig. 8)

Supplementary Figure 8 NMR characterization of the N-IDR in the full-length protein context. (a) ¹⁵N-¹H HSQC spectrum of [U-¹⁵N]-labeled N-Core-MBP in the absence of RNA

(navy, multiple contours). The spectrum of the isolated [U-¹⁵N]-labeled N-IDR is overlaid (coral, single contour). (b) 1D slices of the L21, N22, and N26 signals measured in the free state (left), in the presence of 0.33 eq. NRAS GQ RNA (center), or in the presence of 0.33 eq. NRAS mutant RNA (right). (c) Plots of peak height ratios in the presence and absence of 0.33 eq. NRAS GQ RNA (left) or NRAS mutant RNA (right). The average peak height ratio across all residues is indicated by a dashed horizontal line. All NMR measurements were performed at 10 °C and 1 GHz in a buffer containing 20 mM potassium phosphate (pH 7.0), 200 mM KCl, 5 mM DTT, 260 units/mL RNasin® Plus RNAase inhibitor, and 5% D₂O. The protein concentration was 30 μM, and the RNA concentration was 10 μM.

One of the main conclusions of this study is that the binding preference toward structured RNA is encoded within the N-IDR of DDX3X. Therefore, it is important to test whether this binding preference for structured GQ RNA, as observed using the isolated N-IDR, is preserved in the context of the full-length protein. To address this, we have conducted additional EMSA using the full-length DDX3X proteins. We monitored the binding between the full-length DDX3X with an N-terminal MBP tag (MBP-FL) and various RNA ligands (see Supplementary Fig. 9 below). Consistent with the NMR results obtained using the the isolated N-IDR, MBP-FL exhibited high-affinity binding to telomeric GQ RNA ($K_d = 54 \pm 14$ nM) and NRAS GQ RNA ($K_d = 43 \pm 4$ nM), while showing little to no band-shift for poly-U₁₂ ssRNA or 14mer tetraloop RNA, and significantly weaker binding to GC-14mer dsRNA ($K_d = 665 \pm 31$ nM). The apparent affinity for NRAS mutant RNA ($K_d = 222 \pm 9$ nM) was about 5-fold weaker than that for NRAS GQ RNA, supporting a structural preference for the GQ moiety in the full-length construct.

Together, these NMR and EMSA results strongly support our conclusion that the RNA-binding properties of the N-IDR are largely retained in the full-length DDX3X construct. To clarify these points, we have added the following explanations and supplementary figures in the revised manuscript.

(Page 13, line 19)

To further validate the binding preference, we performed EMSA experiments using MBP-FL as an orthogonal test. Consistent with results from the isolated N-IDR, MBP-FL exhibited high-

affinity binding to telomeric GQ RNA ($K_d = 54 \pm 14$ nM) and NRAS GQ RNA ($K_d = 43 \pm 4$ nM), while showing little to no band-shift for poly-U₁₂ ssRNA or 14mer tetraloop RNA, and significantly weaker binding to GC-14mer dsRNA ($K_d = 665 \pm 31$ nM) (Supplementary Fig. 9a). The apparent affinity for NRAS mutant RNA ($K_d = 222 \pm 9$ nM) was about 5-fold weaker than that for NRAS GQ RNA, supporting a structural preference for the GQ moiety in the full-length construct. Notably, the N-IDR-truncated construct, MBP-Core-C, exhibited significantly reduced affinity for both telomeric GQ RNA ($K_d = 579 \pm 69$ nM) and NRAS GQ RNA ($K_d = 201 \pm 5$ nM), strongly suggesting that the N-IDR plays a major role in GQ-specific binding (Supplementary Fig. 9b). Collectively, these NMR and EMSA results provide strong evidence that DDX3X preferentially recognizes GQ RNA, and that this selectivity is primarily mediated through N-IDR-RNA interactions.

(Supplementary Fig. 9)

Supplementary Figure 9 EMSA analysis of the interaction between MBP-FL/MBP-Core-C and various RNA ligands. EMSA binding experiments were performed for MBP-FL (a) or MBP-Core-C (b) using poly-U₁₂-FAM ssRNA, GC-14mer-FAM dsRNA, FAM-14mer tetraloop RNA, FAM-NRAS GQ RNA, FAM-telomeric GQ RNA, and FAM-NRAS mutant RNA. Gel images and fitted binding curves used to estimate K_d values are shown. MBP-FL protein was titrated from 0 to 1 μ M. The bound fraction was estimated from the intensity of the free RNA probe and fitted to a standard Hill-type equation to obtain apparent K_d values. Free and bound probes were separated using polyacrylamide gels: 8% for all RNAs except the 14mer tetraloop, which was run on a 12% gel. All gels were prepared with 0.5 \times TBE buffer. Error bars represent the standard deviation ($n = 3-5$), with center indicating mean values.

1. Page 3: there is an extra space between that and unwinds in the sentence “an enzyme that unwinds structured RNA...”.

2. Page 11: there is an extra space between first and conducted in the sentence “we first conducted NMR binding experiments under....”

We thank the reviewer for pointing this out. The extra spaces have been removed in the revised manuscript.

3. Page 26: reference missing for inverse PCR cloning method.

We thank the reviewer for pointing out the missing reference. We have added the following reference in the revised manuscript.

96. Liu, H. & Naismith, J. H. An efficient one-step site-directed deletion, insertion, single and multiple-site plasmid mutagenesis protocol. *BMC Biotechnol.* **8**, 91 (2008).

4. Page 27: D-glc has been reported with the D in superscript.

We thank the reviewer for carefully reviewing our manuscript. We have corrected the term “D-glucose” in the revised text.

5. En dash should be preferred to hyphen when intervals are reported (for example, page 28: 132–607 instead of 132-607).

We thank the reviewer for pointing this out. We have revised the manuscript to use en dashes instead of hyphens when indicating intervals throughout the text.

6. Page 28: 1,000 mM should be corrected into 1000 or 1'000.

We thank the reviewer for pointing this out. We have revised the manuscript to use “1000 mM” throughout the text.

7. Page 29: reference missing for the nearest-neighbour approach.

We thank the reviewer for pointing out the missing reference. We have added the following reference in the revised manuscript.

99. Cantor, C. R., Warshaw, M. M. & Shapiro, H. Oligonucleotide interactions. 3. Circular dichroism studies of the conformation of deoxyoligonucleotides. *Biopolymers* **9**, 1059–1077 (1970).

8. Page 33: 10 °C should not carry a space.

We thank the reviewer for pointing out the extra space before °C. We have corrected this throughout the manuscript.

9. Page 33: the word paper is misspelled in “paepr by Caliendo et al”.

We thank the reviewer for catching the typo, which has been corrected in the revised manuscript.

We hope that these revisions have adequately addressed all the points raised by the reviewers and that the manuscript appears suitable for publication in *Nature Communications*. I’m looking forward to hearing from you at your earliest convenience.

Ichio Shimada, Ph.D.

Professor Emeritus

The University of Tokyo

Team director

Laboratory for Dynamic Structure of Biomolecules

RIKEN Center for Integrative Medical Sciences (IMS)